# All-optical dissipative discrete time crystals

Hossein Taheri [1✉], Andrey B. Matsko [2], Lute Maleki[3] & Krzysztof Sacha [4]

Time crystals are periodic states exhibiting spontaneous symmetry breaking in either time-independent or periodically-driven quantum many-body systems. Spontaneous modification of discrete time-translation symmetry in periodically-forced physical systems can create a discrete time crystal (DTC) constituting a state of matter possessing properties like temporal rigid long-range order and coherence, which are inherently desirable for quantum computing and information processing. Despite their appeal, experimental demonstrations of DTCs are scarce and significant aspects of their behavior remain unexplored. Here, we report the experimental observation and theoretical investigation of DTCs in a Kerr-nonlinear optical microcavity. Empowered by the self-injection locking of two independent lasers with arbitrarily large frequency separation simultaneously to two same-family cavity modes and a dissipative Kerr soliton, this versatile platform enables realizing long-awaited phenomena such as defect-carrying DTCs and phase transitions. Combined with monolithic microfabrication, this room-temperature system paves the way for chip-scale time crystals supporting real-world applications outside sophisticated laboratories.

[1] Department of Electrical and Computer Engineering, University of California at Riverside, 3401 Watkins Drive, Riverside, CA 92521, USA. [2] Jet Propulsion Laboratory, California Institute of Technology, 4800 Oak Grove Drive, Pasadena, CA 91109-8099, USA. [3] OEwaves Inc., 465 North Halstead Street, Suite 140, Pasadena, CA 91107, USA. [4] Instytut Fizyki Teoretycznej, Uniwersytet Jagielloński, ulica Profesora Stanisława Łojasiewicza 11, PL-30-348 Kraków, Poland. ✉email: hossein.taheri@ucr.edu

Symmetry is a central concept and unifying theme of physics. In quantum mechanics, stationary solutions of the Schrödinger equation must follow the symmetries of the system Hamiltonian. A many-body system, however, may fail to obey these symmetries. The formation of ordinary spatial crystals constitutes a well-known example in which a periodic distribution of atoms prevails even though space translation symmetry in the governing theory does not favor any particular location in the system[1]. This epitomizes spontaneous symmetry breaking (SSB), perhaps the most significant aspect of the universal notion of symmetry which appears in various branches ranging from condensed matter physics to the standard model of electroweak interactions[2]. At the same time, while for centuries physicists treated space and time on different footings, Einstein underscored their shared relative nature and combined them into *spacetime*. In 2012, Frank Wilczek wondered if this similitude could be extended by searching for a temporal analogue for the SSB of ordered atoms in crystalline solids[3]. This quest flamed a heated scientific debate[4–7], kindled much excitement, and gave rise to a prolific area of research centered around what is now considered a new phase of matter, *time* crystals (TCs).

While the original search for TCs targeted continuous time translation symmetry (TTS) in non-driven quantum systems[3], it was soon realized that intriguing aspects of temporal SSB can rather be observed in periodically driven, so-called Floquet, non-equilibrium systems. An isolated many-body quantum system driven by a periodic external force may spontaneously assume a stable state evolving with a period different from that of the drive[8–10]. A *discrete* time crystal (DTC) emerges when the period of the system response is an integer multiple of the drive period. DTCs now lie at the focus of intense research, with various time crystalline states, such as fractional DTCs and quasi-crystals, proposed and yet waiting to be observed in experiments; see refs. [11–13] and references therein. In isolated translationally invariant Floquet systems, interactions and disorder can suppress driving-induced thermalization to create infinitely long-lived DTCs. Furthermore, exponentially long-lived DTCs can arise in such systems in the absence of disorder when the external forcing frequency is much larger than the system's local energy scales. Understanding ergodicity breaking mechanisms in DTCs is another avenue of current active research[14].

Closed systems, while a significant stepping stone, do not capture the full potential of TCs. In fact, the study of DTCs in *dissipative* open quantum systems is highly germane and immensely important, especially when TC applications, e.g., for quantum memory and computation, are concerned. Dissipation can heat up driven isolated quantum many-body systems and destroy time crystallinity. Yet, coupling to an external environment drains energy from a driven system and a periodic steady state can prevail[15–17]. Only recently has the investigation of dissipative DTCs gained traction[18–26].

We report in this work the experimental demonstration and theoretical study of all-optical, room-temperature, quantum DTCs in Kerr-nonlinear microresonators. This demonstration relies on the simultaneous self-injection locking of two independent continuous-wave (CW) pump lasers to two same-family cavity modes with arbitrary multi-FSR (free spectral range) frequency separation and a dissipative optical soliton. A DTC is realized when the periodicity of the soliton pulse train becomes an integer multiple of the drive period defined by the beating of the pumps through robust subharmonic generation. The existence of patent discrete TTS and its spontaneous breaking via the emergence of discrete symmetry with a larger integer-multiple period distinguishes DTC formation in our system from subharmonic entrainment[27,28]. Compared to a recent theoretical proposal for continuous-time SSB and boundary TCs in coupled Kerr cavities[29], our work demonstrates discrete-time SSB, and moreover in a simpler experimental setup constituting one resonator. We observe different *m*-tupling DTCs in which the integer *m* (the response-to-drive period ratio) can be chosen much larger than 2, hence readily realizing recently-predicted so-called *big* DTCs[30–33]. We present a thorough analysis, emphasizing that DTC size *m* and its state can be tuned by frequency tuning of the external laser pumps, and illustrating that these photonic DTCs possess temporal long-range order and can be realized robustly over a range of system parameters. We explore possible extensions to new states such as imperfect DTCs, the temporal counterpart of solid-state crystals with defects (i.e., vacancies, dislocations, and interstitials)[34]. Our results inaugurate an ideal photonic platform to experimentally demonstrate a whole set of time crystalline phenomena which not only have remained inaccessible in hitherto explored systems, but some have not even been deemed possible thus far. This platform will empower future theoretical and experimental investigations of long-sought DTC properties such as phase transitions, and support applications demanding coherence and precision like quantum computation and timekeeping. Combined with the highly developed microfabrication of monolithic optical high-Q (quality factor) resonators, our proposed system can lead to the demonstration of compact low–phase-noise photonic frequency dividers as well as chip-scale DTCs, not only creating unprecedented opportunities for further exploration of the physics of TCs, but also paving the way for liberating them from complex laboratories and adopting TCs in real applications.

## Results

**Concept**. Many-body Hamiltonian of a bosonic system can possess spatial or temporal translation symmetry which may be spontaneously broken if bosons interact sufficiently strongly. Formation of optical or matter-wave solitons—robust solitary waves preserving their shape upon propagation—in systems of photons or massive particles exhibits a prominent example studied theoretically and experimentally over the past few decades[35–43]. These symmetry-broken states represent stable solutions which can accurately be described through mean-field models.

In ultra-cold atomic ensembles, if inter-atomic interactions are attractive and sufficiently strong, it becomes energetically favorable for the atoms to group into a single localized wave-packet evolving along a classical orbit with a period that is an integer multiple of the external drive periodicity[13]. Such a Bose–Einstein condensate (BEC) breaks the discrete TTS, thereby realizing a DTC. In the optical platform utilized here, a localized photon wave-packet (a self-synchronized dissipative optical soliton) arises from the nonlinear interaction of photons in the Kerr resonator[44,45]. While a monochromatically pumped Kerr microcomb does not define a discrete TTS (see the Supplementary Information), two energetic pumps with judiciously chosen power and frequency can excite subharmonics and hence break the discrete symmetry defined by their beatnote. Specifically, the simultaneous driving of a high-Q Kerr-nonlinear resonator at two different frequencies $f_{P_1}$ and $f_{P_2}$ primarily creates a periodic pattern (henceforth called the modulated background waveform) rotating around the cavity, whose periodicity is dictated by the spectral spacing between the two pump frequencies. For two pumps separated by $M$ FSRs of the resonator ($M$ being an integer), the periodicity is given by $T = 1/\left(f_{P_2} - f_{P_1}\right) = T_R/M$, where $T_R = L/v_g$ is the round-trip time of the cavity, $L$ denotes the resonator circumference and $v_g$ the group velocity at the relevant frequency range. The modulated background creates a potential

grid in the resonator and acts as a *rotating lattice trap* for optical solitons. For a certain region in the power vs. pump-resonance detuning plane for the driving lasers, the system can spontaneously generate one or multiple solitons per cavity round-trip time which will reside on top of the background pattern, trapped in the potential array. The periodicity of the resulting pulse train will in general differ from that of the modulated background and effective driving field ($T$), giving rise to discrete temporal SSB. Depending on the number of soliton peaks per cavity round-trip time and their distribution with respect to the modulated background, the periodicity of the generated pulse train will be $mT$, where $1 \le m \le M$ is an integer. Period multiplication and DTC formation occurs for $m > 1$.

Were we interested in spatial crystals, we would inquire about the periodicity of the system in space at a fixed moment of time, i.e., the moment of detection. We are, however, interested in temporal crystalline structures and should therefore exchange the roles of space and time to observe the TTS described above and its spontaneous breaking. We choose a spatial point in the path of the photons (i.e., the location of an imaginary detector) and observe whether the probability of the detector clicking (counting photons) evolves periodically with time[13]. Two examples illustrating the two different cases of even and odd integer $M$ (i.e., $M = 8$ and $M = 5$ FSR spacing between the pumps with periodicity $T_1 = T_R/8$ and $T_2 = T_R/5$, respectively) are shown in Fig. 1. Here we look at the collective photon waveform in a corotating reference frame in a ring-shaped resonator, which can be accurately modeled using a variant of the nonlinear Schrödinger equation (NLSE), including detuning, damping, and two-pump driving; see the Methods Section. The potential lattice created by the beating of the pumps is shown in Fig. 1a, e. For $M = 8$, if a soliton is spontaneously formed in every second well of the lattice potential, as in Fig. 1b, discrete TTS will be broken such that the

system evolves with period $2T_1$, giving rise to a period-doubling DTC. If, however, a trapped soliton is missing from every fourth well, then SSB forms a period-quadrupling DTCs (i.e., period $4T_1$); Fig. 1c. Finally, if only one soliton per round-trip spins around the resonator, or if the positioning of multiple solitons in the periodic lattice does not possess any certain symmetry, the pattern should traverse a full circle to resume its original shape and a period-octupling DTC, one evolving with periodicity $8T_1$, will be created; see Fig. 1d for an example depicting 6 solitons asymmetrically distributed around the resonator circumference. We note that if one soliton is trapped in each of the potential wells, then discrete TTS will not be violated. In this case, when the system is observed in the laboratory frame (e.g., as the pulse train couples out of the microresonator), a soliton arrives every subsequent driving period $T_1$ at a given point in space.

For $M = 5$ too, when one soliton is trapped in each potential well, discrete-time SSB will not occur; (Fig. 1h). Symmetry considerations, however, reduce the variety of observable DTCs in this case. If only one soliton is trapped in the modulated background potential lattice (Fig. 1f), or if all but one of the potential wells host a soliton (Fig. 1g), then TTS will be spontaneously broken and a period-quintupling DTC will be formed (periodicity $5T_2$). Different ratios of the response to drive periodicity, and hence realizations of discrete TTS breaking, can be envisioned for other pump separations. For instance, if solitons sit in every third periodic lattice site in a resonator driven by two pumps spectrally spaced by $M = 9$ FSRs, then a period-tripling DTC will be realized. Indeed the DTC platform introduced here can readily accommodate $m$-tupling DTCs with $m \gg 1$, creating so-called *big* DTCs with dramatic SSB[33].

We should emphasize that DTCs can be formed in a microresonator driven by two pumps over a range of laser frequencies and powers. Figure 2a, b show steady-state comb

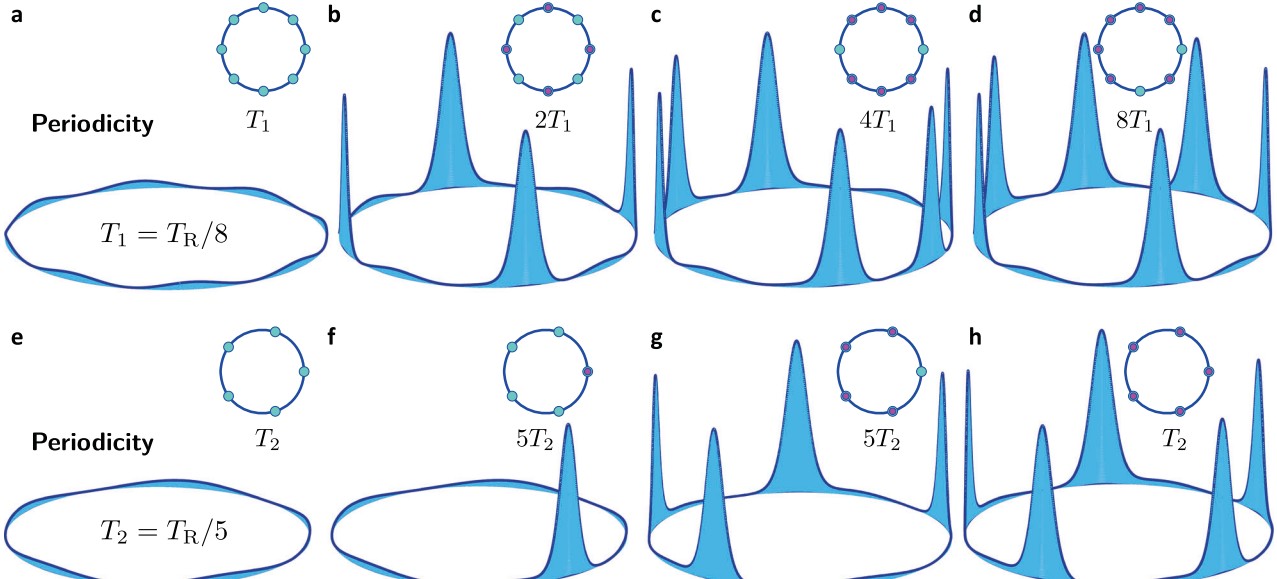

**Fig. 1 DTCs in a Kerr nonlinear optical resonator pumped by two lasers.** Symmetry breaking, period multiplication, and DTC formation in an optical resonator pumped by two lasers separated by **a–d** $M = 8$ and **e–h** $M = 5$ cavity FSRs. **a, e** The beating between the two pumps modulates the background waveform and creates a rotating lattice trap near the resonator periphery, where optical modes are localized. **a** For $M = 8$ the temporal periodicity of the lattice trap is $T_1 = T_R/8$. **b** Spontaneous formation of a soliton in every second well of the lattice potential breaks the discrete TTS giving rise to a period-doubling DTC (periodicity $2T_1$). **c** When a trapped soliton is missing in every fourth potential well, a period-quadrupling DTC (with period $4T_1$) is formed. **d** If the distribution of multiple solitons in the lattice does not possess any certain symmetry, a period-octupling DTC (with periodicity $8T_1$) will be created. Here, 6 solitons are asymmetrically trapped around the lattice. **e** Lattice periodicity is $T_2 = T_R/5$ for $M = 5$. Symmetry considerations reduce the variety of observable DTCs in this case. When only one soliton is trapped in the potential lattice (**f**), or when all but one of the potential wells host a soliton (**g**), a period-quintupling DTC is formed (periodicity $5T_2$), while if one soliton is trapped in each potential well, discrete TTS will not be broken (**h**). Insets illustrate top-view schematics with an empty circle (cyan) for each lattice trap and a filled circle (magenta) denoting a soliton.

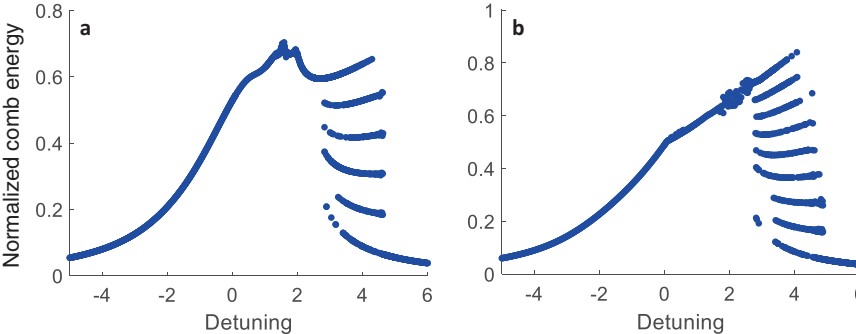

**Fig. 2 DTC regions of existence.** Comb energy versus detuning curves for **a** 5-FSR and **b** 8-FSR separation between the two pumps, when laser powers are kept fixed. Each data point corresponds to one numerical integration of the governing equations from a random initial waveform till a steady state (for non-chaotic states) prevailed. Spontaneous soliton formation occurs in step-like regions to the right of the curves. Dispersion parameter matching the experimental resonator ($D_2 = 2\pi \times 6.8$ kHz) was used. The horizontal axis shows the normalized detuning of the stronger pump ($-\sigma_1/\kappa$), for fixed detuning difference between the lasers ($\sigma_2 = \sigma_1 + D_2 M^2/4$); see Methods. In **a** normalized pump powers were (1.4, 0.9), and in **b** they were (1.5, 0.99).

energy vs. pump-resonance detuning, respectively, for $M = 5$ and $M = 8$, where the horizontal axis is the detuning of the stronger pump normalized to the resonator half-width at half-maximum (HWHM) linewidth while the vertical axis is the normalized intra-cavity energy. Each data point in Fig. 2 indicates one simulation run till steady states, for non-chaotic microcombs, prevailed. Here, the step-like multi-stability region on the right of each curve denotes where discrete TTS breaking occurs, the lowest branch corresponding to the modulated background [cf. Fig. 1a, e]. It is seen that in each case, the number of step-like branches (disregarding the lowest step) equals $M$, because the two pumps separated by $M$ FSRs discretize the resonator circumference (i.e., each round-trip time) and create preferable residing lattice sites for the spontaneously formed solitons. Each step pertains to a constant number of solitons per round-trip time, starting from 1 (the branch right above the lowest) and ending with $M$, the top-most branch. As remarked earlier, depending on the positioning of the solitons with respect to the background lattice, a different $m$-tupling DTC will be realized.

The introduced DTCs are robust with respect to system parameters and possess temporal long-range order. We have verified that all of the various TTS breaking states depicted in Fig. 1 are stable over hundreds of cavity photon lifetimes (thousands of the driving period, and much longer than the lifetime of DTCs demonstrated in initial experiments[46,47]). Two examples are depicted in Fig. 3a, d; cf. Fig. 1g, c. Figure 3b, e show snapshots of the pulse trains at the end of the integration time in Fig. 3a, d, respectively. The horizontal axis in panels (a, b, d, e) is $\theta$, the azimuthal angle around the circular resonator, which is related to the fast time $\tau$ through $\theta = 2\pi\tau/T_R$; see the Methods Section. The tilted trajectory of the soliton peaks moving upward in Fig. 3a, d depends on the frequency separation between the pumps and their detuning from respective resonances, as well as a slight soliton center frequency shift (recoil) resulting from the presence of the second pump[48]. This tilt amounts to a fixed-speed rotation around the resonator and can be removed with a change in the angular velocity of the co-moving reference frame.

Before presenting experimental results, we note that periodic pulse trains correspond to frequency combs in the conjugate Fourier domain, i.e., an equidistant array of frequency tones spaced by the repetition rate $1/T_R$[49]. The frequency comb spectra of the DTC states of Fig. 3b, e are plotted in Fig. 3c, f. The horizontal axis shows frequency normalized to the resonator FSR, and the spectrum is centered on one of the pumped mode frequencies labeled 0. The red arrows mark the pumps and the red dashed lines show hyperbolic secant (sech) soliton envelopes in Fig. 3c, f. Multiple frequency tones are seen to be generated

between the two pump harmonics in both representative examples. This *subharmonic* generation in the frequency domain accompanies periodic multiplication in the time domain which was discussed earlier and is a signature of DTCs.

**Experiments**. For the experimental demonstration of the photonic DTCs introduced above, we utilized a prism-coupled high-Q whispering gallery mode (WGM) magnesium fluoride (MgF$_2$) crystalline resonator of 1.06 mm radius (32.8 GHz FSR), pumped by two lasers. The setup schematic is illustrated in Fig. 4a; see the Methods Section for further information on resonator preparation, pulse train generation, and measurements. The radio frequency (RF) signal was produced by the output-coupled optical wave demodulated on a fast photodiode to ensure the coherent nature of the generated pulse train. For the two pumps locked to two adjacent cavity modes, a narrow RF signal is generated; see the blue curve in Fig. 4b. We observed that the generation of subharmonics and solitons riding the modulated background when the pumps lock to two non-adjacent resonator modes further reduces the phase noise at 32.8 GHz, narrowing the RF signal peak as a result of frequency division and the stronger mutual coupling between the harmonics. One example for 3 FSRs separating the pumps is plotted by the red curve in Fig. 4b.

Two representative spectra corresponding to SSB and DTC formation are plotted in Fig. 4c, f, respectively for $M = 4$ and $M = 3$ FSRs separating the driving lasers. The pumps are marked by red arrows and the red dashed curves show hyperbolic secant envelopes. The stronger pump was kept nearly at 1545 nm in both cases. The matching numerical modeling spectra are depicted to the right of the experimental data, in Fig. 4d, g; excellent agreement between experiment and theory is observed. The corresponding time-domain pulse trains per round-trip time are shown on the far right, in panels (e, h) of Fig. 4, with snapshots of the pulse trains revolving around the resonator circumference as insets. From the insets, it is obvious that Fig. 4c–e represent a period-quadrupling DTC while Fig. 4f–h demonstrate a period-tripling DTC.

In our experiment, we have observed multi-stability similar to what is depicted in Fig. 2, where middle branches between the lowermost and uppermost soliton steps correspond to different DTC realizations; see also Supplementary Figs. 3c and 4 (blue curves). These multi-stable states consisted of (1) four-wave mixing (FWM) of the two pumps creating harmonics separated by their beatnote (as in Supplementary Fig. 1b), and (2) subharmonic generation between the pumps, which correspond to DTC formation (as in Fig. 4c, f, with RF beatnote plotted in red in Fig. 4b). Note that different subharmonic arrays translate into

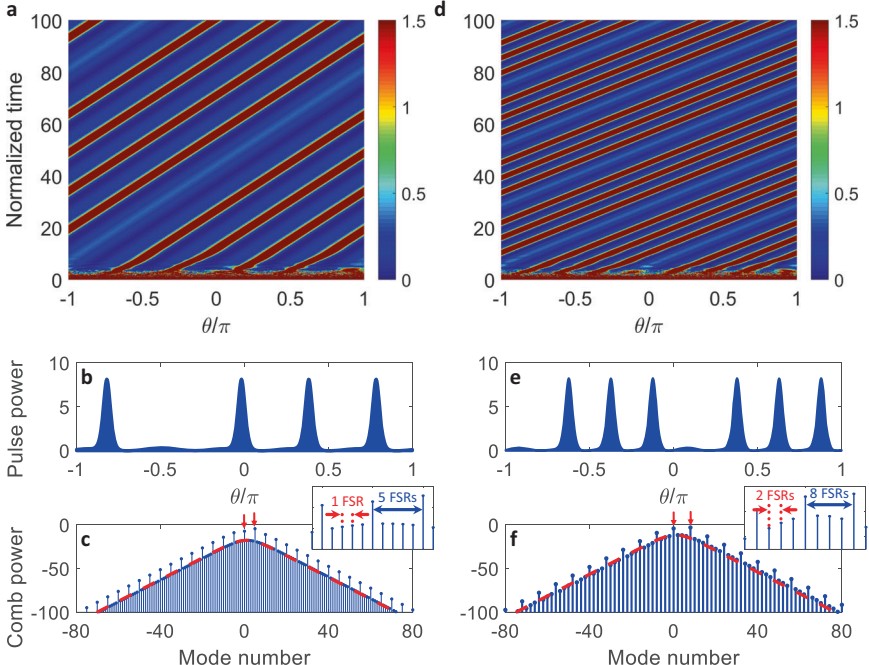

**Fig. 3 Temporal long range order of the generated DTCs.** Representative numerical integration results demonstrating temporal long-range order in two of the states in Fig. 1. Left panels **a–c** correspond to Fig. 1g, and right panels **d–f** to Fig. 1c. **a, d** Temporal evolution of pulse trains over 100 normalized times (200 cavity photon lifetimes and thousands of the driving period). Pulses are initialized through hard excitation, the horizontal axis is the angle around the resonator, and the vertical axis is the evolution time. Dark red lines denote soliton trajectories. **b, e** Snapshots of the pulse trains at the end of the integration time in **a, d**. **c, f** Frequency spectra (logarithmic scale) of the DTC states plotted in **b, e**. Each spectrum is centered on one of the pumped modes labeled 0, and the horizontal axis shows frequency normalized to the resonator FSR. Vertical red arrows mark the pumps and red dashed curves show hyperbolic secant envelopes. Insets are zoomed-in spectra around the center, showing the beatnote of the pumps (blue) and the separation of adjacent subharmonics (red). Generation of multiple subharmonics between the pumps is manifest. Numerical parameters match those of Fig. 2, with $-\sigma_1/\kappa = 4$.

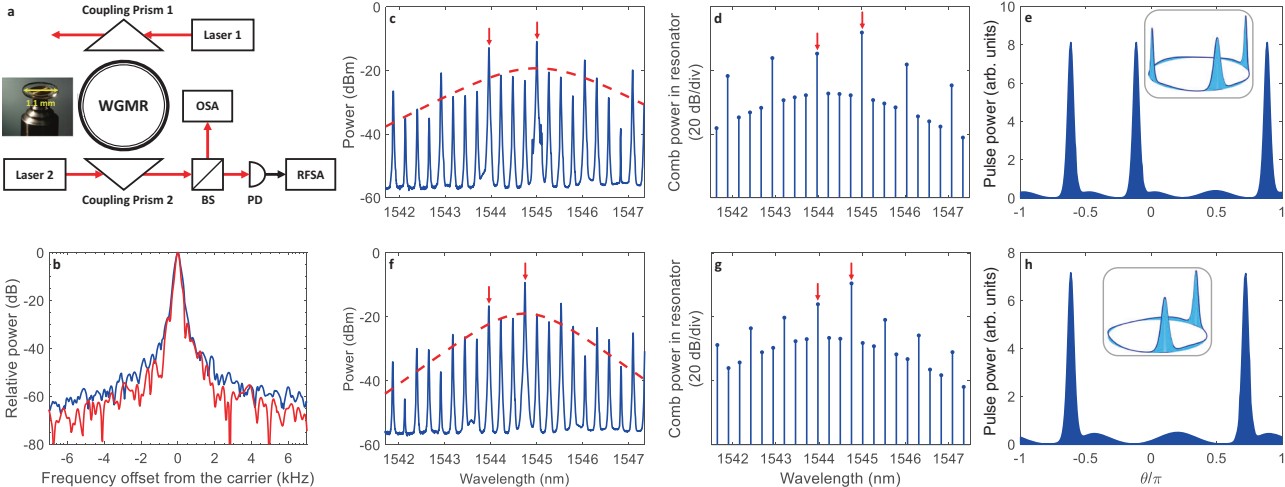

**Fig. 4 Experimental observation of all-optical DTCs. a** Experimental setup; WGMR: whispering gallery mode resonator, OSA: optical spectrum analyser, RFSA: radio frequency spectrum analyser, BS: beam splitter, PD: photodetector. **b** RF signal ensuring stable pulse train generation. For the two pumps locked to two adjacent cavity modes, a narrow RF signal is generated (blue, 32.8 GHz carrier frequency). When locking to non-adjacent modes, the generation of frequency harmonics between the pumps (subharmonics) and hence solitons trapped in the background lattice further narrows the RF signal peak. An example is the red curve (32.8 GHz carrier frequency), corresponding to the experimental spectrum plotted in **f** in which $M = 3$. **c, f** Spectra corresponding, respectively, to period-quadrupling ($M = 4$) and period-tripling ($M = 3$) DTCs. Pumps are marked by red arrows, and red dashed curves show soliton envelopes. The stronger pump was at 1545 nm in **c** and 1544.8 nm in **f**. **d, g** Numerical modeling spectra matching **c** and **f**. **e, h** Time-domain pulse trains per round-trip time (numerical), with snapshots of pulses revolving around the resonator circumference as insets.

various pulse numbers per round-trip time. After finding the appropriate parameter regime for subharmonic generation in our experiments, we always observed one of the states (1) or (2) in every subsequent run. In particular, in every realization of subharmonics between the two pumps, we observed the same envelope and a narrow RF beatnote, hinting at the stable nature of the realized DTC states.

## Discussion

A periodically forced dissipative many-body system qualifying as a DTC meets certain criteria. First, it possesses discrete TTS, evident from the time-dependence of its equation of motion. However, steady states (i.e., steady-state solutions of the equation of motion) evolving with a cycle that is an integer multiple of the period dictated by the drive can emerge spontaneously in the system. Second, the symmetry-broken steady states emerge without relying on fine tuning and are stable over a range of parameters. Third, the many-body system is in the thermodynamic limit and the symmetry-broken states can be accurately described by a mean-field approach. Subharmonic generation in the dichromatically pumped Kerr microcavity system introduced in this work qualifies as a dissipative DTC because it meets all of the said criteria. The two pumps define a discrete TTS in the system, manifest in the equations of motion (detailed in Methods), which is broken by the realization of certain periodic steady states of the system. These states, accompanied by subharmonic generation, demonstrate SSB at integer response to drive periodicity ratios. The subharmonics in symmetry-broken states emerge robustly, both in numerical modeling and experiments, with various (e.g., 2-, 3-, and 4-FSR) frequency separations of the pumps and possess temporal long-range order; see also Supplementary Information, Section V. Finally, the DTCs are in the thermodynamic limit of infinitely many photons and are well captured by a mean-field model (the modified LLE, see Methods).

Besides the DTC states introduced thus far, it is possible to create a nearly perfect $m$-tupling DTC with a missing, dislocated, or extra soliton spoiling crystalline symmetry. Such defective DTCs mirror spatial crystal defects such as vacancies, dislocations, and interstitials in condense matter physics[34]. Panels (a, d) in Fig. 5 depict two examples. In Fig. 5a, one soliton is missing from a period-doubling DTC while in Fig. 5d a soliton is dislocated by one lattice site (cf. Fig. 1b). The location of the missing or misplaced soliton (with respect to the perfect DTC) in each case is indicated by a red arrow in the top-view ring schematic appearing in the top left corner. The bottom panels (b, e) and (c, f) in Fig. 5 plot, respectively, snapshots of the optical pulse train per round-trip time (again, the red arrows hinting at the position of the missing or dislocated soliton) and their corresponding power spectra. In Fig. 5d, e, a dotted arrow connects the current position of the dislocated soliton to its placement in the perfect crystal. The sech-shaped envelop of the frequency spectra in Fig. 5c is the same as that shown in Fig. 3f and is not redrawn here. If an extra soliton is spontaneously formed in any of the unoccupied lattice sites in Fig. 1b, a DTC with interstitial defect will be formed. It is worth noting that each of these pulse trains has a unique frequency (Fourier) spectrum and can be identified unambiguously in the frequency domain. Moreover, the appearance of DTCs carrying defects is more conspicuous and consequential in big TCs (for $1 \ll m < M$) and Fig. 5 is intended to clearly illustrate the phenomenon. Finally, we should point out that the analogy drawn here between defects in DTCs and those in solid state spatial crystals is, strictly speaking, not exact because in the latter the spontaneous breaking of continuous, rather than discrete, TTS occurs.

The DTC platform introduced here is closely related to microresonator-based Kerr frequency combs (Kerr microcombs)[44]. While many aspects of soliton microcomb formation have been explored over the past decade, our pumping scheme combines dichromatic driving and soliton self-injection locking, thereby effectuating turnkey soliton formation in the presence of clear discrete TTS. Driving a high-Q Kerr resonator with a CW laser pump can break *continuous* TTS symmetry by generating a soliton pulse train[50]. Cherenkov dispersive wave (DW) emission in the presence of higher-order dispersion (HOD) or avoided mode crossing (AMC) will modulate the background[51–53], and trap pulse

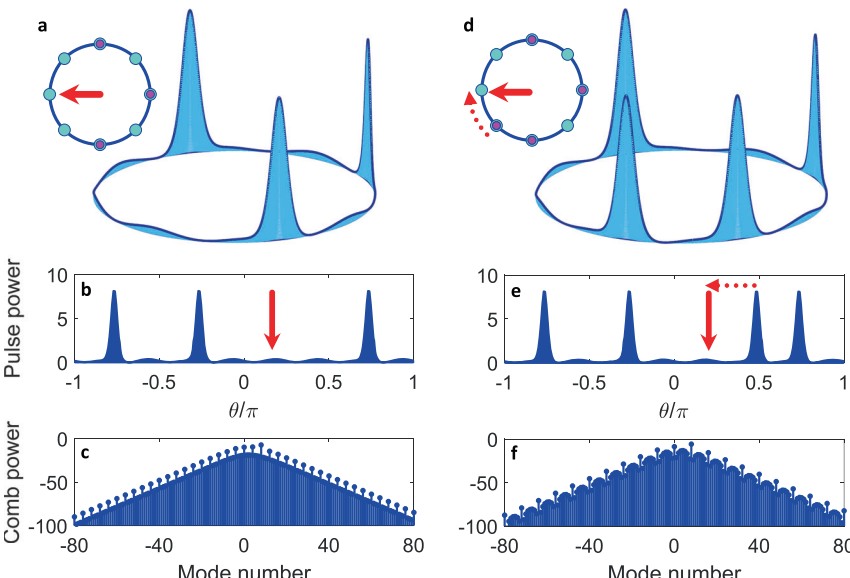

**Fig. 5 Examples of DTCs carrying defects.** A vacancy in which one soliton is missing from a period-doubling DTC **a**, and dislocation in which one soliton is misplaced by one lattice site **d**; cf. Fig. 1b. The location of the missing or misplaced soliton (with respect to the perfect DTC) in each case is indicate by a red arrow in the top-view ring schematic (insets). **b**, **e** Snapshots of the optical pulse train per round-trip time, matching the top panels **a** and **d**, respectively, with red arrows hinting at the position of the missing or dislocated soliton. **c**, **f** Power spectra corresponding to **b** and **e**. Dotted arrows in **d** and **e** connect the current position of the dislocated soliton to its placement in the perfect crystal. Numerical parameters are the same as those in Fig. 3d–f.

peaks in so-called soliton crystals[48,54]. However, as detailed in the Supplementary Information, resonator HOD and AMC do not establish any discrete TTS and, accordingly, monochromatically-pump resonators cannot host DTCs.

As described earlier, when, instead of one, two CW pumps (or another temporally structured pump wave) drive the resonator, solitons will be trapped in the potential introduced through the background modulation[48,55–57]. In our dichromatically pumped microresonator, the cavity is engineered to ensure HOD and mode anti-crossings do not interfere with soliton formation in the proximity of the pumps. Additionally, both driving lasers coherently lock to two resonator modes and a spontaneously formed soliton microcomb[48,58]. The physics of the hyperparametric process resulting in stable subharmonic generation in this platform guarantees the division of the pumps' beatnote to equal subharmonics (integer response to drive periodicity ratio) and hence elegantly allows for confirming DTC behavior through standard frequency-domain measurements without resorting to complex temporal techniques[28,59]; single-shot pulse measurement methods can facilitate further investigations into the rigidity of photonic DTCs. Independent of the resonator geometry, these phenomena are expected to occur also in integrated ring and fiber-ferrule Fabry–Pérot resonators[57]. We speculate that DTCs can similarly be observed even in other resonator or mode-locked laser types such as fiber cavities[60,61] or harmonically mode-locked lasers; in the latter case, certain frequency pinning mechanisms for ensuring robust pulse dropout are necessary. It is worth emphasizing that dual-microcomb spectroscopy setups, in contrast to our experiments, rely on two very close pump frequencies (typically different by less than 1% of an FSR) each creating its own microcomb; see ref.[44] and refs. [36–40] therein. Furthermore, addition of a second pump (or sideband) frequency to improve thermal stabilization does not lock the second frequency to the microcomb and merely adds extra power inside the cavity[62].

In light of recent advancements in integrated nonlinear and quantum photonics[44,63,64], the variety of emerging material platforms[53], and the flexibility offered by advanced dispersion engineering, revolutionary progress in the study of TCs is possible using the platform introduced in this work. Exploiting the plethora of experimental techniques and commercial equipment available to photonics for the investigation of the various aspects of TCs will prove crucial. For instance, utilizing delicate laser tuning[65] and synchronous pumping techniques[57] empowers the deterministic creation of target DTC states and exploring their interaction[66] in small experimental setups. Furthermore, building upon mode-locked laser pumping of fiber resonators[60], synchronous pumping can be used to drive solitons in larger microresonators. Generation of microcomb solitons with few-GHz range repetition rates is possible, and at these rates, efficient (i.e., with small half-wave voltage $V_\pi$) ultrafast electro-optic modulation combined with pulse shaping will furnish excellent flexibility for the controlled realization of various DTC phases and transitions between them[56,57]. Most excitingly, with progress in the monolithic integration of lasers and optical modulators, small-footprint, room-temperature, all-optical DTC may soon become a reality.

Although still in their infancy, TCs are inspiring future technology, and ideas for their applications are crystallizing. Novel paradigms for overcoming coherence limitations in quantum systems involve transitioning to the driven out-of-equilibrium regime. Indeed, among the most anticipated properties of DTCs are their robustness and temporal long-range order which enable maintaining coherence much longer than is currently possible in equilibrium systems. From the perspective of Kerr microcomb technology, the dichromatic self-injection locking pumping scheme utilized here can be used for microwave photonic

frequency division and multiplexing[58,67]; the repetition rate is locked to the beatnote of the two pumps, resulting in characteristically different behavior compared to standard monochromatically driven combs. Furthermore, two-point locking of the pumps (for instance by pinning lasers externally to frequency references such as two atomic transitions) enables stabilizing even a narrow-band microcomb and can obviate the exacting requirement of octave-spanning spectral coverage which has so far been a holy grail of microcomb research. As such, precision microcomb-based metrology and timekeeping can be extended to uncharted frequency windows and new platforms in which material dispersion currently limits comb bandwidth and consequently $f - 2f$ self-referencing[48]. Finally, recent studies[56,68] have shown several technical advantages for seeding microcomb formation with a modulated pump. Yet, available modulator bandwidths do not allow generating sidebands multiple cavity FSRs away from the pump in a microresonator, especially in smaller diameters with hundreds of GHz to THz-level FSRs. Therefore, microcomb stabilization by two independent lasers constitutes a decisive step which brings the benefits of parametric seeding to small resonators, and further enhances the possibility of miniaturization, and reduced size, weight, and power consumption for future applications. In this work we have focused on a few FSRs separating the two pumps but suspect that the marriage of dichromatic pumping with self-injection locking of both pumps to Kerr cavity modes of the same modal family involves very rich physics, warranting dedicated investigations, e.g., into the role of the beatnote of the pumps.

We have demonstrated a versatile photonic platform utilizing two pump lasers concurrently self-injection-locked to different modes of a Kerr cavity, which exhibits spontaneous breaking of discrete TTS and dissipative DTC formation. Our two-pronged theoretical and experimental study shows that in a high-Q resonator driven at two frequencies, strongly interacting photons can spontaneously generate one or multiple temporal solitons, which crystallize in the rotating optical lattice formed by the beating between the pumps through robust subharmonic generation and can give rise to DTCs with various sizes (response-to-drive period ratios). Both theory and experiment verify that the generated DTCs possess temporal long-range order. Operating in room temperature, this platform lends itself to simplified investigations of unexplored time crystalline properties such as phase transitions and mutual interactions, and along with a subsequent dissipative DTC example reported recently in a BEC of atoms in a cavity[69], it hints at the diversity of systems hosting dissipative TCs. Our results also show that integrated optics provides a robust and flexible platform for mimicking physical condensed matter phenomena associated with TCs. Paired with monolithic microfabrication and well-developed techniques of quantum integrated photonics, this demonstration empowers fieldable chip-scale DTCs, thereby paving the path for extricating the time crystalline phase of matter from complex laboratory setups and employing them in real-world applications, e.g., in quantum computation and timekeeping.

## Methods

**Equations and numerical modeling**. Soliton formation, resulting in SSB and DTC generation described in this work, can be accurately modeled using a variant of the NLSE, including detuning, damping, and driving, which is often referred to as the Lugiato–Lefever equation (LLE)[70]. Here we use an LLE variant modified for two-pump driving[48]. In the laboratory reference frame, this equation takes the form

$$\frac{\partial A}{\partial t} = \left(-\kappa - i\sigma_1 - ig|A|^2 - D_1\frac{\partial}{\partial\theta} - i\frac{D_2}{2}\frac{\partial^2}{\partial\theta^2}\right)A + \mathcal{F}(\theta, t), \qquad (1)$$

in which $\kappa$ is the resonance HWHM, $\sigma_1 = 2\pi f_{P_1} - \omega_{j_0}$ represents the detuning of the first pump from its neighboring resonance $\omega_{j_0}$, $g$ is the FWM gain, $D_1/2\pi = 1/T_R$ is the resonator FSR near the first pump, and $D_2 = \omega_{j_{0+1}} + \omega_{j_{0-1}} - 2\omega_{j_0}$ denotes the

group velocity dispersion coefficient. Cavity resonant modes are labeled with integer eigenmode numbers $j$, and $D_1$ and $D_2$ (expressed in rad/s) are the first two coefficients in the Taylor series expansion of resonant mode frequencies $\omega_j$ in eigenmode number at the first pumped resonance $\omega_{j_0}$, i.e., $\omega_j = \omega_{j_0} + D_1(j - j_0) + D_2(j - j_0)^2/2$; in the frequency range of interest, higher-order terms were negligibly small for the resonator utilized in our experiments. The variable $t$ is the evolution time (sometimes referred to as the *slow* time), and $\theta$ is the azimuthal angle around the resonator, related to the *fast* time $\tau$ via $\theta = 2\pi\tau/T_R$ (modulo $2\pi$). The intra-cavity field $A(\theta, t)$ is normalized such that $\int_{2\pi}d\theta\,|A(\theta, t)|^2/2\pi$ equals the total number of photons in the cavity at each time $t$. FWM gain is found from $g = n_2 c\hbar\omega_{j_0}^2/(n_0^2 V_{j_0})$, where $n_0$ and $n_2$ are the linear and nonlinear indices of refraction, $c$ is the vacuum speed of light, $\hbar$ is the reduced Planck constant, and $V_{j_0}$ is the effective nonlinear mode volume of the first pumped mode[53]. The drive term $\mathcal{F}(\theta, t)$ is given by

$$\mathcal{F}(\theta, t) = \sqrt{\kappa_{c_1}}\mathcal{F}_1 + \sqrt{\kappa_{c_2}}\mathcal{F}_2 e^{i\left[2\pi\left(f_{P_2} - f_{P_1}\right)t - M\theta\right]}, \tag{2}$$

where $M$, as in the main text, is the number of FSRs separating the pump frequencies (i.e., $f_{P_2} - f_{P_1} = M/T_R$), $\kappa_{c_{1,2}}$ are the coupling coefficients to the two pumps, and $|\mathcal{F}_{1,2}|^2$ represent the rate of photons pumped by the lasers[48]. The beating between the two pumps appears in the exponent of the second driving term, where $M\theta = 2\pi(M/T_R)\tau = 2\pi(f_{P_2} - f_{P_1})\tau$.

Inspection of the drive term, Eq. (2), shows that Eq. (1) is periodic and invariant under transformations $t \to t + 1/\left(f_{P_2} - f_{P_1}\right)$ or $\theta \to \theta + 2\pi/M$ [equivalently $\tau \to \tau + 1/\left(f_{P_2} - f_{P_1}\right)$]. This periodicity defines the discrete TTS of the systems, which can be spontaneously broken by soliton formation in the resonator, as described in the main text.

Equation (1) can be simplified by transitioning to a reference frame rotating with angular velocity $D_1$ (i.e., $\theta \to \theta - D_1 t$, one round per $T_R$), rendering

$$\frac{\partial A}{\partial t} = \left(-\kappa - i\sigma_1 - ig|A|^2 - i\frac{D_2}{2}\frac{\partial^2}{\partial\theta^2}\right)A + \sqrt{\kappa_{c_1}}\mathcal{F}_1$$
$$+ \sqrt{\kappa_{c_2}}\mathcal{F}_2 \exp\left\{i(\sigma_2 - \sigma_1 + D_2 M^2/2)t - iM\theta\right\}, \tag{3}$$

in which $\sigma_2$ is the pump-resonance detuning of the second pump. For numerical integration, a non-dimensional form of Eq. (3) was found by normalizing time to twice the cavity photon lifetime ($t/\kappa$), detunings and dispersion coefficient to the HWHM ($-\sigma_{1,2}/\kappa$ and $-D_2/\kappa$), and intra-cavity waveform and pump powers to the sideband generation threshold $A_{\rm th} = \sqrt{\kappa/g}$ in a monochromatically pumped cavity ($A/A_{\rm th}$ and $\sqrt{\kappa_{c_{1,2}}}\mathcal{F}_{1,2}/A_{\rm th}$). We emphasize that the $t$-dependence on the right-hand side of Eq. (1) can be removed readily by a proper change of variables (e.g., $\theta \to \theta - 2\pi(f_{P_2} - f_{P_1})t/M$), yet the resulting equation will carry a $\partial A/\partial\theta$ term, further differentiating it from a damped, driven NLSE.

It is noteworthy that Eq. (1) can be derived from an equivalent set of nonlinear coupled-wave equations, each following the temporal evolution of one frequency comb harmonic, as detailed in ref. [48]. Numerical modeling was performed using both the split-step Fourier transform [Eq. (3)] and adaptive-step Runge–Kutta integration (couple-wave equations), with excellent agreement. To properly match experimental conditions, hard excitation with high-energy initial fields was utilized. In soft excitation of a cold cavity, the effect of vacuum fluctuations was incorporated through the addition of independent noise terms.

**Resonator preparation.** The resonator was fabricated out of a magnesium fluoride ($MgF_2$) cylindrical preform by mechanical polishing. The preform rim was shaped into an oblate spheroid optimized for evanescent field coupling with a free space beam. Resonator radius was ~1.06 mm, while the radius of the vertical curvature was 0.2 mm. The resonator had an FSR of 32.8 GHz and loaded resonance bandwidth of $\sim 2\kappa = 2\pi \times 200$ kHz. Resonance loaded bandwidth was tuned by adjusting the air gap between the coupling prisms and the resonator surface. The optical power emitted by the laser never exceeded 5 mW. Roughly 3 mW of power entered the resonator because of imperfect laser beam spatial structure and other non-idealities, indicating an insertion loss smaller than 3 dB. Resonator group velocity dispersion at the pumping frequency was $\beta_2 \simeq -4.9$ ps$^2$/km, corresponding to $D_2 = 2\pi \times 6.8$ kHz. Formally, the normalized dispersion parameter $D_2/\kappa$ impacts comb generation efficiency, so through modifying the coupling, we tuned the overall conversion efficiency.

**Pulse train generation.** The resonator was pumped by two distributed feedback (DFB) lasers. To couple laser light in and out of the resonator, we utilized evanescent prism couplers made of BK7 glass. The coupling efficiency is better than 60%. The resonator generates microcombs corresponding to optical soliton trains (with hyperbolic secant spectral envelopes) if pumped with mW level optical power at around 1545 nm. The criteria for soliton pulse train generation constituted simultaneous observation of a stable sech-shaped frequency comb spectral envelope (on an OSA) as well as a low-noise radio frequency (RF) signal (on a fast photodiode)[67]. Microcomb repetition rate defines the frequency of the microwave signal. This measurement was performed using an RFSA, as shown in Fig. 4a. High spectral purity of the RF signal indicates the high degree of coherence of the frequency comb.

When two lasers pump resonator modes, as shown in Fig. 4a, generally harmonics at both laser frequencies, plus some sidebands are generated. When laser power exceeds a certain threshold and the frequency detuning is properly selected for both lasers, subharmonic generation and a mode-locked frequency comb emerges[48,55]. In both numerical modeling and experiments, the power of one laser was set above the threshold while the other had a power slightly below it. A coherent pulse train in this case was identified by an RF signal with smaller phase noise compared to the beat note of the two optical pumps directly demodulated on a photodiode; see Fig. 4b.

**Soliton self-injection locking.** To generate a stable pulse train, one has to lock each laser to its neighboring resonator mode at an optimal frequency offset. We achieved this using the self-injection locking technique[58], in which no isolator is placed between each pump and the resonator, so the laser locks to a resonator mode when a small amount of light coupled to the mode scatters back toward the laser (resonant Rayleigh scattering). Through tuning the locking point by regulating the optical phase delay between the laser chip and the resonator, the frequency of one of the lasers was shifted to a spectral position where a microcomb was observed. DTC generation was then straightforward by selecting proper pump powers. To this end, we started from higher power levels, too high to sustain a microcomb. Laser currents were then reduced till stable subharmonic generation was observed. At this stage, the power of one pump was above that required for initiating the hyperparametric process with a single laser while that of the other was below this threshold, and importantly neither pump could generate a soliton microcomb alone. One notable advantage of self-injection locking is simplified optical soliton generation in the cavity through soft excitation. In contrast, soliton microcomb generation without injection locking often relies on hard excitation or switching[50,65]. It should be stressed that the absence of strong anti-crossing–induced frequency pinning in the microcomb spectra confirms the negligible effect of AMCs in our experiments; see Supplementary Information, Sections IV and V.

## Data availability
The data that support the plots within this paper and other findings of this study are available from the corresponding author upon reasonable request.

## Code availability
The codes used for this study are available from the corresponding author upon reasonable request.

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

## Acknowledgements

H.T. and A.B.M. acknowledge helpful discussions with W. Liang and T. Herr. The research performed by A.B.M was carried out at the Jet Propulsion Laboratory, California Institute of Technology, under a contract with the National Aeronautics and Space Administration (80NM0018D0004). H.T. was supported in part by an HBCU/MSI grant from the Jet Propulsion Laboratory. K.S. acknowledges support of the National Science Centre, Poland via Project No. 2018/31/B/ST2/00349.

## Author contributions

H.T. conceived the idea, initiated and supervised the project, performed the theoretical analysis and numerical modeling, contributed to the design of experiments, and analysed the data. A.B.M. contributed to idea development, numerical simulations, and data analysis, and designed the experiments. L.M. provided resources and managed the experiments. K.S. contributed to conceptualization and data interpretation. H.T. wrote the manuscript with input from all authors. All authors reviewed and approved the final manuscript.

## Competing interests

The authors declare no competing interests.
