## [Peer Review File · Nature Communications]

REVIEWER COMMENTS

Reviewer #1 (Remarks to the Author):

The paper is to report a new concept of time crystal in the micro-cavity. The concept is interesting both for scientific community and public areas. However, I don't think the results and definition of time-crystal seems like sound.

1. The concept of time crystal was proposed by the Papp's group first, which was expected to generate the breather soliton lasers [1, 2]. Basically, it should be the fact that the breather period of breather soliton tends to be exactly integral folds of roundtrip time at the vicinity. Then, this phenomenon was demonstrated in ultrafast lasers [3]. However, this concept "time crystal" was be changed in the final publications from both two groups, instead of subharmonic entrainment [3,4].
2. Actually, I believe that the reported effect should be the phenomenon of harmonic entrainment, because the harmonic pulse phenomenon has been predicted and demonstrated in the paper. The authors should comment and explain it.
3. If the authors insist on the concept of time crystal, please present the experimental evidence of tending to exact integral folds of roundtrip time or the repetition time of pulses. Also, the authors should explain the mechanism why such pulse train should be the time crystal. I believe that the effect can be proved by dispersive temporal interferometer.
4. Creating big time crystals was proposed with ultracold atoms. The spontaneous modification of discrete time translation symmetry in a periodically driven physical system can create a discrete time crystal (DTC). However, the physical mechanism for generating DTC was not clear. Please comment this.
5. The reported phenomenon in the paper looks like harmonic mode-locking, except that there is the periodic deletion of pulses in the roundtrip circulating. The pulse deletion is a common phenomenon in harmonically mode-locked lasers. Therefore, the reported phenomenon can be considered as the harmonic entrainment. I suggest the authors use the concept "harmonic entrainment".
6. Because the paper used the mechanism of "Soliton Self-injection Locking" to explain the mechanism. So, there must be a coupling mechanism for exceptional points. The effect of exceptional points must be considered. The authors should comment or explain it well.

The results and the concept in the paper is much interesting, because the expectation of time crystal is realized in a micro cavity. I recommend the paper should be well revised before publication.

[1] Scott B. Papp and Daniel C. Cole, Kerr-breather-soliton time crystals, FF1D.6 CLEO: QELS, 2019.

[2] Daniel C. Cole and Scott B. Papp, Kerr-breather-soliton time crystals,
<https://arxiv.org/abs/1811.02523>

[3] Tianhao Xian, Li Zhan, Wenchao Wang, and Wenyan Zhang, Subharmonic Entrainment Breather Solitons in Ultrafast Lasers, Phys. Rev. Lett. 125, 163901 (2020)

[4] D. C. Cole and S. B. Papp, Subharmonic Entrainment of Kerr Breather Solitons, Phys. Rev. Lett. 123, 173904 (2019).

The manuscript “All-Optical Dissipative Discrete Time Crystals” by Taheri et al reports on the experimental demonstration and theoretical study of discrete time crystals (DTC) in Kerr-nonlinear microresonators. Using two self-injection-locked laser pumps, the authors excite soliton states with the different spatial distribution of the soliton pulses in a crystalline microresonator controlled by the pump separation and demonstrate similar behavior in NLSE-based simulations.

In my opinion, the results presenting in the manuscript do not meet the excellence and novelty requirements stated by the Nature Communications. First, the result is based on a well-known phenomenon of soliton state formation in microresonators, which have been studied extensively since 2014. It is well-known and was experimentally demonstrated in a number of photonic platforms that the distribution of soliton pulses on the microresonator circumference can be defined by the underlying nonhomogeneous potential created on the CW background, which can originate from various sources – modal crossings, dispersive wave formation or even birefringence (Wang et al, Optica, 2017). Such potential can indeed trap the solitons and arrange them in spatially symmetrical structures with the different time-translational symmetries. One of the prominent results, which has already been demonstrated in this field (and to which the authors make a surprisingly weak link) are the *defective* and *perfect* soliton crystals, which formation and dynamics have been already carefully experimentally studied in the works of Cole et al (Nature Photonics, 2018) and Karpov et al (Nature Physics, 2019). Second, the dual-pumped self-injection locked scheme experimentally exploited by the authors can be claimed to be new however, the results that the soliton could be arranged by introducing the second pump seem quite obvious and expected to me. I can furthermore also see here some potential drawbacks in the scheme realized with the self-injection locked configuration of both pumps, which I highlighted below in my comments (see point 6). Third, the paper seems to present a very simple experimental result in the area of soliton microcombs, which is just given another perspective. While the new perspective is always good when it brings a better or new understanding of the underlying phenomena, I, unfortunately, can not see it in the present work. Thus, the link to the time crystals presented in the manuscript is more seems like an attempt to hide the rather basic experimental results of generating soliton states in Kerr microresonator behind the modern and active research topic title of time crystals.

There are a few more concrete questions and comments to the authors, listed below, which may be helpful to address in the manuscript:

1. The introduction to the part in my opinion inflates the importance of the time crystals on one hand, but at the same time fails to establish and explain a clear link of the microresonator solitons to the time crystals. The link suggested by the authors seems to be rather superficial and does not aim to bring a new understanding of driven microresonator system.
2. Second, the Concept part of the manuscript, describing the principle of soliton trapping using potential wells created by the second laser and then the following explanation of the symmetry breaking can be significantly reduced just highlighting that the authors suggest to directly treat the spatial symmetry of the soliton states from the temporal perspective. As stated above the idea of soliton trapping is a known one its description can be efficiently shortened.
3. There are serious doubts on the repeatability of the generated soliton states demonstrates by the authors and their deterministic generation. Since the soliton states arise from the chaotic MI in the approach suggested by the authors, there is no guarantee that the spatial distribution of solitons shown in the manuscript can be consistently (and not stochastically) repeated. I suspect that the authors just obtained two demonstrated states by a lucky coincidence, and would highly

appreciate if other states with different TTS would have been demonstrated (e.g. as in He et al, Nature Communications, 2020).

4. In addition to a previous point, I find the experimental results to be very sparse - in my opinion there is a lack of experimental details on how the states have been generated, tuning profiles, repeatability of the generated states. I also see clear patterns of the avoided modal crossings in the experimental optical spectra presented in Fig.4 (c, f), which as far as I understood were not taken into account (and which in fact could be a reason behind the formation of the observed "DTC").
5. At the end of the introduction the DTC presented in the manuscript are called "quantum". No explanation or proof of this statement has been proposed.
6. Another question, which I would like to ask the authors concerns the generation of "Big" DTCs, claimed to be available through the proposed dual-pump method. As far as I can see, the second pump (not the one used for the generation of the soliton states) has to coincide (or stay very close) with the soliton comb lines. Otherwise, the group velocity mismatch of the rotating optical lattice trap and generated cavity solitons will not allow efficient soliton trapping. On one hand, in a self-injection locked scheme the range of available laser-cavity detuning is strongly limited for the second pump. On the other hand, the pump-cavity detuning for the soliton comb lines with respect to their resonances increases quadratically (due to the GVD) as their index number, counted from the pump, grows. This means that in order to create big DTC, the second pump should be placed at multiple FSR offset from the pump (M should be quite big), which however will prohibit its self-injection locking to the respective resonance.

To conclude, I would not recommend this manuscript for the publishing in Nature Communications.

Reviewer #3 (Remarks to the Author):

The manuscript presents the experimental observation of a discrete time crystal (DTC), i.e. an uncommon phase of a driven physical system where the latter develops a periodic dynamics with a period which is multiple integer of the period of the driving mechanism.

The system considered here is a Kerr-nonlinear optical microcavity, a simple photonic apparatus that is well—known, highly controllable and versatile. The novelty of the system is the peculiar driving scheme, which consists in the simultaneous application of two laser pumps with suitably chosen frequencies. The equation of motion describing the dynamics of this system presents a discrete time-translational symmetry, which should results in a periodic motion of the soliton front wave in the microcavity with a period T . The authors demonstrate the possibility to have pulse trains who move with a larger frequency mT (m being an integer number) and hence the appearance of the DTC phase. In my opinion, this result is highly relevant not only because it demonstrates the occurrence of the elusive DTC phase in a simple photonic system, but also because the DTC emerges from a spontaneous symmetry breaking, allowing thus to make a correspondence with the microscopic phenomena at the basis of a large set of phase transitions in condensed matter systems.

The experimental results are soundly supported by a thorough theoretical analysis based on the Lugiato-Lefever equation, which is the common approach used in the study of Kerr-nonlinear microresonators. The good agreement between the experimental measurement and the theoretical prediction for the power spectra [Fig. 4] witnesses the accuracy of the numerical modeling of the physical system.

Furthermore, the article is very clear and well written, and I think it is accessible to a broad readership.

The main issue of the manuscript concerns the validity of some assertions made by the authors, with special reference to the possibility of using this photonic system as a simulator of solid-state crystals. Indeed, the photonic DTC studied in this work emerges from the spontaneous breaking of a discrete symmetry, while the real-space translational symmetry that is broken in solid-state crystals is continuous. As the authors has correctly noticed, “Simultaneous driving of a high-Q Kerr-nonlinear resonator at two different frequencies creates a periodic pattern” [line 130 and following]. I believe that, in the parallelism between soliton wave forms in the photonic DTC and atoms in condensed matter, this periodic pattern should correspond to an external confining potential, which defines precisely the position of the lattice sites. However, in a solid-state crystal, this external potential is typically absent and the periodic arrangement of the atoms emerges only from their mutual interaction. The consequences of this difference are important when considering the defective DTC

crystals [see first paragraph of Section “Discussion”, line 310 and following]. For instance, the authors interpret the missing soliton in the regular pattern of Fig. 5-(a) as a vacancy in a real crystal, but the dynamics of these two systems is different. The pattern with the missing soliton in the photonic DTC should persist over a long time. On the contrary, if one considers a one-dimensional crystal of N atoms in a chain of length L (hence, with an inter-particle separation $a = L/N$) and creates a vacancy removing one atom, the vacancy quickly disappears as the atoms rearrange themselves in a regular pattern with a larger inter-particle distance $a' = L/(N-1)$. For this reason, I think that some assertions about the parallelism between photonic DTC and solid-state crystals (see e.g. “We explore possible extensions to new states such as defective DTCs, the temporal counterpart of solid-state crystals with point defects” at line 98) are misleading and the authors should clarify this issue before the publication of the article.

In addition to that, I have some comments that I would like the authors to consider in the revision of the manuscript. Following, the list of my suggestions.

* The authors mention in several parts the search for big DTCs, claiming that their photonic platform “can readily accommodate m -tupling DTCs with m much larger than 1” [line 212]. I agree with the authors and I think that the possibility of realizing a big DTC in their device is a very relevant aspect (considered also the current interest on the subject), which should be examined in more depth. For instance, it would be interesting to address some questions like: Which is the maximum m obtained so far in the experiment? How does it compare to previous speculations presented in literature (see e.g. the results of Ref. [29])? Which are the current experimental limitations which prevent the large m -tupling of the period? I understand that these questions spark my personal interest on the subject, but I believe that a deeper discussion on big DTC could also be of general interest and increase the impact of this work.

* In fig. 4, the authors compare the experimental and the theoretical power spectrum (panels c-d and f-g). I wonder why the authors do not superimpose the two plots, which may show better the agreement between experiment and theory. Moreover, from the comparison of panels c and f, I have the impression that the high peaks in the experimental spectrum for $M=4$ are better defined than those of $M=3$. Is that accidental and due to the particular measurements chosen to present the results, or rather there is a systematic effect for which the spectra are better defined for large M ? In the latter case, does it have some consequences on the spontaneous breaking of the discrete time translation symmetry for different values of M ? Also, I think it would be interesting to understand better which considerations are made in order to choose M (i.e. the frequency detuning between the two pumps).

* I had some problems in reading fig. 3-(c) and 3-(f): it took me a while to understand that in the comb spectrum two consecutive high peaks are separated by $m-1$ low peaks. I suggest the authors to adapt the plots in these two panels to make them more readable.

* Generally, I find the bibliography exhaustive and complete. Yet, as the authors said, “recently the investigation of dissipative DTCs has gained traction” [line 81] and I would like to mention the following, very recent works on the emergence of time crystals in driven-dissipative systems: Seibold et al., Phys. Rev. A 101, 033839;

Lledo et al., New J. Phys. 22 075002;

Minganti et al., arXiv:2008.08075 [quant-ph].

I would kindly ask the authors to consider them and assess if it is worthy to include them in the bibliography.

In conclusion, I believe that the quality of the manuscript is high, but I am still not convinced that it satisfy the criteria for publication on Nature Communication. I would invite the authors to revise their manuscript to address my specific concerns, before reaching a final decision.

Response to Reviewer # 1

We thank the Reviewer for their careful review and endorsement of the manuscript and for making constructive comments and valuable suggestions. Our responses to these comments and additions to the text addressing the Reviewer's concerns are detailed below. We have also added a *Supplementary Information* section to further elaborate some technical points.

Note: The list of the references introduced by the authors in this response letter appear at the end of the respective response to each reviewer.

Reviewer #1 (Remarks to the Author):

The paper is to report a new concept of time crystal in the micro-cavity. The concept is interesting both for scientific community and public areas. However, I don't think the results and definition of time-crystal seems like sound.

Comment:

1. The concept of time crystal was proposed by the Papp's group first, which was expected to generate the breather soliton lasers [1, 2]. Basically, it should be the fact that the breather period of breather soliton tends to be exactly integral folds of roundtrip time at the vicinity. Then, this phenomenon was demonstrated in ultrafast lasers [3]. However, this concept "time crystal" was be changed in the final publications from both two groups, instead of subharmonic entrainment [3,4].

Response:

We suspect that the Reviewer refers to discrete time crystals (DTCs) in Kerr microcomb systems and mode locked lasers (MLLs) and not time crystals (TCs) in general, because as published literature clearly shows, the notion of TC was proposed in 2012 (Shapere and Wilczek, 2012; Wilczek, 2012) and DTCs in closed systems were experimentally demonstrated in 2017 (Choi *et al.*, 2017; Zhang *et al.*, 2017), long before the conference paper and arXiv manuscript referenced by the Reviewer appeared online. When understood as referring to DTCs in microcombs or MLLs, the Reviewer's statement does not accurately characterize the referenced results. As we have described below (and further elaborated in response to the Reviewer's 3rd comment later on), to demonstrate a DTC, we first have to specify the discrete symmetry and then identify its breaking. A system with a monochromatic or incoherent pump (a Kerr microcomb or an MLL comb) does not allow meeting this critical criterion. Therefore, Refs. [1-4] cited by the Reviewer, important as they are in their own right, do not qualify as DTCs.

The observations reported in Refs. [1-4] of the referee report do not relate the periodicity of the response (here, the period of soliton breathing) to that of the governing Hamiltonian/equations (e.g., pump periodicity), an essential requirement for identifying discrete time translation symmetry (DTTS), its breaking, and consequently the realization of a DTC. Instead, they relate the breathing period to the soliton roundtrip time, and it is well known that the repetition rate (the reciprocal of the roundtrip time) and pump frequency are physically unrelated in these systems and certainly their ratio is not an integer number, as is needed for establishing DTTS breaking and DTC formation.

In particular, regarding the arXiv preprint by Cole and Papp and its published version [Cole and Papp, Phys. Rev. Lett. 123, 173904 (2019)] in which the claim of observing time crystallinity is *eliminated* and from which the words "time crystal" have been *removed altogether*, we believe that the reason why the analogy is inaccurate is the absence of the clear definition of a periodic driving force. Explicitly, Cole and Papp show that, for a certain finite range of the cavity finesse, the breathing period of a microcomb soliton breather can be an integer multiple of the roundtrip time. Yet, the pump laser frequency, which is eliminated from the equations of motion, cannot be related to DTTS or its spontaneous breaking in the system. Additionally, the roundtrip time and the pump period are not commensurate by an integer. Therefore, Refs.

[1,2,4] of the review report fail to demonstrate DTTS and its breaking. Hence their reported phenomenon cannot be referred to a DTC.

We stress that an Ikeda map is merely a way to describe propagation dynamics in a resonator; it is blind to the periodicity of the physical driving force and certainly applying it does not amount to DTTS in the system. To further illustrate our arguments, let us consider the quantum description of a harmonic oscillator. Any state of the oscillator evolves with the harmonic oscillator period T . To follow the evolution of the system, we may define a unitary time evolution operator $U(\Delta t) = \exp(-i H \Delta t)$, where H is the Hamiltonian which evolves the system over the time period $\Delta t = T/n$, with an arbitrary integer n . We can consider $U(\Delta t)$ as a map, i.e., the periodic application of $U(\Delta t)$ allows one to obtain the state of the system at a longer time t . The system always evolves with the period T which is n times longer than the period Δt defined by the map. However, the Δt -periodic application of the time evolution operator does not mean a DTC is formed; the introduced map is only a way of describing the system and is not related to any physical periodic force applied to it.

We should also emphasize that the *possible connection* between TCs and Kerr solitons was discussed in the following work, which was published before Ref. [1] of the referee report:

H. Taheri and A. B. Matsko, “Dually-pumped Kerr microcombs for spectrally pure radio frequency signal generation and time-keeping,” Proceedings Volume 10904, Laser Resonators, Microresonators, and Beam Control XXI; 109040P (2019). <https://doi.org/10.1117/12.2513833>

The above theoretical study targeted timing jitter reduction by soliton trapping in a microcavity, in which frequency harmonics were spontaneously generated from a *single pump* and stabilized by a weak second laser locking to one of the comb harmonics through frequency sweeping. While not focusing on DTCs, the above work correctly recognized that DTC formation requires two pumps and is not possible with a single laser. In the DTCs reported in our current manuscript, frequency harmonics are generated by both of the two pumps and therefore true subharmonic generation is observed, as is required for DTC formation.

To summarize, while we agree that the entrainment phenomena introduced in Refs. [1-4] of this reviewer report are new and intriguing, we believe that, as argued above and as evidence by the due peer-review process leading to their publication, these reports do not qualify as DTCs. We should emphasize that our intention in the current response letter is not to judge the peer review of other published papers. However, to reflect the argument presented above and address the Reviewer’s concerns, we have added Refs. [3,4] to the manuscript (Refs. [27, 28] in the revised text) and have put them in perspective through the following statement (the last paragraph of the Introduction on page 2):

“A DTC is realized when the periodicity of the soliton pulse train becomes an integer multiple of the drive period defined by the beating of the pumps through robust subharmonic generation. The existence of patent discrete TTS and its spontaneous breaking via the emergence of discrete symmetry with a larger integer-multiple period distinguishes DTC formation in our system from subharmonic entrainment [27, 28].”

Comment:

2. Actually, I believe that the reported effect should be the phenomenon of harmonic entrainment, because the harmonic pulse phenomenon has been predicted and demonstrated in the paper. The authors should comment and explain it.

Response:

We thank the Reviewer for their keen observation. Unfortunately, the word “entrainment” is not unambiguously defined in the nonlinear sciences. For instance, in their respectable survey of synchronization phenomena in nonlinear sciences, Pikovsky *et al.* first note that “Indeed, the understanding of such basic terms as *synchronization*, *locking* and *entrainment* differs, reflecting the background, individual viewpoints and taste of a researcher;” see Section 1.3.1 in (Pikovsky, Rosenblum and Kurths, 2003) cited below. They then specify early on in their text that they use entrainment and locking synonymously. In the context of optics,

locking occurs when a frequency already exists in the system (*master frequency*) and another frequency locks to it (*slave frequency*). It can then be said that the slave frequency locks to the master frequency or is entrained by it. Indeed, entrainment is a notion describing a certain relationship between one oscillator coupled to another. In our system, frequency subharmonics do not lock to any other pre-existing frequency and appear originally because of the nonlinearity of the resonator material. Therefore, it *cannot* be said that harmonic entrainment occurs in this system. As a result, while we appreciate the Reviewer's point, we believe that using this terminology will be misleading in the context of the current work.

Comment:

3. If the authors insist on the concept of time crystal, please present the experimental evidence of tending to exact integral folds of roundtrip time or the repetition time of pulses. Also, the authors should explain the mechanism why such pulse train should be the time crystal. I believe that the effect can be proved by dispersive temporal interferometer.

Response:

Proving time crystallinity requires establishing temporal long-range order and robust subharmonic generation, and our measurements clearly substantiate both. Our system elegantly allows confirmation of DTC formation with no need for complex and elaborate real-time pulse measurement techniques. As explained in the Results and Methods sections of the original manuscript, the small phase noise of the RF signal (subharmonic beating) unequivocally evidences a stable microcomb while the spectrum recorded directly on an optical spectrum analyzer verifies the pulse train waveform per cavity roundtrip time and subharmonic generation. Additionally, the drive frequency (the beating of the two pump lasers) can directly be observed on the microcomb spectrum and accurately measured with a photodetector. The combination of these straightforward measurements readily confirms realization of a DTC. Furthermore, the physics of the hyperparametric process resulting in subharmonic generation guarantees that the beating between the pump lasers will be an integer multiple of the subharmonic for a quite comb. Besides experimental data, we have presented cogent confirming numerical modeling data excellently matching the experiments. As a result, direct frequency-domain measurements are perfectly adequate in our experiments and using time-domain or single-shot measurement approaches will mask the elegance and simplicity of our photonic DTC platform.

We should emphasize that the concurrent measurement of the optical spectrum and the low-phase-noise RF signal are well-established and widely-used techniques for establishing stable pulse train creation and finding relevant parameters such as the repetition rate in the microcomb community (Saha *et al.*, 2013; Herr *et al.*, 2014; Kippenberg *et al.*, 2018; Gaeta, Lipson and Kippenberg, 2019). It has been utilized confidently and successfully for nearly a decade now and has particularly been confirmed by pulse measurement techniques such as frequency-resolved optical gating modified for the large repetition rate and small pulse energy of a microcomb soliton (Herr *et al.*, 2014).

While there is indeed no need for utilizing such techniques as dispersive Fourier transform, time-stretch, or dispersive temporal interferometry to prove DTC formation, the techniques suggested by the Reviewer are helpful for subsequent studies focusing particularly on the rigidity of photonic DTCs. We have noted this in the manuscript by adding the following sentence (third paragraph under Discussion, right column at the bottom of page 5 and continuing on to page 6):

“The physics of the hyperparametric process resulting in in stable subharmonic generation in this platform guarantees the division of the pumps’ beatnote to equal subharmonics (integer response to drive periodicity ratio) and hence elegantly allows for confirming DTC behavior through standard frequency-domain measurements without resorting to complex temporal techniques [28, 59]; single-shot pulse measurement methods can facilitate further investigations into the rigidity of photonic DTCs.”

Comment:

4. Creating big time crystals was proposed with ultracold atoms. The spontaneous modification of discrete time translation symmetry in a periodically driven physical system can create a discrete time crystal (DTC). However, the physical mechanism for generating DTC was not clear. Please comment this.

Response:

In a periodically driven quantum system, the so-called Floquet states (eigenstates of the Floquet Hamiltonian) must follow the discrete time translation symmetry of the drive. However, it may happen that a many-body system, due to interactions between particles, prefers evolving with a longer period. Then, Floquet states are still periodic with the driving period but they are macroscopic superposition states (the so-called Schrodinger cat-like states) and quickly decohere to symmetry-broken states and hence DTCs form. It is in full analogy to the spontaneous formation of ordinary space crystals where the symmetry-preserving ground state can be considered as a macroscopic superposition of crystals located at all possible positions in space. Once we measure the position of a single atom, the center of mass of the many-body system becomes localized, the continuous space translation symmetry is broken, and a space crystal emerges (see, Refs. [12-14] in the revised manuscript, also present in the previous version).

In ultra-cold atoms, if interactions between atoms are attractive and sufficiently strong, it becomes energetically favorable to group all atoms in a single localized wave-packet evolving along a classical orbit with a period larger than that of the drive, e.g., twice longer than the period of an external drive; see, e.g., Ref. [8] of the manuscript (same in the previous and update version). Such an evolving Bose-Einstein condensate (BEC) breaks the time translation symmetry – the symmetry preserving state would be a superposition of two such BECs, but it would be a Schrödinger cat state and could never be observed in the experiment (see also Refs. [12-14] in the revised manuscript).

In the optical system considered in our manuscript, a single localized wave-packet (i.e. a single soliton solution) breaks the DTTS of the drive and, like in the ultra-cold atoms case, a DTC forms. In this context, solitons are produced as a result of the nonlinear interaction of pump photons with the resonator material, which leads to the generation of other photons at new harmonics and ultimately to a broadband frequency comb with a huge number of photons at equally-spaced frequencies through cascades four-wave mixing. In this context, solitons are isolated attractors which appear spontaneously and are sustained due to interaction with the continuous wave laser pump. It has been shown that this spontaneous process is accompanied with self-synchronization which ensures the phase-locked oscillation of the optical frequency comb harmonics through their intensity and phase coupling governed by the conservation of energy and momentum under the Kerr Hamiltonian, ultimately creating a coherent mode-locked pulse train (Taheri *et al.*, 2017). While a monochromatically pumped Kerr microcomb does not define a DTTS, two energetic pumps with judiciously chosen power and frequency can create subharmonics which break the DTTS defined by the beatnote. Despite some similarities (e.g., the presence of bosons in both systems), the DTC behavior of this optical system can be much richer and more abundant in terms of the variety of observable effects than DTC based on ultra-cold atoms, because one can observe different scenarios of time translation symmetry breaking and also the formation of defects in the time crystalline structures as described in our manuscript.

To address the Reviewer’s concern and to further highlight this advantage of our photonic platform, we have added the following to the text (the beginning of the second paragraph under Results, *Concept*, on page 2):

“In ultra-cold atomic ensembles, if inter-atomic interactions are attractive and sufficiently strong, it becomes energetically favorable for the atoms to group into a single localized wave-packet evolving along a classical orbit with a period that is an integer multiple of the external drive periodicity [13]. Such a Bose-Einstein condensate (BEC) breaks the discrete TTS, thereby realizing a DTC. In the optical platform utilized here, a localized photon wave-packet (a self-synchronized dissipative optical soliton) arises from the nonlinear interaction of photons in the Kerr resonator [44, 45]. While a monochromatically pumped Kerr microcomb does not define a discrete TTS (see the Supplementary Information), two energetic pumps with

judiciously chosen power and frequency excite subharmonics and can hence break the discrete symmetry defined by their beatnote.”

Comment:

5. The reported phenomenon in the paper looks like harmonic mode-locking, except that there is the periodic deletion of pulses in the roundtrip circulating. The pulse deletion is a common phenomenon in harmonically mode-locked lasers. Therefore, the reported phenomenon can be considered as the harmonic entrainment. I suggest the authors use the concept “harmonic entrainment”.

Response:

In our response to the referee’s Comments 1 and 2 above, we explained in detail why our results demonstrate DTC behavior and should not be conflated with “harmonic entrainment.”

As for harmonic mode locking, it is aimed at increasing the repetition rate of a pulsed laser which will normally generate one pulse per cavity roundtrip time. In harmonic mode locking, the pulse train is entrained to the modulation frequency so that the frequency harmonics between the modulation-induced sidebands die out. Because of lack of discrete time translation symmetry breaking, this phenomenon does not constitute DTC formation. We emphasize that the generation of pulse trains at increased repetition rates (so-called hyperparametric oscillations or Turing patterns with multiple peaks per cavity roundtrip time) is a common phenomenon occurring robustly and repeatably in the modulational instability regime in Kerr microcombs (Coillet *et al.*, 2013; Godey *et al.*, 2014), yet it does not entail time crystalline behavior.

On the other hand, harmonic mode locking suffers from fundamental instabilities, e.g., caused by supermode noise, which destabilize the pulse train and lead to pulse dropouts, as highlighted by the Reviewer. The noise-induced hopping in the oscillating modes is unstable and inherently random. The ensuing symmetry breaking state is therefore *not a stable attractor of the system*. Consequently, the process lacks rigidity and temporal long-range order and does not hence form a DTC.

To address the Reviewer’s concern, we have added the following sentence in the Discussion section (third paragraph under Discussion, on page 6):

“We speculate that DTCs can similarly be observed even in other resonator or mode-locked laser types such as fiber cavities [60, 61] or harmonically mode-locked lasers; in the latter case, certain frequency pinning mechanisms for ensuring robust pulse dropout are necessary.”

Comment:

6. Because the paper used the mechanism of “Soliton Self-injection Locking” to explain the mechanism. So, there must be a coupling mechanism for exceptional points. The effect of exceptional points must be considered. The authors should comment or explain it well.

Response:

In the context of self-injection locking, scattering of the laser light back into the pump occurs as a result of the coupling between clockwise and counter-clockwise cavity modes. This notion is in principle connected to exceptional points (EPs). However, the backscattered light is much weaker than the pump and EP effects such as those utilized recently in optical microresonator experiments, e.g., in (Lai *et al.*, 2019), are not pronounced; we note that the latter reference uses two lasers with close frequencies, one pumping the clockwise mode and the other exciting the counter-clockwise one, so that there are comparably large intensities in both counter-propagating modes. Furthermore, recent studies of the dynamics of self-injection locking in Kerr microresonators have not underscored a notable role played by EPs; see, e.g., (Galiev *et al.*, 2020; Voloshin *et al.*, 2021). Therefore, while we appreciate the Reviewer’s comment and do not rule out the possibility of finding intriguing results if this route is pursued, we strongly believe that studying EPs in the

context of our microresonator-based DTC platform is outside the scope of the current work and should be pursued in subsequent works.

The results and the concept in the paper is much interesting, because the expectation of time crystal is realized in a micro cavity. I recommend the paper should be well revised before publication.

- [1] Scott B. Papp and Daniel C. Cole, Kerr-breather-soliton time crystals, FF1D.6 CLEO: QELS,2019.
- [2] Daniel C. Cole and Scott B. Papp, Kerr-breather-soliton time crystals, <https://arxiv.org/abs/1811.02523>
- [3] Tianhao Xian, Li Zhan, Wenchao Wang, and Wenyan Zhang, Subharmonic Entrainment Breather Solitons in Ultrafast Lasers, *Phys. Rev. Lett.* 125, 163901 (2020)
- [4] D. C. Cole and S. B. Papp, Subharmonic Entrainment of Kerr Breather Solitons, *Phys. Rev. Lett.* 123, 173904 (2019).

References cited:

- Choi, S. *et al.* (2017) ‘Observation of discrete time-crystalline order in a disordered dipolar many-body system’, *Nature*, 543(7644), pp. 221–225. doi: 10.1038/nature21426.
- Coillet, A. *et al.* (2013) ‘Azimuthal Turing Patterns, Bright and Dark Cavity Solitons in Kerr Combs Generated With Whispering-Gallery-Mode Resonators’, *IEEE Photonics Journal*, 5(4), pp. 6100409–6100409. doi: 10.1109/JPHOT.2013.2277882.
- Drummond, P. D. *et al.* (1993) ‘Quantum solitons in optical fibres’, *Nature*, 365(6444), pp. 307–313. doi: 10.1038/365307a0.
- Gaeta, A. L., Lipson, M. and Kippenberg, T. J. (2019) ‘Photonic-chip-based frequency combs’, *Nature Photonics*, 13(3), p. 158. doi: <https://doi.org/10.1038/s41566-019-0358-x>.
- Galiev, R. R. *et al.* (2020) ‘Optimization of Laser Stabilization via Self-Injection Locking to a Whispering-Gallery-Mode Microresonator’, *Physical Review Applied*, 14(1), p. 014036. doi: 10.1103/PhysRevApplied.14.014036.
- Godey, C. *et al.* (2014) ‘Stability analysis of the spatiotemporal Lugiato-Lefever model for Kerr optical frequency combs in the anomalous and normal dispersion regimes’, *Phys. Rev. A*, 89(6), p. 063814.
- Herr, T. *et al.* (2014) ‘Temporal solitons in optical microresonators’, *Nat. Photon.*, 8(2), pp. 145–152.
- Kippenberg, T. J. *et al.* (2018) ‘Dissipative Kerr solitons in optical microresonators’, *Science*, 361(6402), p. eaan8083. doi: <https://doi.org/10.1126/science.aan8083>.
- Lai, Y.-H. *et al.* (2019) ‘Observation of the exceptional-point-enhanced Sagnac effect’, *Nature*, 576(7785), pp. 65–69. doi: 10.1038/s41586-019-1777-z.
- Pikovsky, A., Rosenblum, M. and Kurths, J. (2003) *Synchronization: a universal concept in nonlinear sciences*. Cambridge University Press.
- Saha, K. *et al.* (2013) ‘Modelocking and femtosecond pulse generation in chip-based frequency combs’, *Opt. Express*, 21(1), pp. 1335–1343.

Shapere, A. and Wilczek, F. (2012) ‘Classical Time Crystals’, *Physical Review Letters*, 109(16), p. 160402. doi: 10.1103/PhysRevLett.109.160402.

Taheri, H. *et al.* (2017) ‘Self-synchronization phenomena in the Lugiato-Lefever equation’, *Physical Review A*, 96(1), p. 013828. doi: 10.1103/PhysRevA.96.013828.

Voloshin, A. S. *et al.* (2021) ‘Dynamics of soliton self-injection locking in optical microresonators’, *Nature Communications*, 12(1), p. 235. doi: 10.1038/s41467-020-20196-y.

Wilczek, F. (2012) ‘Quantum Time Crystals’, *Physical Review Letters*, 109(16), p. 160401. doi: 10.1103/PhysRevLett.109.160401.

Zhang, J. *et al.* (2017) ‘Observation of a discrete time crystal’, *Nature*, 543(7644), pp. 217–220. doi: 10.1038/nature21413.

Response to Reviewer # 2

We thank the Reviewer for their comments on the manuscript. Our responses to these comments and additions to the text addressing the Reviewer's concerns are detailed below. We have also added a *Supplementary Information* section to further elaborate some technical points.

Note: The list of the references introduced by the authors in this response letter appear at the end of the respective response to each reviewer.

Reviewer #2 (Remarks to the Author):

The manuscript "All-Optical Dissipative Discrete Time Crystals" by Taheri et al reports on the experimental demonstration and theoretical study of discrete time crystals (DTC) in Kerr-nonlinear microresonators. Using two self-injection-locked laser pumps, the authors excite soliton states with the different spatial distribution of the soliton pulses in a crystalline microresonator controlled by the pump separation and demonstrate similar behavior in NLSE-based simulations.

Comment:

In my opinion, the results presenting in the manuscript do not meet the excellence and novelty requirements stated by the Nature Communications.

Response:

As the Reviewer aptly points out in one of their main comments below, we have introduced a new pumping scheme in a nonlinear optical platform to study a novel physical phenomenon. We have demonstrated that this approach reveals new effects and offers a fresh perspective. Compared to the previous, few and far between, experimental demonstrations of discrete time crystals (DTC), it combines elegance and simplicity to the advantage of further fundamental research and future applications. Indeed, a nonlinear cavity interrogated with few optical and RF sources and analyzers is a convenient platform to explore the physics of TCs. The offered advantages should count in favor of this platform, not against it.

Furthermore, as we have mentioned in the revised manuscript and discussed in the *Supplementary Information*, pondering time crystal (TC) properties in Kerr microcombs is new but not unheard of. An earlier attempt to connect Kerr microcombs to time crystallinity (Cole and Papp, 2018, 2019) was unsuccessful on account of failing to establish the required symmetry of the system and its relation to symmetry breaking in an observable response. This might have partially been because of the intricate defining properties of time crystals (TCs, admittedly an evolving notion, evidenced by the documented literature in this burgeoning research area). Here, we succeed in making this connection correctly by a creative pumping scheme.

In the revised manuscript and added *Supplementary Information*, we have elaborated the key role and prerequisite of clearly establishing system and response symmetry and how they relate to the differences between a dichromatically pumped soliton microcomb and microcombs previously reported in the literature. As the Reviewer knows already, optics has time and time again empowered the realization of intriguing and long-awaited physical phenomena. Through our response below and the revised and expanded submission, we hope to be able to convince the Reviewer that this is yet another example of the strength, utility, and beauty of optical Kerr microcombs which have proven to be an ideal testbed for observing and investigating intriguing notions in physics.

Comment:

First, the result is based on a well-known phenomenon of soliton state formation in microresonators, which have been studied extensively since 2014. It is well-known and was experimentally demonstrated in a number of photonic platforms that the distribution of soliton pulses on the microresonator circumference can be

defined by the underlying nonhomogeneous potential created on the CW background, which can originate from various sources – modal crossings, dispersive wave formation or even birefringence (Wang et al, Optica, 2017). Such potential can indeed trap the solitons and arrange them in spatially symmetrical structures with the different time-translational symmetries. One of the prominent results, which has already been demonstrated in this field (and to which the authors make a surprisingly weak link) are the defective and perfect soliton crystals, which formation and dynamics have been already carefully experimentally studied in the works of Cole et al (Nature Photonics, 2018) and Karpov et al (Nature Physics, 2019).

Response:

We agree that there are apparent parallels between pulse trapping in soliton crystals stabilized by resonator-induced effects such as avoided mode crossings (AMCs) or integrated/residual dispersion sign change. Yet, we strongly disagree with the assessment that subharmonic generation when pumping with two lasers is a known effect or the same as dispersive wave (DW) emission in microcombs. The notion of time crystallinity has been maturing over the past decade, especially over the last few years following the first experimental realization of discrete time crystals in closed spin systems. Therefore, we suspect that the above comment originates from an oversimplification of the defining properties of TCs and the conflation of the origins of crystallinity in DTCs and pulse trapping in microcomb soliton crystals. Briefly, soliton crystals are different from the DTCs because in soliton crystals (i) the pulse train exists with or without the DW ordering the distribution of soliton peaks around the resonator and is independent of it; (ii) the DW is excited by the monochromatic pump and does not define any discrete symmetry in the system, as evidenced by the governing equations (as we have described in detail in the *Supplementary Information*); and (iii) the repetition rate of the pulse train of the soliton crystal is fundamentally independent of the pump frequency. These properties sit in drastic contrast to those of the two-pump microcombs we have demonstrated. From this standpoint, all earlier demonstrations involving soliton trapping and/or soliton generation accompanied with DWs are irrelevant to DTCs. Below we expound why previously demonstrated monochromatically pumped microcombs do not constitute DTCs. This applies also to microcomb soliton crystals which are generated when certain third- and higher-order group velocity dispersion profiles or AMCs (originating from the interaction of different resonator mode families) lead to a high-power harmonic away from the pump in the comb spectrum; see particularly sections I-III of the *Supplementary Information* and also our Response to Comments 1 and 2 below. The introduced references Karpov *et al.* (Nature Physics, 2019) and Wang *et al.* (Optica, 2017) have been included in the revised and expanded submission, as Refs. [65] of the main text and [19, 21] of the *Supplementary Information*.

No-go theorems in the context of TCs rule out *continuous* TCs. Furthermore, in a driven system, a *discrete* TC (DTC) can arise only when a discrete time translation symmetry (DTTS) exists and is broken by a periodic observable whose periodicity is an integer multiple $m > 1$ of the drive periodicity. In monochromatically pumped Kerr microcomb soliton crystals in general, and in those discussed in all of the references highlighted by the Reviewer and those cited by us in the original manuscript in particular, the potential trapping solitons around the resonator results from the properties of the resonator (e.g., geometry and dispersion), which does not lead to any particular DTTS in the Kerr Hamiltonian, nor in the governing equations of the system. This is reflected in the Reviewer’s comment where they note: “... the distribution of soliton pulses on the microresonator circumference can be defined by the underlying nonhomogeneous potential created on the CW background, which can originate from various sources – modal crossings, dispersive wave formation or even birefringence.” Avoided mode crossing, higher-order dispersion, and birefringence are all resonator properties which do not define any discrete symmetry in the equations of motion. As a corollary, speaking of DTTS breaking in such systems is unjustified.

In talking about time translation symmetry breaking (TTSB) in monochromatically-pumped microcombs, the observable to base symmetry breaking assessment on is scarcely identified in explicit terms. One can readily show that irrespective of the specific chosen observable, such systems cannot constitute DTCs:

- If photon count at a fixed position is being observed, then we are dealing with *continuous* time translation symmetry at the input, because the *quasi-classical* (or *coherent*) state of the continuous wave (CW)

pump laser has Poissonian photon statistics. On the other hand, when a soliton is generated in the output, photon count probability will increase near the soliton peak. As a result, continuous TTSB occurs, which clearly does *not* constitute a DTC.

- On the other hand, if the response observable is the electric field, then the pump is periodic but the response, strictly speaking, is only *almost* periodic (because of the carrier-envelope phase). Therefore, the output-over-input periodicity ratio is, in general, not an integer. This shows that a soliton or soliton crystal driven by one CW pump has apparent similarities to a DTC but does not constitute one.

We note that the *photon count probability perspective* described above has proven to be a valuable one. Recent experiments have explained the timing jitter of soliton microcombs from this viewpoint and shown that, remarkably, the soliton can in essence be considered a particle with the characteristic position-momentum quantum uncertainty (Bao *et al.*, 2021). Indeed, signatures of the quantum nature of solitons have previously been studied extensively in the context of fiber solitons; see, e.g., (Drummond *et al.*, 1993). From this perspective, the soliton can be considered a massive particle in its ground mechanical quantum state. The large number of photons in this case and the collective effect of photon interactions in the thermodynamic limit notably match the requirements of DTC behavior in Kerr microcombs such that *with two-pump excitation*, this approach can create DTCs, as we have demonstrated in our work.

It is worth emphasizing that in microcomb soliton crystals, the beating of the pump and DW partially modulates the CW background, and creates a periodic potential lattice around the resonator, similar to what happens in the case of dual-pump microcombs. However, the potential trapping solitons around the resonator results from the properties of the resonator (e.g., shape and material) and *not the drive*, which, as noted above, does not enter any DTTS properties into the equations of motion. Indeed, soliton distribution and ordering in microcomb soliton crystals is defined by such various resonator properties as avoided mode crossings, higher-order dispersion, or even birefringence, all of which originate are *fundamentally independent of the pump*. The following examples help clarify the differences.

- (a) Temperature change will shift the modes of the cavity and hence the frequency of the DW. As a result, a laser pumping another nearby mode at a fixed frequency or one pumping the same temperature shifted mode will in general create a periodic potential with a different periodicity around the cavity, and this periodicity is not tied to the pump frequency. In the two-pump system, in contrast, the excitation frequency and the DTTS it defines are always set by the drive, i.e., the beating of the two pumps, so that even with temperature-shifted modes, the same driving frequency can readily be realized. This driving frequency dictates the DTTS of the system, as evidenced by the two panels in Fig. 2 (main text of the original manuscript).
- (b) In a single-pump microresonator setup with a fixed laser frequency, a microcomb soliton whose span does *not* cover the spectral position of the more energetic DW harmonic can be formed, simply by reducing the pump power. In this case, the CW background will not be modulated *at all*. So, in the same resonator and with the same pump frequency, no periodic potential will form around the resonator, and no soliton trapping and crystallization will occur. In contrast, in the dichromatically pumped system, discrete temporal symmetry is *still present* even if the pumps are weak and excite no subharmonics. This is a fundamental distinction, and an essential one, in defining DTCs. This argument also illustrates that subharmonic generation does *not* occur in the generation of (monochromatically pumped) microcomb soliton crystals.

The roles of the system symmetry and drive in TTSB discussed above clarify why entrainment behavior recently predicted theoretically in some microcombs and observed experimentally in mode-locked lasers does not constitute DTC formation (Cole and Papp, 2019; Xian *et al.*, 2020), even though they were originally thought to be DTCs (Cole and Papp, 2018). It should also be noted that the possibility of observing boundary TCs in a photonic dimer (two coupled Kerr microcavities) was recently proposed theoretically (Seibold, Rota and Savona, 2020); this theoretical work concerns breaking *continuous* time translation symmetry and is fundamentally different from DTCs. Our proposed approach, which relies on a single microresonator rather than two, is advantageous because it *significantly simplifies implementation*, especially

considering the fixed coupling of cavities in an integrated platform. These references have been discussed in the revised manuscript (4th paragraph in the introduction, left column on page 2):

“A DTC is realized when the periodicity of the soliton pulse train becomes an integer multiple of the drive period defined by the beating of the pumps through robust subharmonic generation. The existence of patent discrete TTS and its spontaneous breaking via the emergence of discrete symmetry with a larger integer-multiple period distinguishes DTC formation in our system from subharmonic entrainment [27, 28]. Compared to a recent theoretical proposal for continuous-time SSB and boundary TCs in coupled Kerr cavities [29], our work demonstrates discrete-time SSB, and moreover in a significantly simpler experimental setup constituting one resonator.”

Comment:

Second, the dual-pumped self-injection locked scheme experimentally exploited by the authors can be claimed to be new however, the results that the soliton could be arranged by introducing the second pump seem quite obvious and expected to me. I can furthermore also see here some potential drawbacks in the scheme realized with the self-injection locked configuration of both pumps, which I highlighted below in my comments (see point 6).

Response:

The Reviewer has correctly identified one of the novelties of our approach, specifically from the perspective of microcombs. Indeed, the dichromatic self-injection locking pumping scheme we have introduced and experimentally demonstrated is exactly one of the distinguishing features of our work and necessary for realizing DTCs. Yet, the trapping process can only be “expected” *when both pumps excite and lock coherently to one soliton microcomb*; as previous experiments have shown, achieving these is no easy task and that is why it has not been achieved before this work (Strekalov and Yu, 2009; Okawachi *et al.*, 2016). Therefore, the statement that soliton trapping by two pumps is “obvious” is unjustified because

- (i) the subharmonic generation and the simultaneous locking of the two pumps to the same microcomb are non-trivial key requirements for soliton trapping; and
- (ii) the dynamics and stability conditions of the resonant nonlinear process excited by a polychromatic optical pump are different from those in the case of monochromatic pumping.

Finally, we note that the goal of the current study is *to introduce a new physical phenomenon* which has been missed in prior research in the communities germane to this cross-disciplinary work; this is *not a device paper*. Moreover, as we explain in our Response to Comment 6 below, the point raised as a drawback is not particularly limiting to the self-injection locking of two lasers separately to two resonator modes, and probable dispersion-induced effects can be significantly alleviated through dispersion engineering, especially in integrated resonators.

To elaborate, creating solitons when the system is pumped by two independent lasers is indeed not trivial, especially because, as previous experimental demonstrations have shown, the system will primarily generate *sidebands* defined by the spectral separation of the pumps, hence suppressing any subharmonics *between* them. The dynamics has been studied in simplified few-mode models and even in that case, the dynamics is found to be quite complex (Hansson and Wabnitz, 2014). Soliton formation in the regime we operate the system in has a threshold while sideband formation in dually-pumped systems is a threshold-less process (Strekalov and Yu, 2009).

In fact, the possibility of the simultaneous locking of two pumps to a single microcomb and also separately to their own neighboring resonator modes is the results which was first predicted by us theoretically (Taheri, Matsko and Maleki, 2017; Taheri and A. B. Matsko, 2019a). Only when both pumps coherently lock to the microcomb *without* suppressing the subharmonics sitting between them, do the solitons survive and get trapped in the lattice defined by the pumps. The fact that the Reviewer claims this soliton trapping effect is “expected” to them *subconsciously* assumes that the two pumps are locked coherently to the soliton microcomb formed in the cavity. As we had shown theoretically (Taheri, Matsko and Maleki, 2017), it is

only when this locking happens that the second pump will have a similar effect as that of a DW prompted by higher-order dispersion (e.g., by the 3rd-order dispersion of the cavity) of mode crossing. While we had predicted this effect in the presence of a *weak* second pump, our subsequent theoretical results (Taheri and A. B. Matsko, 2019a) and the current experiments show that with self-injection locking, this feat is possible even with a second pump with power comparable to that of the first one.

Comment:

Third, the paper seems to present a very simple experimental result in the area of soliton microcombs, which is just given another perspective. While the new perspective is always good when it brings a better or new understanding of the underlying phenomena, I, unfortunately, can not see it in the present work. Thus, the link to the time crystals presented in the manuscript is more seems like an attempt to hide the rather basic experimental results of generating soliton states in Kerr microresonator behind the modern and active research topic title of time crystals.

Response:

Our results contribute a new perspective. Far from obvious, they in fact bridge an important and undeniable link between nonlinear photonics and true DTTS breaking in a driven non-equilibrium optical system thereby achieving photonic DTCs for the first time. It is this new perspective which we claim connects a prolific and technically developed area of photonics to a flourishing area of condensed matter physics, setting the stage for many more significant observations and demonstrations to come. In doing so, our work also introduces an important standpoint which empowers realizing DTCs in other Kerr-nonlinear systems (be it photonic, phononic, superconducting, or otherwise) and shows why previously demonstrated Kerr soliton microcombs are not DTCs. Especially with the additions made to the manuscript, the new perspective offered to both TC and microcomb research areas are even clearer. Therefore, our work provides yet another historic example of the utility and versatility of optics in demonstrating novel physical phenomena. As explained below, the statement that our experiments are basic while also requesting further experimental details in subsequent Comments 3 and 4 is puzzling because:

- (i) our experiments involving two independent pumps self-injection locked to the same cavity is reported *for the first time* in the paper. It was not shown before that it is possible to achieve locking of two lasers to the same cavity using this approach. Achieving this is definitely not obvious because the gain bandwidth of each of the lasers is much larger than their separation frequency (multiple FSRs) such that the emission of one laser penetrates the gain bandwidth of the other one. This phenomenon can result in locking the lasers together, thus preventing the generation of DTCs. Consequently, the novelty of the experiments by itself warrants publication in a widely read and recognized journal such as *Nature Communications*. The Reviewer partially identified this novelty in the previous Comment.
- (ii) our observation of DTCs was guided by a solid theoretical study, including our theoretical results over the past few years. As shown in our earlier papers, DTC behavior can be masked by DW-like pulse trapping (when the second pump is very weak compared to the first one) (Taheri, Matsko and Maleki, 2017) as well as chaotic phenomena (when one or both of the pumps are too strong). Only careful selection of the experimental parameters enabled the observation of subharmonic generation in the current work.

There are a few more concrete questions and comments to the authors, listed below, which may be helpful to address in the manuscript:

Comment:

1. The introduction to the part in my opinion inflates the importance of the time crystals on one hand, but at the same time fails to establish and explain a clear link of the microresonator solitons to the time crystals. The link suggested by the authors seems to be rather superficial and does not aim to bring a new understanding of driven microresonator system.

Response:

In this work, resonator and microcombs are primarily tools to demonstrate, for the first time, the notion of dissipative DTCs in an optical platform. To that end, we have focused on the physics of the phenomenon, with an elaborate Concept subsection (as admitted by the Reviewer in their Comment 2 below), and have also reviewed and cited relevant effects and publications. Indeed, in the Results section of our submitted manuscript, under “Concept” and specifically in the second and third paragraphs of this subsection, *we had clearly explained what observable we are looking at, why and how it defines DTTS, and how soliton formation break this symmetry*. DTTS in our platform is also described very clearly and demonstrated mathematically in the first part of the Methods sections. Consequently, this Comment suggests that the Reviewer might have overlooked these parts of the manuscript. Furthermore, evidenced by the literature, TCs have proven a remarkably intriguing area of condensed matter physics, introducing truly novel phenomena such as avoiding thermalization in a driven system with vanishingly small coupling to its environment, or condensed matter phases in the time domain. We have not exaggerated the importance of TCs, but have simply reviewed the relevant literature.

That said, to further address the Reviewer’s concern regarding clarifying the connection of TCs and microresonator solitons, we have added to the manuscript text a summary of our Response above to their Comment “First, ...,” and a detailed discussion in an added *Supplementary Information* explaining why microcombs observed before this work were not DTCs; see in particular sections I-III of the *Supplementary Information*:

- I. Absence of DTTS in Monochromatically-pumped Microcombs
→ In this section, we have examined the symmetries of microcombs pumped by one CW laser and shown that they can realize the spontaneous breaking of *continuous* time translation symmetry but not DTTS.
- II. Absence of DTTS in Microcombs with Dispersive Waves Resulting from Higher-order Dispersion
→ In this section, we have shown that integrated dispersion sign change and DW emission do not define DTTS in microcomb systems. Therefore, subharmonic generation and DTC formation cannot be observed in DW-emitting microcombs.
- III. Absence of DTTS in Microcomb Soliton Crystals Stabilized by Avoided Mode Crossings
→ In this section, we have shown that AMCs do not define DTTS in microcomb systems and hence microcomb soliton crystals are not DTCs.

Below we again reiterate the general elements needed to call a system a DTC and relate them to the case of the Kerr system studied in our manuscript.

- External periodic drive is needed in order to define DTTS of a system. In Kerr systems, microresonators are pumped by CW laser radiation. The frequency of laser radiation can be eliminated from the equations that govern system behavior and consequently the radiation frequency cannot define DTTS in the system. It is a typical experimental situation where one measures photon counts but not the temporal evolution of the electric field. If, however, the electric field evolution is detected, then one can associate the laser radiation period with DTTS of a system and investigate if time evolution of the electric field in a resonator breaks such DTTS into another DTTS corresponding to an integer multiple of the laser radiation period. Although it may, in principle, be possible to fulfil these conditions, here we focus on the photon count experiment and on DTTS in the frequency range of the order of the roundtrip frequency of the resonator. Such DTTS cannot emerge if a single laser beam is applied because then the photon counts are constant in time. However, two laser pumps can create a rotating lattice trap where optical modes can be localized. It results in a drive that changes periodically with period T_R/M (where M is an integer) and defines the DTTS of the system. Steady states that break such a symmetry correspond to DTCs and the aim of our manuscript is to describe and demonstrate them in the experiments.
- Ordinary space crystals form due to the interactions between particles. A similar situation is present in DTCs where for sufficiently strong inter-particle interactions, a quantum phase transition is observed in which symmetry broken states that can live forever in the limit of a large number of particles emerge. In

Kerr systems, average number of photons is extremely large and strong effective interactions between the photons in the cavity are based on the Kerr nonlinearity.

- A DTC does not require fine tuning. That is, there is a finite range of system parameters where a DTC can be observed. In the Kerr system investigated in our manuscript, this robust operation regime is guaranteed by self-injection-locked pumping. As seen in Fig. 2 of the manuscript, there is wide range of detuning values over which subharmonic generation occurs robustly and reliably. With TEC, the generated microcombs are maintained simply over hours.
- A generic periodically driven many-body system heats up to infinite temperature. There are basically two mechanisms which can prevent DTCs from heating: (i) A periodically driven many-body system is integrable (e.g. there exist local integrals of motion like in many-body localized systems); (ii) There is damping which drains energy from the system to evade thermalization. In the latter case we call time crystals dissipative DTCs. In the Kerr system considered in our manuscript we deal with such dissipative DTCs.

Comment:

2. Second, the Concept part of the manuscript, describing the principle of soliton trapping using potential wells created by the second laser and then the following explanation of the symmetry breaking can be significantly reduced just highlighting that the authors suggest to directly treat the spatial symmetry of the soliton states from the temporal perspective. As stated above the idea of soliton trapping is a known one its description can be efficiently shortened.

Response:

As we have explained extensively in the previous Responses, we do not just focus on the trapping of solitons, but on the trapping of solitons *which results from the spontaneous breaking of DTTS in the system*. We have shown how to realize periodic driving which allows us to define DTTS and demonstrate that there are steady states which break this symmetry. As a result, mere reliance on referencing DW-assisted soliton trapping will be both incorrect and misleading in the context of DTCs. Moreover, given the interdisciplinary nature of the present work (condensed matter physics, photonics, nonlinear optics) and considering that the scope and audience of *Nature Communications* cover a broader group than researchers in photonics, we strongly believe that the current description is absolutely necessary.

Restating our description in the text, in spatial crystals one asks if particles are periodically distributed in space at a fixed moment of time (which corresponds to the moment of the detection). In time crystals the role of space and time are exchanged. That is, one fixes the position in space (which corresponds to the location of a detector) and ask if a detector can click periodically in time. There are systems which can reveal both crystalline structures in space and in time. For example, periodically driven ultra-cold atoms moving in a toroidal trap can reveal both time and space crystalline structures because there are periodic boundary conditions both in space and in time (see Ref. [13] in the revised manuscript). A similar situation is observed in the system considered in our manuscript, i.e., both crystalline structures in space and in time can be observed.

We stated above that the distinction noted above between DW-induced soliton trapping in monochromatically pumped Kerr microresonators and DTTS breaking in the current dichromatically pumped platform is expounded in the *Supplementary Information* section added to the manuscript in this review process.

Comment:

3. There are serious doubts on the repeatability of the generated soliton states demonstrates by the authors and their deterministic generation. Since the soliton states arise from the chaotic MI in the approach suggested by the authors, there is no guarantee that the spatial distribution of solitons shown in the manuscript can be consistently (and not stochastically) repeated. I suspect that the authors just obtained two

demonstrated states by a lucky coincidence, and would highly appreciate if other states with different TTS would have been demonstrated (e.g. as in He et al, Nature Communications, 2020).

Response:

The Reviewer objects that our system operates in the modulational instability (MI) regime, but that is not the case. We suspect that they may have mistaken the current experimental results with our earlier theoretical work (Taheri and A. Matsko, 2019).

We emphasize that *the assumption that the soliton states demonstrated in our platform “arise from the chaotic MI” is completely wrong*. Nowhere in the manuscript have we indicated, or even implied, that the solitons are generated in the MI regime. On the contrary, the manuscript shows clearly, both theoretically and experimentally, that the solitons do not appear as a result of any laser sweeping from the MI regime (supporting Turing rolls) to that allowing soliton formation. In fact, it is now common knowledge that, unlike the initially more common method of soliton generation through laser sweep and passage through the chaotic microcombs phase, the self-injection locking technique does not need pumping in the MI region and does not involve laser tuning. Also, in our previously submitted manuscript we had already noted explicitly that the matching numerical modeling data utilizes *hard excitation*.

To further address the Reviewer’s doubts, we have shown in the *Supplementary Information* section that by pumping three different pairs of resonator modes with different frequency separations, we have consistently observed robust subharmonic generation; see also our Response to Comment 4 below. Since the subharmonics always fill out all of the cavity modes sitting between the two pumps, the corresponding pulse train always corresponds to a full train with one removed pulse (one missing peak in the lattice). Further varieties can be realized by modifications of the pumping scheme, but are outside the scope of this first demonstration of dissipative DTCs.

Comment:

4. In addition to a previous point, I find the experimental results to be very sparse - in my opinion there is a lack of experimental details on how the states have been generated, tuning profiles, repeatability of the generated states. I also see clear patterns of the avoided modal crossings in the experimental optical spectra presented in Fig.4 (c, f), which as far as I understood were not taken into account (and which in fact could be a reason behind the formation of the observed “DTC”).

Response:

The Methods section includes details on the experiments. To address the Reviewer’s Comment, we have added more on this procedure to the Methods section of the manuscript (the last paragraph under *Soliton Self-injection Locking*):

“Through tuning the locking point by regulating the optical phase delay between the laser chip and the resonator, the frequency of one of the lasers was shifted to a spectral position where a microcomb was observed. DTC generation was then straightforward by selecting proper pump powers. To this end, we started from higher power levels, too high to sustain a microcomb. Laser currents were then reduced till stable subharmonic generation was observed. At this stage, the power of one pump was slightly above that required for initiating the hyperparametric process with a single laser while that of the other was below this threshold, and importantly neither pump could generate a soliton microcomb alone.”

The Reviewer suggests that avoided mode crossings (AMCs) could be the cause of observing crystallinity in our experiments. But, as we detail here, this suggestion cannot be correct. As is well known in the Kerr microcomb community (Xue, Qi and Weiner, 2016), AMCs whether in the normal or the anomalous dispersion regime result from the interaction of two spatial mode families and lead to high-power harmonics in the microcomb spectrum *at the frequency where the mode families cross*. As a result, the AMC signature in the comb spectrum should always pin to the same frequency. In our experiments, this frequency pinning effect is absent and the high-power harmonics appear at the position of the two pumps and the harmonics

that their nonlinear interaction excites. This is evidenced by the fact that these more energetic comb harmonics are always separated by the beating frequency of the two pumps, and additionally always follow them, as is clearly seen in Figs. 4 (c, f) and Fig. S2 (2-FSR between pumps) in the *Supplementary Information*. In these examples, the frequency of the first pump is nearly unchanged. It is only the frequency of the second pump (and hence its beating with the first pump) which is changed. Nonetheless, a notably fixed high-power harmonic independent of the beat-note of the pumps is not observed in the microcomb spectra.

We should add that it is an accepted fact among all experimentalists in the Kerr microcomb community that AMCs will occur in microresonators and their signatures will appear in the spectra of microcombs. The main question to be asked is whether the dynamics and stability of the experimentally achieved microcomb will be impacted by the AMC or not. In our case, the AMCs are weak enough to ensure that their signature, as explained above, is vanishingly small (they do not produce a DW in the microcomb spectra) and that their impact on microcomb stability is negligible. To clearly communicate this point to the readers, apart from further experimental details noted above, we have added the following to the manuscript (Methods section, the end of the paragraph under *Soliton Self-injection Locking* on page 9):

“It should be emphasized that the absence of anti-crossing-induced frequency pinning in the microcomb spectra confirms the negligible effect of avoided mode crossings in our experiments; see *Supplementary Information*, section IV.”

In section IV of the *Supplementary Information* we have included additional experimental data and also added the following:

“To elaborate, an AMC pins a high-power harmonic in the microcomb spectrum at the position of the mode crossing [25]. While AMCs are practically inevitable in experiments, whether or not they impact the excitation and stability of a microcomb generated in a certain region can partially be judged based on the existence of a strong comb tooth pinned in the spectrum and fixed as the pump frequency is shifted. This pinning signature is absent in our experiments, neither when the two pumps generate harmonics through four-wave mixing (FWM), Fig. S1, nor when they excite subharmonics and the beatnote of the pumps is changed in successive FSRs in the same pumping region, Fig. S2. This is particularly manifest in the spectrum for 2-FSR separation of the pumps, Fig. S2(e), which shows no sign of an AMC. Instead, high-power harmonics always follow the external pumps and the FWM harmonics that appear at frequency intervals dictated by their beatnote. Therefore, subharmonic generation in our experiments is not impacted by AMC and mode crossings play no role in stabilizing the observed microcombs. This fact is supported also by our numerical modeling data which demonstrates stable pulse train propagation over hundreds of cavity photon lifetimes (thousands of drive periods); see Figs. 3(a, d) in the main text. We have not incorporated any AMC or GVD parameter beyond D_2 in our numerical simulations.”

Comment:

5. At the end of the introduction the DTC presented in the manuscript are called “quantum”. No explanation or proof of this statement has been proposed.

Response:

Spontaneous breaking of space or time translation symmetry is a quantum phenomenon. If quantum Hamiltonian possesses certain symmetry, its eigenstates must also be the eigenstates of the corresponding symmetry operator. However, in the case of time (or space) crystals, symmetry-preserving eigenstates are many-body Schrödinger cat like states which quickly decohere into symmetry broken states and we can observe crystalline structures in time (or in space); see Ref. [13] of the revised manuscript. In classical systems, consequences of Hamiltonian symmetries are not so dramatic. For example, if we locate a classical particle in a single well of a spatially periodic potential, then it will stay there forever if its energy is smaller than the potential barrier between the wells. Quantum mechanically, the situation is different because a particle will tunnel into neighboring potential wells. Thus, while in the classical description the steady state of a particle breaking space (or time) translation symmetry is no surprise, in the quantum case realization of a

symmetry broken steady state is nontrivial and requires an interacting many-body system. Classical DTCs are investigated in the literature but they refer to different phenomena compared to quantum DTCs. That is, classical DTCs are periodically driven many-body classical systems which break ergodicity and reveal synchronized motion and breaking of time translation symmetry is not an issue because, as we have explained, it is possible even in a single particle case. As an aside, the quantum timing jitter of the generated soliton train in a dichromatically-pump microcomb is determined by the quantum frequency noise of the pumps. In a standard Kerr comb (pumped by a single CW laser), the timing jitter is not correlated with pump noise (Bao *et al.*, 2021).

Comment:

6. Another question, which I would like to ask the authors concerns the generation of “Big” DTCs, claimed to be available through the proposed dual-pump method. As far as I can see, the second pump (not the one used for the generation of the soliton states) has to coincide (or stay very close) with the soliton comb lines. Otherwise, the group velocity mismatch of the rotating optical lattice trap and generated cavity solitons will not allow efficient soliton trapping. On one hand, in a self-injection locked scheme the range of available laser-cavity detuning is strongly limited for the second pump. On the other hand, the pump-cavity detuning for the soliton comb lines with respect to their resonances increases quadratically (due to the GVD) as their index number, counted from the pump, grows. This means that in order to create big DTC, the second pump should be placed at multiple FSR offset from the pump (M should be quite big), which however will prohibit its self-injection locking to the respective resonance.

Response:

This is a good question. As already noted, we have demonstrated self-injection locking of two lasers to the same resonator. We have found that the lasers can be self-injection locked independently at different frequency separations of the modes. Self-injection locking does not lock the lasers together, but *locks each separately to their respective neighboring resonator mode*. The alleged limitation in terms of group velocity dispersion is therefore surmountable and not a major obstacle. Additionally, dispersion engineering of microresonators can reduce the probable concern raised by the Reviewer, especially in integrated resonators we have proposed in the Discussion section for future demonstrations. It has been shown, both theoretically and experimentally, that suppressing the second-order and even third-order dispersion of microresonators is possible in material platforms currently in widespread use in the Kerr microcomb community (Li *et al.*, 2017; Ghasemkhani *et al.*, 2019; Taheri and A. B. Matsko, 2019b; Anderson *et al.*, 2020). From the perspective of the physics of the problem, which is the main focus of the current work, we suspect this effect can be avoided easily in optical platforms such dispersion managed fiber cavities. Either way, further investigation of the fundamental constraints of the simultaneous self-injection locking of two pumps to a cavity is *beyond* the scope of the current study and part of our future investigations; this work is *not a device paper*, but one targeting the optical demonstration of a physical concept.

Comment:

To conclude, I would not recommend this manuscript for the publishing in Nature Communications.

Response:

Above, we have explained thoroughly why the Reviewer’s evaluation of the contributions of our submitted manuscript is not warranted. That said, we should emphasize that the experimental demonstration of novel physical phenomena in known photonic platforms has time and time again proved valuable, from both a fundamental science and an applied perspective. Many papers on such demonstrations have been published in *Nature Communications*. Two very recent examples are

A. Dutt, M. Minkov, Q. Lin, L. Yuan, D. A. B. Miller, and S. Fan, Experimental Band Structure Spectroscopy along a Synthetic Dimension, *Nature Communications* 10, 1 (2019)

and

A. Bergman, R. Duggan, K. Sharma, M. Tur, A. Zadok, and A. Alù, Observation of Anti-Parity-Time-Symmetry, Phase Transitions and Exceptional Points in an Optical Fibre, *Nature Communications* 12, 1 (2021).

Both papers use optical fibers and *off-the-shelf equipment* for the demonstration of novel physical effects which have recently been predicted. Many such papers (see, for instance, the second example above), claim the simplicity of their setup and experiments as a point of strength in their demonstration. In this light, we strongly disagree with the Reviewer and believe that our demonstration of DTCs in a photonic system opens the door to a plethora of new experiments with great potential for novel discoveries. In addition to that, the inter-disciplinary character of this work paired with the novelty of phenomenon (dissipative DTCs) and approach (self-injection locking of two independent pumps) all support publication in *Nature Communications*.

References cited:

Anderson, M. H. *et al.* (2020) ‘Zero-dispersion Kerr solitons in optical microresonators’, *arXiv:2007.14507 [physics]*. Available at: <http://arxiv.org/abs/2007.14507> (Accessed: 27 January 2021).

Bao, C. *et al.* (2021) ‘Quantum diffusion of microcavity solitons’, *Nature Physics*, pp. 1–5. doi: 10.1038/s41567-020-01152-5.

Cole, D. C. and Papp, S. B. (2018) ‘Kerr breather-soliton time crystals’, *arXiv:1811.02523 [physics]*. Available at: <http://arxiv.org/abs/1811.02523> (Accessed: 20 July 2019).

Cole, D. C. and Papp, S. B. (2019) ‘Subharmonic Entrainment of Kerr Breather Solitons’, *Physical Review Letters*, 123(17), p. 173904. doi: 10.1103/PhysRevLett.123.173904.

Drummond, P. D. *et al.* (1993) ‘Quantum solitons in optical fibres’, *Nature*, 365(6444), pp. 307–313. doi: 10.1038/365307a0.

Ghasemkhani, M. *et al.* (2019) ‘Dissipative Quartic Kerr Solitons for WDM Applications’, in *Frontiers in Optics + Laser Science APS/DLS (2019), paper JTU4A.84. Laser Science*, Optical Society of America, p. JTU4A.84. doi: 10.1364/FIO.2019.JTU4A.84.

Hansson, T. and Wabnitz, S. (2014) ‘Bichromatically pumped microresonator frequency combs’, *Physical Review A*, 90(1), p. 013811. doi: 10.1103/PhysRevA.90.013811.

Li, Q. *et al.* (2017) ‘Stably accessing octave-spanning microresonator frequency combs in the soliton regime’, *Optica*, 4(2), pp. 193–203.

Okawachi, Y. *et al.* (2016) ‘Quantum random number generator using a microresonator-based Kerr oscillator’, *Optics Letters*, 41(18), pp. 4194–4197. doi: 10.1364/OL.41.004194.

Seibold, K., Rota, R. and Savona, V. (2020) ‘A dissipative time crystal in an asymmetric non-linear photonic dimer’, *Physical Review A*, 101(3), p. 033839. doi: 10.1103/PhysRevA.101.033839.

Strekalov, D. and Yu, N. (2009) ‘Generation of optical combs in a whispering gallery mode resonator from a bichromatic pump’, *Phys. Rev. A*, 79(4), p. 041805.

Taheri, H. and Matsko, A. (2019) ‘Crystallizing Kerr Cavity Pulse Peaks in a Timing Lattice’, in *Frontiers in Optics + Laser Science APS/DLS (2019)*, paper JTU3A.90. *Laser Science*, Optical Society of America, p. JTU3A.90. doi: 10.1364/FIO.2019.JTu3A.90.

Taheri, H. and Matsko, A. B. (2019a) ‘Dually-pumped Kerr microcombs for spectrally pure radio frequency signal generation and time-keeping’, in *Laser Resonators, Microresonators, and Beam Control XXI. Laser Resonators, Microresonators, and Beam Control XXI*, International Society for Optics and Photonics, p. 109040P. doi: 10.1117/12.2513833.

Taheri, H. and Matsko, A. B. (2019b) ‘Quartic dissipative solitons in optical Kerr cavities’, *Optics Letters*, 44(12), pp. 3086–3089. doi: 10.1364/OL.44.003086.

Taheri, H., Matsko, A. and Maleki, L. (2017) ‘Optical lattice trap for Kerr solitons’, *The European Physical Journal D*, 71(6), p. 153.

Xian, T. *et al.* (2020) ‘Subharmonic Entrainment Breather Solitons in Ultrafast Lasers’, *Physical Review Letters*, 125(16), p. 163901. doi: 10.1103/PhysRevLett.125.163901.

Xue, X., Qi, M. and Weiner, A. M. (2016) ‘Normal-dispersion microresonator Kerr frequency combs’, *Nanophotonics*, 5(2), pp. 244–262. doi: 10.1515/nanoph-2016-0016.

Response to Reviewer # 3

We thank the Reviewer for their careful review and endorsement of the manuscript and for making constructive comments and valuable suggestions. Our responses to these comments and modifications of the text addressing the Reviewer's concerns are detailed below. We have also added a *Supplementary Information* section to further elaborate some technical points.

Note: The list of the references introduced by the authors in this response letter appear at the end of the respective response to each reviewer.

Reviewer #3 (Remarks to the Author):

The manuscript presents the experimental observation of a discrete time crystal (DTC), i.e. an uncommon phase of a driven physical system where the latter develops a periodic dynamics with a period which is multiple integer of the period of the driving mechanism.

The system considered here is a Kerr-nonlinear optical microcavity, a simple photonic apparatus that is well-known, highly controllable and versatile. The novelty of the system is the peculiar driving scheme, which consists in the simultaneous application of two laser pumps with suitably chosen frequencies. The equation of motion describing the dynamics of this system presents a discrete time-translational symmetry, which should result in a periodic motion of the soliton front wave in the microcavity with a period T . The authors demonstrate the possibility to have pulse trains who move with a larger frequency mT (m being an integer number) and hence the appearance of the DTC phase. In my opinion, this result is highly relevant not only because it demonstrates the occurrence of the elusive DTC phase in a simple photonic system, but also because the DTC emerges from a spontaneous symmetry breaking, allowing thus to make a correspondence with the microscopic phenomena at the basis of a large set of phase transitions in condensed matter systems.

The experimental results are soundly supported by a thorough theoretical analysis based on the Lugiato-Lefever equation, which is the common approach used in the study of Kerr-nonlinear microresonators. The good agreement between the experimental measurement and the theoretical prediction for the power spectra [Fig. 4] witnesses the accuracy of the numerical modeling of the physical system.

Furthermore, the article is very clear and well written, and I think it is accessible to a broad readership.

Comment:

The main issue of the manuscript concerns the validity of some assertions made by the authors, with special reference to the possibility of using this photonic system as a simulator of solid-state crystals. Indeed, the photonic DTC studied in this work emerges from the spontaneous breaking of a discrete symmetry, while the real-space translational symmetry that is broken in solid-state crystals is continuous. As the authors has correctly noticed, "Simultaneous driving of a high-Q Kerr-nonlinear resonator at two different frequencies creates a periodic pattern" [line 130 and following]. I believe that, in the parallelism between soliton wave forms in the photonic DTC and atoms in condensed matter, this periodic pattern should correspond to an external confining potential, which defines precisely the position of the lattice sites. However, in a solid-state crystal, this external potential is typically absent and the periodic arrangement of the atoms emerges only from their mutual interaction. The consequences of this difference are important when considering the defective DTC crystals [see first paragraph of Section "Discussion", line 310 and following]. For instance, the authors interpret the missing soliton in the regular pattern of Fig. 5-(a) as a vacancy in a real crystal, but the dynamics of these two system is different. The pattern with the missing soliton in the photonic DTC should persist over a long time. On the contrary, if one considers a one-dimensional crystal of N atoms in a chain of length L (hence, with an inter-particle separation $a = L/N$) and creates a vacancy removing one atom, the vacancy quickly disappear as the atoms rearrange themselves in a regular pattern with a larger

inter-particle distance $a' = L/(N-1)$. For this reasons, I think that some assertions about the parallelism between photonic DTC and solid-state crystals (see e.g. “We explore possible extensions to new states such as defective DCTs, the temporal counterpart of solid-state crystals with point defects” at line 98) are misleading and the authors should clarify this issue before the publication of the article.

Response:

The Reviewer is perfectly right that ordinary space crystals form due to spontaneous breaking of the continuous translation symmetry while in DTCs discrete translation symmetry is spontaneously violated. Wilczek’s initial idea concerned spontaneous breaking of the continuous time translation symmetry but it turned out that it was not possible to observe it in the most interesting case of the ground state of a many-body system [5] (reference from the main text, both the previous and current version). Later DTCs were proposed where formation of crystalline structures is different than in the condensed matter case. In DTCs one observes that due to interactions between particles, a many-body system spontaneously self-reorganizes its motion and starts moving with a period different from the period dictated by the drive. A new periodic motion is established, i.e., a novel crystalline structure in time emerges. The Reviewer may already know that the differences between the formation of DTCs and ordinary spatial crystals are discussed in numerous papers; see, e.g., review articles and a recent book, Refs. [12-14] of the revise submitted manuscript (also present in the previous version).

Despite the differences, DTCs are a playground where one can investigate further analogies between space crystals and periodic structures in the time dimension. The analogies can go beyond spontaneous symmetry breaking phenomena and, e.g., Anderson and many-body localization, Mott insulator phase and topological phases can all be observed in the time domain; see Refs. [12,13] in the revise manuscript. In our manuscript, we discuss defects in DTCs and we believe that while the analogy to condensed matter counterpart is *not exact*, it is clear what we mean by vacancies, dislocations or interstitials. To address the Reviewer’s concern, we warn the reader in the revised version of the manuscript (at the end of the first paragraph in the Discussion section) that due to the fact that in DTCs not continuous, but *discrete* translation symmetry is spontaneously broken, the analogy to solid state defects is not exact:

“Finally, we should emphasize that the analogy drawn here between defects in DTCs and those in solid state spatial crystals is, strictly speaking, not exact because in the latter the spontaneous breaking of continuous, rather than discrete, TTS occurs.”

In addition to that, I have some comments that I would like the authors to consider in the revision of the manuscript. Following, the list of my suggestions.

Comment:

* The authors mention in several parts the search for big DTCs, claiming that their photonic platform “can readily accommodate m -tupling DTCs with m much larger than 1” [line 212]. I agree with the authors and I think that the possibility of realizing a big DTC in their device is a very relevant aspect (considered also the current interest on the subject), which should be examined in more depth. For instance, it would be interesting to address some questions like: Which is the maximum m obtained so far in the experiment? How does it compare to previous speculations presented in literature (see e.g. the results of Ref. [29])? Which are the current experimental limitations which prevent the large m -tupling of the period? I understand that these questions spark my personal interest on the subject, but I believe that a deeper discussion on big DTC could also be of general interest and increase the impact of this work.

Response:

In theoretical models we may consider DTCs as big as we wish. However, switching to experiments often readily encounters practical, often non-fundamental, limitations. So far period-doubling and period-tripling DTCs have been realized experimentally, see Refs. [46, 47] in the revised manuscript. In Ref. [29], mentioned by the referee (Ref. [32] in the revised version), the theoretical results correspond to m -tupling DTCs with m of the order of 10, but there is no analysis concerning the experimental realization. In Refs. [30, 33]

of the revised manuscript there is an extensive analysis of experimental conditions for ultra-cold atoms DTCs and it turns out that $m = 20-100$ is attainable with the optimal value being around $m = 30$. In the Kerr system considered in our manuscript, our theoretical analysis shows that observing m in the range of 10 (F_1, F_2, α_2) = (1.5, 0.98, 3.5) and 20 (F_1, F_2, α_2) = (1.6, 0.99, 3.1) is readily possible for the resonator with quality factor and dispersion profile utilized in our experiments; see below where the left figure corresponds to $m = 10$ and the right figure corresponds to $m = 20$. Here $F_{1,2}$ and $\alpha_{1,2}$ are normalized pump intensity and detuning values and in both cases $\alpha_1 = \alpha_2 - d_{\text{int}}/2$, d_{int} being the residual or integrated dispersion; normalization and notation follows our earlier theoretical work in Ref. [48] of the revised manuscript. Please note that in figures below, the bottom panels plot microcomb spectra as a function of mode number not wavelength, hence the apparent flipping of the x -axis compared to the figures plotted in the manuscript.

As can be seen from the second figure below, for larger beatnote of the pumps, pulse width can be too large for the lattice sites and cover more than one spots in the array. Although even in this case symmetry breaking can in principle occur, we note that pulse duration is related to the resonator group velocity dispersion (GVD) and pump detuning and can be reduced by decreasing GVD in the pumping region. It has been shown, both theoretically and experimentally, that suppressing the second-order and even third-order dispersion of microresonators is possible in material platforms such as silicon nitride which are currently in widespread use in the Kerr microcomb community (Li *et al.*, 2017; Ghasemkhani *et al.*, 2019; Taheri and Matsko, 2019; Anderson *et al.*, 2020); we had discussed DTC formation in such integrated platforms in the Discussion section. We believe that further investigation of the fundamental constraints of big DTC formation in our proposed platform is beyond the scope of the current study; we plan to investigate these and other relevant effects in subsequent publications. Finally, we note that in principle, other optical platforms such as fiber cavities (again mentioned in the Discussion section) can also be used to demonstrated big DTCs, e.g., dispersion managed fibers.

Comment:

* In fig. 4, the authors compare the experimental and the theoretical power spectrum (panels c-d and f-g). I wonder why the authors do not superimpose the two plots, which may show better the agreement between experiment and theory. Moreover, from the comparison of panels c and f, I have the impression that the high peaks in the experimental spectrum for $M=4$ are better defined than those of $M=3$. Is that accidental and due to the particular measurements chosen to present the results, or rather there is a systematical effect for which the spectra are better defined for large M ? In the latter case, does it have some consequences on the spontaneous breaking of the discrete time translation symmetry for different values of M ? Also, I think it would be interesting to understand better which considerations are made in order to choose M (i.e. the frequency detuning between the two pumps).

Response:

We thank the Reviewer for their keen attention to details. As for the presentation of the theoretical and experimental data separately, we have chosen to do so following the common practice in the Kerr microcomb community and also in the interest of presenting clearer visualization and less cluttered figures. Additionally, there are inevitable weak effects such as avoided mode crossings (AMCs) resulting from the spatial interaction of different cavity mode families which leave signatures on the microcomb spectrum. We are unsure what the above comment exactly refers to in terms of the high peaks being “better defined,” but suspect it may refer to AMC signatures which can naturally be more strongly suppressed depending on where the pump is closer to or farther away from them in the spectrum. However, we do not believe any difference comes from a systematic effect benefitting large M . Finally, as highlighted in our Response to the previous Comment, GVD is one of the primary factors to be considered when choosing M .

We should emphasize that it is a well-understood and accepted fact in the Kerr microcomb community that AMCs will occur in microresonators and their signatures will appear in the spectra of microcombs. The main concern is whether the dynamics and stability of the experimentally achieved microcomb will be impacted by the AMC or not. In our case, the AMCs are weak enough to ensure that their signature is vanishingly small (they do not produce strong dispersive waves in the microcomb spectra) and that their impact on microcomb stability is negligible. To clearly communicate this point to the readers, we have added the following to the manuscript (Methods section, the end of the paragraph under *Soliton Self-injection Locking*):

“It should be emphasized that the absence of anti-crossing-induced frequency pinning in the microcomb spectra confirms the negligible effect of avoided mode crossings in our experiments; see *Supplementary Information*, section IV.”

Furthermore, in section IV of the *Supplementary Information* we have included supporting experimental data for 2-FSRs between the pump and the following explanation:

“To elaborate, an AMC pins a high-power harmonic in the microcomb spectrum at the position of the mode crossing [25]. While AMCs are practically inevitable in experiments, whether or not they impact the excitation and stability of a microcomb generated in a certain region can partially be judged based on the existence of a strong comb tooth pinned in the spectrum and fixed as the pump frequency is shifted. This pinning signature is absent in our experiments, neither when the two pumps generate harmonics through four-wave mixing (FWM), Fig. S1, nor when they excite subharmonics and the beatnote of the pumps is changed in successive FSRs in the same pumping region, Fig. S2. This is particularly manifest in the spectrum for 2-FSR separation of the pumps, Fig. S2(e), which shows no sign of an AMC. Instead, high-power harmonics always follow the external pumps and the FWM harmonics that appear at frequency intervals dictated by their beatnote. Therefore, subharmonic generation in our experiments is not impacted by AMC and mode crossings play no role in stabilizing the observed microcombs. This fact is supported also by our numerical modeling data which demonstrates stable pulse train propagation over hundreds of cavity photon lifetimes (thousands of drive periods); see Figs. 3(a, d) in the main text. We have not incorporated any AMC or GVD parameter beyond D_2 in our numerical simulations.”

Comment:

* I had some problems in reading fig. 3-(c) and 3-(f): it took me a while to understand that in the comb spectrum two consecutive high peaks are separated by $m-1$ low peaks. I suggest the authors to adapt the plots in these two panels to make them more readable.

Response:

To address the Reviewer’s concern, we have thinned down the comb lines in the spectrum to make sure the subharmonics are visible in the referenced figure. We have also added insets to the figure to better show the subharmonic. The caption of Fig. 3 has been updated accordingly. The updated figure is included on next page for easier reference.

Comment:

* Generally, I find the bibliography exhaustive and complete. Yet, as the authors said, “recently the investigation of dissipative DTCs has gained traction” [line 81] and I would like to mention the following, very recent works on the emergence of time crystals in driven-dissipative systems:

Seibold et al., Phys. Rev. A 101, 033839;

Lledo et al., New J. Phys. 22 075002;

Minganti et al., arXiv:2008.08075 [quant-ph].

I would kindly ask the authors to consider them and assess if it is worthy to include them in the bibliography.

Response:

We thank the Reviewer for bringing these references to our attention and will be happy to include them in the bibliography. We have included the first of the above suggested papers as Ref. [29] in the revised manuscript. The manuscript is currently at the maximum number of allowable references by the journal, so we have asked the Editor for a possible extension and in case journal policy allows it, we will gladly cite the remaining two references suggested by the Reviewer as well as.

In conclusion, I believe that the quality of the manuscript is high, but I am still not convinced that it satisfy the criteria for publication on Nature Communication. I would invite the authors to revise their manuscript to address my specific concerns, before reaching a final decision.

References cited:

Anderson, M. H. *et al.* (2020) ‘Zero-dispersion Kerr solitons in optical microresonators’, *arXiv:2007.14507 [physics]*. Available at: <http://arxiv.org/abs/2007.14507> (Accessed: 27 January 2021).

Ghasemkhani, M. *et al.* (2019) ‘Dissipative Quartic Kerr Solitons for WDM Applications’, in *Frontiers in Optics + Laser Science APS/DLS (2019), paper JTU4A.84. Laser Science*, Optical Society of America, p. JTU4A.84. doi: 10.1364/FIO.2019.JTu4A.84.

Li, Q. *et al.* (2017) ‘Stably accessing octave-spanning microresonator frequency combs in the soliton regime’, *Optica*, 4(2), pp. 193–203.

Taheri, H. and Matsko, A. B. (2019) ‘Quartic dissipative solitons in optical Kerr cavities’, *Optics Letters*, 44(12), pp. 3086–3089. doi: 10.1364/OL.44.003086.

REVIEWER COMMENTS

Reviewer #3 (Remarks to the Author):

The authors have answered convincingly to all the questions raised in the first round of review. In the revision of the manuscript, they have addressed all the flaws present in the first version: now the conclusions drawn are really sound, and the presentation of the results is very clear and accurate.

Therefore, I am convinced the article is now suitable for publication on Nature Communications.

I thank the authors for their extended replies to my comments. Regrettably, I must admit that the reply didn't add clarity to the manuscript, but rather raised more questions and even concerns from my side.

Let me start with factual mistakes made by the authors as well as highlight places and data, which in my opinion look most doubtful. For convenience, I will refer to the pages of their reply letter to me (Reviewer #2) in {}.

1. In the new abstract the authors state “the first experimental demonstration of a concurrent self-injection locking of two continuous-wave lasers to different modes of a Kerr cavity”. This is incorrect because the phenomenon was already demonstrated in [doi: 10.1038/ncomms8957] by partially the same authors.
2. In response to my first not-numbered comment {2} the authors “strongly disagree that subharmonic generation, when pumping with two lasers is a known thing”. I suggest to look in the Figures 9 and 10 of the first paper on bichromatically driven microcavities [doi:10.1103/PhysRevA.90.013811], also cited by the authors as Ref.[55], where the subharmonic generation was demonstrated in simulations. Experimentally, the result of subharmonic generation (as well as similar soliton spectra) has been demonstrated with bichromatically pumped microresonators in [https://doi.org/10.1364/CLEO_SI.2020.SF2B.6], where the pumps have been separated by large number of FSRs. Furthermore, the subharmonic generation is being identified by the authors as “solitons trapped in the potential lattice”, which has been demonstrated earlier with single-soliton states by pulsed-pumping experiments and modulated pumps. Following the manuscript logic, all these cases have temporally structured pumps and are similar, and hence have been demonstrated.
3. The distinction proposed by the authors between the soliton ordering induced by the second pump and soliton crystals seems artificial and looks as attempt to exaggerate the novelty of the observed results: three distinctions of DTC {2} suggested by the authors, which are in “drastic” contrast to soliton crystals are not clearly formulated: (i) is not drawing any distinction between soliton crystals and DTC. (ii) DW is indeed defined by the pump, but there is a clear sign of symmetry appearance in the soliton crystals. (iii) the repetition rate of the pulse train of the soliton crystal depends on the pump frequency, it particularly depends on its position with respect to a modal crossing. Nevertheless, I accept the distinction suggested by the authors between the soliton crystals ordered using external drive or intrinsic resonator properties and following the notation of recent paper [https://doi.org/10.1038/s41467-021-23172-2] I advise authors to call these states synthesized soliton crystals once they will experimentally prove that the observed comb states are indeed solitons.
4. Similar to the first point, claim on {5} “It was not shown before that it is possible to achieve locking of two lasers to the same cavity using this approach”, is incorrect as I already pointed out above. It was demonstrated in [doi: 10.1038/ncomms8957]. The authors may claim that it was first done for soliton microcomb generation. In the current version of the statement, however, it seems that the try to claim not-existing novelty. In my comment in original reply I specifically highlighted that I was talking not about self-injection locking of two lasers, but about “pumping scheme”.
5. I couldn't find claimed on {5} “solid theoretical study” of DTC in the referred author's previous work [https://doi.org/10.1140/epjd/e2017-80150-6]. Ordered soliton states, observed there in simulations are also called “soliton crystals”, no distinction to the DTC is being proposed or discussed.
6. In the Response to my comment 3 {8} the authors say that “*the assumption that the soliton states demonstrated in our platform “arise from chaotic MI” is completely wrong.*” As well as

“the manuscript shows clearly, both theoretically and experimentally, that the solitons do not appear as a result of any laser sweeping from the MI regime (supporting Turing rolls) to that allowing soliton formation.” This is incorrect. First, as I pointed in my initial comments the description of experimental results are very sparse, and no information about the procedure was given. So the reader has to draw his own conclusions about this. Second, Caption of Fig.2 of the manuscript explicitly says: *“Fig.2. Comb power versus detuning curves for (a) 5-FSR and (b) 8-FSR separation between the two pumps, when laser powers are kept fixed.”* Noise-like MI region can be observed in both Fig.2 (a) and (b) at Detuning ~ 2 and sweep of the detuning is clearly shown in simulations.

Furthermore, no evidence of the repeatability is shown. From experimental point of few the paper is still based on two spectra shown in Fig. 4 (which are the same as in SI, Fig.2S), and a non-soliton comb state shown in Fig.2S(e), where the pumps are separated by 2-FSR. This worries me, as no evidence of non-stochasticity is shown, and my question essentially remained unanswered.

In addition, looking at the characteristic shape of the state obtained in Fig.2S(e) I can conclude that it is more relevant to the comb states earlier demonstrated in [<https://doi.org/10.1364/OL.44.001472>] and [<https://doi.org/10.1038/s41467-020-19804-8>], which have different mechanism and characteristic hyperbolic-like spectral envelope generated by one of the pumps.

7. In response to my comment 4 {8}, authors claim the absence of avoided modal crossings in their system, despite AMCs are clearly present in their experimental spectra:

- First, AMCs in the soliton spectra represent themselves as a deviation from a typical

sech^2 soliton profile, and they are not necessarily revealed as a single pronounced optical mode in the comb spectrum. Besides deviations marked in red on the figure, which appear in all three spectra at the same position and clearly constitute an AMC, I can also see AMCs on other positions, such as ~ 1542.3 nm and supposedly at 1544 nm, which coincide with one of the pumps. Particularly, the position of AMX at 1544 is evidenced from the figure (e), where this mode (first mode on the left from the left pump) apparently starts to break out from the smooth profile of the comb state.

- Second, crystalline microresonators are known to have very large number of spatial modes, and as a consequence large number of modal crossings. Please see dispersion measurements of a crystalline microresonator in [<https://doi.org/10.1103/PhysRevLett.113.123901>]. It is quite hard to achieve low number of spatial modes in such photonic platform, because that would need very small size of the protrusion [<https://doi.org/10.1364/OPTICA.2.000221>], and it is certainly not the case with the radius of curvature of 0.2mm. To relieve my doubts

and prove that used microresonator has been indeed dispersion engineered to exclude AMCs around pumped region as stated in the manuscript, I would suggest authors to demonstrate dispersion measurements of the cavity in the region of interest, which would help to alleviate my concerns. Currently, from presented spectra I can only see multiple AMCs in the results, which can contribute to the ordering of the pulses.

- Third, for both experimental spectra demonstrated by the authors, one of the pumps is located at 1544 nm, where the AMC can be located, but masked by this pump. As I wrote above, even in the additional spectrum demonstrated in Supplementary the mode seems to be deviating from the others. The claim that “frequency pinning is absent” clearly incorrect, because comparing the only two provided spectra the line corresponding to the left pump is clearly pinned. As I already mentioned in my initial comment, the data provided in the manuscript are very limited, and do not support the author claims in my view.
 - Fourth, the statement that the impact of AMC is vanishingly small in their microresonator, because they are not pronounced can not be accepted, because AMCs are clearly visible in experimental spectra, and as was shown in multiple resonators in [<https://doi.org/10.1038/s41567-019-0635-0>] crystallization can happen without pronounced AMCs. In principle this can be easily understood from the LLE equation – any deviation from pure D_2 -only GVD would cause some oscillations on the CW background leading to the soliton interactions via their extended tails.
 - Fifth, as the authors know from previous works on soliton crystals, the AMCs can be located very far from the pump region and contribute to the ordering. Another major concern about the data is that only central part of the comb is shown, which makes the claim of absence of AMCs unsupported.
8. I let myself further express concerns about experimental results besides the ones above. Given very narrow measurement window of the comb spectra the solitonic nature of the combs is questionable (as also pointed out by Reviewer 1). I do not agree that measurements of the beatnote and the optical spectrum are sufficient to claim the presence of soliton pulses in the context of the present manuscript. First, narrow measurement window (only 6 nm) cuts at least half of the information-bearing comb spectrum, and does not let to conclude whether the envelope indeed follows the soliton shape. Second, many strong deviations from the sech^2 envelope profile in multiple places across narrow measurement window, which are clearly seen even in a log-scale, demonstrate significant deviation from the spectral soliton profiles shown in simulations. I would not be surprised if such deviations would be even stronger outside of the measurement window. Third, non-solitonic states can easily provide low-noise performance of the beatnote, as was demonstrated in the first and seminal paper on Kerr combs [<https://doi.org/10.1038/nature06401>]. Thus, given the spectrum uncertainty, the narrow beatnote cannot be used to claim the presence of soliton pulses and their arrangement. Fourth, I would consider the spectrum and beatnote measurements as a prove of solitonic nature if a clear soliton spectral shape was demonstrated, as for example in already cited by the authors paper [Bao et al, 2021], however once authors starting to claim such major advancement and breakthrough in the field of time crystals as they do, one cannot rely on doubtful or handwavy confirmations of key experimental results. I also note that all seminal papers on new **concepts** in the field of soliton microcomb such as demonstration of first solitons [Herr et al, 2014], demonstration of pulsed-pumped solitons [Obrzud et al, 2017]

and even soliton crystals [Cole et al, 2017] provided either FROG or other correlation measurements to undoubtedly demonstrate pulse formation.

To conclude, I do not see a major experimental issue in measuring presence of pulses via FROG or other techniques, and I could not find reasonable explanation of author's reluctance in doing such measurements given the importance of the claims they made in the manuscript.

9. The explanation why the observed soliton ordering can be called "quantum DTC" in the response {9-10} is still unclear to me. First the authors say that Spontaneous breaking of space-time translation symmetry is a quantum phenomenon without mentioning that it was for years studied in classical systems as well []. Then they are making a statement that consequences of Hamiltonian in classical case is not as dramatic as in quantum case with the example of particle in periodic well. Authors say that "in quantum case realization of a symmetry broken steady state is nontrivial and requires an interacting many-body system", and in the next sentence they give a definition of classical DTC: "classical DTCs are periodically driven many-body classical systems which break ergodicity and reveal synchronized motion and breaking of time translation symmetry is not an issue, because ...". I may have to ask to be more concise and straightforward in a sense what a quantum DTCs are: what is their definition and how the manuscript results match the definition. I note, that in the manuscript, on one hand authors assert that they used LLE equation (which is classical) and have demonstrated excellent agreement with their observations, which means that the system is well described via classical approach, but on the other the results are claimed to be "quantum".

Furthermore, claim "quantum timing jitter of the generated soliton train is determined by the quantum frequency noise of the pumps" is not clear and is not supported. First, in contrast to cited work [Bao, 2021] authors didn't measure their jitter noise down to a quantum noise level. Second, the results shown in the cited paper referred to the noise of two counterpropagating solitons generated from the same pump, which is AOM shifted for counter propagating modes by different amount and thus intrinsically highly coherent. While in the present manuscript the pumps originate from different laser cavities and are not shown to be correlating down to quantum level [doi: 10.1038/ncomms8957].

10. Regarding whether the pulse ordering of the solitons due to modulated background, can be claimed a DTC and how it is different from soliton crystals:
 - a. As authors correctly pointed out the time crystals are being a topic of debates since their first suggestion by Wilczek in 2012. As I can see from the manuscript and author explanation, a rigorous definition of the time crystallinity has not been proposed to the reader and no theoretical criteria-matching work has been carried out to confirm that the observation can be claimed discrete time crystals. The definition given in the introduction corresponds to the *closed* systems (*isolated* in the manuscript), and then is being immediately reduced to just "A discrete time crystal (DTC) emerges when the period of the system response is an integer multiple of the drive period". This form is used for the rest of the manuscript. The prove that the observed system can indeed be classified as DTC in the manuscript is mainly focused on explanation that certain ordered soliton states obtained in simulations or potentially observed in experiments reveal a sign of symmetry breaking. I find such approach to be dangerously superficial, as it misleads not only authors but the readers suggesting that a single not-distinctive signature of the phenomena can be used to claim its observation.
 - b. No definition of Dissipative DTC is being proposed in the manuscript. The authors silently imply that dissipative DTC is a DTC, which has just been defined by themselves

as the state where the period of the system response is an integer multiple of the drive period, but with a dissipation. The authors recognize both of these features in parametrically driven damped NLSE and claim that the ordered soliton states in such systems can be claimed dissipative DTCs. Following such logic one can name “dissipative DTC” just a periodically driven pendulum experiencing period doubling bifurcation, which is obviously not the case, because no new *phase state of matter* exist here. The same would apply to other nonlinear parametrically driven systems revealing the response at multiple integer of the drive.

- c. To confirm the above, I would like to bring to authors attention the review on Discrete time crystals [<https://doi.org/10.1146/annurev-conmatphys-031119-050658>], which has been already cited by more than 80 times since 2020, and which while being one of the key papers in the field, to my surprise, was ignored by the authors:
“it might be tempting to classify several superficially similar-looking nonequilibrium phenomena (e.g., period doubling bifurcations, second harmonic generation, Faraday waves, etc.) as time crystals (5, 13–15). However, such time-dependent phenomena generally cannot be classified as phases of matter and share few similarities with equilibrium order.”
- d. Another important feature of a DTC, which was not mentioned by the authors and was not verified in their work is that the system is expected to reveal the spontaneous symmetry breaking in *its lowest energy state*. Lowest energy state of the bichromatically driven system is not a state with multiple solitons but, depending on the detuning and system parameters, is either a modulated background or a single soliton state, which both don’t reveal any symmetry breaking as confirmed by the authors.
- e. I would like to draw the author’s attention to recently published Nature Communications paper [<https://doi.org/10.1038/s41467-021-23172-2>] on the same concept of ordering microcavity solitons using two pumps (not self-injection locked though), named “synthesized soliton crystals”. I would like to highlight two aspects – first is the quality of data, demonstrating the *concept*, and the ability of the authors reproducibly realize crystals with different separations, and second – is the absence of any qualitative distinction claimed by the authors between dichromatically synthesized and native soliton crystals.
- f. To resolve the question of what can be called time crystal (discrete time crystal, dissipative discrete time crystal), and whether the claimed observation of pulse ordering in the manuscript should be distinguished from the soliton crystals, I suggest to the authors give their concrete definitions (without pulling together “convenient” 1-2 matching signatures from literature) and implement detailed criteria-matching work. As one of the prominent examples of such careful work in the field of time crystals I can refer the authors to the work by Sullivan *et al*, which first appeared in arxiv (<https://arxiv.org/abs/1807.09884>) with the title “Dissipative discrete time crystals”. However, after detailed investigation of their results it was later published in New Journal of Physics with changed title “Signatures of discrete time crystalline order in dissipative spin ensembles”. I consider this as responsible and honest move.

As a general remark, I suggest authors to be straighter and more succinct in their further replies. It was often admittedly hard to follow even simple logic of their explanations hidden in the ornate replies spiced with multiple-times iterated statements of novelty and uniqueness of the manuscript results.

Reviewer #4 (Remarks to the Author):

Being more straightforward and clear would definitely help authors to make less inaccuracies and contradictions, as well as would not require me to follow up with even longer reply.

In conclusion, I must note that all above represents merely a half of questions and concerns, which I got after reading the authors response. This is very unusual case for my peer-reviewing practice of many years, which makes me express my concerns rather strongly. I suggest the authors to carefully redo their experiments following mine and other Reviewers concerns, provide more representative data, and carefully revise their claims on the observations of time-crystallinity. In suggesting this, I am particularly worrying for the professional reputation of the authors and would not like to see similar case as to recent *Science* publication [DOI: 10.1126/science.abl5286] happen to them.

Reviewer #5 (Remarks to the Author):

I carefully read the revised paper and the response to the three reviewers.

In general, I can say that I was impressed by the accuracy and completeness of the authors' responses, and I found them very convincing. I was asked to consider in particular the review of the first Referee, whose main criticism is that the observed behavior should be interpreted as "subharmonic entrainment" rather than a "dissipative discrete time crystal". I found the reply of the authors more than exhaustive and I think that they have clearly demonstrated that the referee's concern does not hold.

Yet, I found a bit weak the response of the authors to Comments 3 and 4 of the second referee, who expressed some doubts about the repeatability of the generated soliton states and evidenced a lack of details on how those states were generated. I think that here the responses of the authors were a bit elusive. In the response to Comment 3 they stressed that the referee was wrong when they stated that "the soliton states arise from the chaotic MI" but they do not answer to the specific objection about repeatability. In the answer to Comment 4 they added some details on the procedure followed in the experiment to generate the soliton states, but I still believe, as the second referee, that it is not possible to control the number of solitons in the crystal. I understand that the aim of the paper is to demonstrate the existence of dissipative discrete time crystal, not of controlling them. Nevertheless, I would appreciate that the authors would be more explicit about this point.

Response to Reviewers:

In what follows, we have responded in detail to all referee comments. Font color black is reserved for Reviewer comments or quotes from their previous or current reports, dark blue is used for our responses and quotes from our previous response letter/manuscript/Supplementary Information, lighter blue shows quotes from other papers (e.g., those cited by a referee), and green highlights additions to the manuscript main text/Supplementary Information made in this review round.

Reviewer #3:

The authors have answered convincingly to all the questions raised in the first round of review. In the revision of the manuscript, they have addressed all the flaws present in the first version: now the conclusions drawn are really sound, and the presentation of the results is very clear and accurate. Therefore, I am convinced the article is now suitable for publication on Nature Communications.

Response:

We thank the Reviewer for their careful review of our response letter and revised manuscript.

Reviewer #5 (and #1):

I carefully read the revised paper and the response to the three reviewers.

In general, I can say that I was impressed by the accuracy and completeness of the authors' responses, and I found them very convincing. I was asked to consider in particular the review of the first Referee, whose main criticism is that the observed behavior should be interpreted as "subharmonic entrainment" rather than a "dissipative discrete time crystal". I found the reply of the authors more than exhaustive and I think that they have clearly demonstrated that the referee's concern does not hold.

Response:

We greatly appreciate the Reviewer's careful evaluation of our response letter and revised manuscript.

Comment:

Yet, I found a bit weak the response of the authors to Comments 3 and 4 of the second referee, who expressed some doubts about the repeatability of the generated soliton states and evidenced a lack of details on how those states were generated. I think that here the responses of the authors were a bit elusive. In the response to Comment 3 they stressed that the referee was wrong when they stated that "the soliton states arise from the chaotic MI" but they do not answer to the specific objection about repeatability. In the answer to Comment 4 they added some details on the procedure followed in the experiment to generate the soliton states, but I still believe, as the second referee, that it is not possible to control the number of solitons in the crystal. I understand that the aim of the paper is to demonstrate the existence of dissipative discrete time crystal, not of controlling them. Nevertheless, I would appreciate that the authors would be more explicit about this point.

Response:

We appreciate the Reviewer's valuable comments and address them separately below.

As the reviewer notes, our focus in the current manuscript is on demonstrating the existence possibility of dissipative discrete time crystals (DTCs) in a Kerr-nonlinear optical system. Controllability and switching between different DTC phases require the ability to preferably enhance the probability of the emergence of certain subharmonics (equivalently, certain soliton distributions per roundtrip time) or to manipulate the relative temporal delay of the generated solitons. These are beyond the scope of the current work and have hence not been targeted here. (We have mentioned some of them in passing in the Discussion Section.) As far as robust subharmonic generation occurs in the system, time-translation symmetry breaking (TTSB) and therefore DTC formation is realized.

With that said, we note that in our experiment we have observed multi-stability similar to what is observed in Fig. 2 of the main text, where each middle branch corresponds to a different DTC realization. (The lowest and top-most steps do not correspond to discrete time translation symmetry breaking because they pertain to either no solitons per roundtrip time or the maximum number of possible solitons per roundtrip time.) The multi-stable states we have observed consisted of (1) four-wave mixing of the pumps creating harmonics separated by the beatnote of the pumps [as in Fig. S1(b), with the corresponding RF beatnote

in Fig. 4(b), the blue curve], and (2) subharmonic generation between the pumps, which corresponds to DTC formation [as in Figs. 4(c, f), with the example RF beatnote in Fig. 4(b), the red curve].

As far as repeatability is concerned, after finding the appropriate parameter regime for subharmonic generation in our experiments, we always observed one of the above-mentioned states (1) and (2) in every run of the experiment. In every realization of subharmonic generation between the two pumps, we observed the same envelope and a narrow RF beatnote, hinting at the stable nature of the realized states. As noted above, these observations evidence robust DTC formation in our system. They are also in full agreement with our numerical modeling data. It is worth emphasizing that in the current study, we have observed subharmonic generation for few-FSR separation of the pumps, but suspect that the marriage of dichromatic pumping with self-injection locking of both pumps to Kerr cavity modes of the same modal family involves very rich physics, warranting dedicated investigations beyond our current study of DTCs in this system.

To further clarify these and address the Reviewer's concern, we have added the following to the main text (page 5, bottom of column 1 and continuing over to column 2):

“In our experiment, we have observed multi-stability similar to what is depicted in Fig. 2, where each middle branch corresponds to a different DTC realization; see also Figs. S3(c) and S4 (blue curves) in the *Supplementary Information*. These multi-stable states consisted of (1) four-wave mixing (FWM) of the two pumps creating harmonics separated by their beatnote [as in Fig. S1(b), with the corresponding RF beatnote plotted in blue in Fig. 4(b)], and (2) subharmonic generation between the pumps, which correspond to DTC formation [as in Figs. 4(c, f), with RF beatnote plotted in red in Fig. 4(b)]. Note that different subharmonic arrays translate into different pulse numbers per round-trip time. After finding the appropriate parameter regime for subharmonic generation in our experiments, we always observed one of the states (1) or (2) in every subsequent run. In particular, in every realization of subharmonics between the two pumps, we observed the same envelope and a narrow RF beatnote, hinting at the stable nature of the realized DTC states. In this work we have focused on a few FSRs separating the two pumps, but suspect that the marriage of dichromatic pumping with self-injection locking of both pumps to Kerr cavity modes of the same modal family involves very rich physics, warranting dedicated investigations, e.g., as to the role of pump beatnotes.”

We have also added the following right before the Conclusion section (second column or page 7):

“Furthermore, in this work we have focused on a few FSRs separating the two pumps, but suspect that the marriage of dichromatic pumping with self-injection locking of both pumps to Kerr cavity modes of the same modal family involves very rich physics, warranting dedicated investigations, e.g., as to the role of pump beatnotes.”

As for second referee's Comment 3, we would like to note that the text of the manuscript (both in the original and the 1st-review-round revised submissions) explicitly stated that “Figures 2(a, b) show steady-state” values of the comb power (the paragraph right below Fig. 2, where this figure is first introduced in the text). The fact that the horizontal axis is the detuning does not mean that we have swept the laser from the MI region to the regime of soliton generation. Each point in the plot refers to one simulation run for a few hundred cavity photon lifetimes (till steady state, for non-chaotic microcombs, prevailed). Indeed, to emphasize the discrete nature of the curve (the fact that it does not pertain to a continuous sweep of the

detuning on the x -axis), the data in Fig. 2 were specifically plotted with *separate* dots. To ensure there is no room for misinterpretation, we have added the following sentences to the caption of Fig. 2:

“Each data point corresponds to one numerical integration of the governing equations from a random initial waveform till a steady state (for non-chaotic states) prevailed.”

Also, we have further added the following statement to the discussion of Fig. 2 in the manuscript (first column on page 4):

“Each data point in Fig. 2 indicates one simulation run till steady states, for non-chaotic microcombs, prevailed.”

We hope that these additions/modifications address the Reviewer’s concerns.

Reviewer # 4 (and #2)

I thank the authors for their extended replies to my comments. Regrettably, I must admit that the reply didn't add clarity to the manuscript, but rather raised more questions and even concerns from my side. Let me start with factual mistakes made by the authors as well as highlight places and data, which in my opinion look most doubtful. For convenience, I will refer to the pages of their reply letter to me (Reviewer #2) in {}.

Response:

We thank the Reviewer for their careful reading of the manuscript and insightful comments. Following the Reviewer's suggestion to provide succinct replies, before proceeding to the full response we first present here a summary, wherein C1-C10 refer to Comments 1-10 of the Reviewer or our respective reply. Detailed responses to each point then follow this summary.

Summary of the point-by-point response letter:

1. The Reviewer has referred to many papers, suggesting that they have shown, either theoretically or experimentally, the same phenomena we have demonstrated here (C1- C4, C6, C10). However, the Reviewer's assertion seems to contradict the content of these papers, as revealed by verbatim quotes and specific data and figures therein. Indeed, careful reading clarifies that *none of the cited papers has demonstrated subharmonic generation (SG), nor self-injection locking (SIL) to modes of the same family. Importantly, none has made any connection to discrete time crystals (DTCs)*. We note in passing that the Reviewer has strongly objected to possible avoided mode crossings (AMCs) in our data (C7, addressed in 2.3 below), but evidently overlooked AMCs in the papers they have highlighted, c.f., [Lu, et al., Nat. Commun. 12, 2021] (C3, C10.e) and [Zhang, et al., Nat. Commun. 11, 2020] (C6).

Here is a recap of our comments on these papers; details can be found in the relevant response.

[Liang, *et al.*, Nat. Commun., 6, 2015] (cited in C1, C4), is an earlier paper by co-authors Matsko and Maleki, which, as part of their study, implemented SIL to *two different mode families*. Importantly, the lasers operated in the *low-power linear regime* such that the cavity Kerr nonlinearity played no role in this portion of their study and certainly *no SG* was observed.

[Hansson and Wabnitz, Phys. Rev. A, 90, 2014] (C2), is a theoretical study and used solitons *injected* into the cavity; these solitons did not appear because of SG.

[Karpov, *et al.*, paper SF2B.6, CLEO 2020] (C2) used two free-running lasers (no SIL) pumping two independent solitons with different local FSRs and observed no SG. Absence of SG is clear from the different harmonic spacings for the two combs excited by the pumps.

In [Obrzud, *et al.*, Nat. Photon., 11, 2017] (C2) and [Lu, *et al.*, Nat. Commun. 12, 2021] (C3), the drive and generated microcombs have the same repetition rates, so no SG/DTC is observed.

[Bao, *et al.*, Opt. Lett. 44, 2019] (C6) does not demonstrate SG, but the generation of two microcombs pumped by two free-running lasers (no SIL) in two different mode families/polarizations (TE and TM) which synchronize their repetition rates through the entrainment of the weaker pulse train by the strong soliton train.

[Zhang, *et al.*, Nat. Commun. 11, 2020] (C6) again used two free-running lasers (no SIL) pumping two independent solitons synchronizing their repetition rates because of temporal overlap, but failed to demonstrate SG.

2. In C5-C8, the Reviewer has repeated comments already addressed in the first review round, evidently missing the very text of our manuscript or the Supplementary Information (SI). We have responded to all of the comments in detail:

(2.1) In C5, we have noted that [Taheri, *et al.*, Eur. Phys. J. D 71, 2017] provided a generic formulation of Kerr cavity dichromatic pumping and utilized it to study the case of a weak second pump introduced *after* a soliton is generated by another strong pump. That study does not mention DTCs because it does not demonstrate SG. Yet, the derived generic mean-field approach can be exploited also in the current study.

(2.2) To address C6, we have again stressed that DTC formation in our system does not entail passage through chaos. While this was explicit in the text, we have added further emphasis to the caption of Fig. 2 and its discussion in the text to leave no room for misinterpretation. We have also added another paragraph further addressing the Reviewer's question on repeatability (columns 1 and 2 on page 5).

(2.3) To decisively address the speculations about the contributions of AMCs to SG (C7 and SI), we performed a thorough analysis by incorporating AMCs *at frequencies suspected by the Reviewer*. Contrary to their assertion, we confirmed that AMCs do not help SG but destabilize it. This finding further distinguishes our study by emphasizing the dominant role of the pumps and puts to rest any objection to our conclusions. Consistent with our data, suppression of AMCs near the pumps is also observed in [Lu, *et al.*, Nat. Commun. 12, 2021] cited by Reviewer in C3 and C10e. As we noted above, the latter paper shows clear AMC-like disruptions in their integrated dispersion data, which has not affected the dual pumping phenomena, further supporting the dominating impact of the pumps over weak AMCs.

(2.4) In C8, the Reviewer has requested temporal pulse measurements. We had already convinced another Reviewer that time-domain measurements are unnecessary and that the observation of SG in the frequency domain proves DTC formation. There is indeed *no need* for pulse shape confirmations because a specific pulse profile is not required for discrete time-translation symmetry breaking.

(2.5) In C9, the Reviewer has again objected to our use of a mean-field model for studying DTCs. Besides giving detailed examples, we have noted that the only other experimental demonstration of dissipative DTCs (published very recently), also essentially uses a mean-field approach with small quantum fluctuations (the truncated Wigner approximation, TWA). We should add that this paper cites the arXiv preprint of our manuscript as another dissipative DTC demonstration.

3. Finally, C10 constitutes a set of questions on time crystals asked only in this review round. We were surprised by these comments on DTC basics including their definition and basic properties of Floquet system such as quasi-energies. These aspects are clearly more suitable for papers describing the fundamentals of DCT and are already available in the literature. Besides answering these questions, we have referred to DTC criteria *previously spelled out in our manuscript*. To ensure the Reviewer's concern is

addressed, we have also explicitly reiterated these criteria and our system’s compliance with all of them in the opening paragraph of the Discussion Section (page 5).

□ □ □ □

We proceed to the Reviewer’s comments and our detailed responses below.

Comment:

1. In the new abstract the authors state “the first experimental demonstration of a concurrent self-injection locking of two continuous-wave lasers to different modes of a Kerr cavity”. This is incorrect because the phenomenon was already demonstrated in [doi: 10.1038/ncomms8957] by partially the same authors.

Response:

Unfortunately, the Reviewer’s statement overlooks *key differences*. In the referenced paper injection locking to a cavity was demonstrated, but the lasers pumped modes belonging to different mode families of the resonator to avoid any nonlinear effects associated with the dual pumping. Additionally, the laser powers were selected to be very small in that work, and hence they did not enter the regime of nonlinear interaction between the pumps. Thus the Kerr nonlinearity of the cavity played no role in the referenced experiment in the cited paper. As a result, the cavity cannot be called “Kerr” in that earlier experiments as only the linear response was studied.

In contrast to the work cited by the Reviewer, in the current work we lock two high power lasers to *the same mode family* in the cavity. Additionally, these lasers have high enough power to interact nonlinearly with each other due to the Kerr nonlinearity of the cavity. Interestingly, the interaction does not disrupt the self-injection locking process. To address the Reviewer’s comment and to make this point clearer, we have modified the statement in the Abstract as follows:

“To the best of our knowledge, this is the first experimental demonstration of a dissipative DTC, as well as the concurrent self-injection locking of two continuous-wave lasers to different modes of the same family in a Kerr cavity.”

A similar change is made to the main text (page 2, column 1):

“... to two **same-family** cavity modes”

Comment:

2. In response to my first not-numbered comment {2} the authors “strongly disagree that subharmonic generation, when pumping with two lasers is a known thing”. I suggest to look in the Figures 9 and 10 of the first paper on bichromatically driven microcavities [doi:10.1103/PhysRevA.90.013811], also cited by the authors as Ref. [55], where the subharmonic generation was demonstrated in simulations. Experimentally, the result of subharmonic generation

(as well as similar soliton spectra) has been demonstrated with bichromatically pumped microresonators in [https://doi.org/10.1364/CLEO_SI.2020.SF2B.6], where the pumps have been separated by large number of FSRs. Furthermore, the subharmonic generation is being identified by the authors as “solitons trapped in the potential lattice”, which has been demonstrated earlier with single-soliton states by pulsed-pumping experiments and modulated pumps. Following the manuscript logic, all these cases have temporally structured pumps and are similar, and hence have been demonstrated.

Response:

(a) The referenced comment in the Reviewer’s previous report concerned effects such as “modal crossings, dispersive wave formation” and “birefringence” which we discussed in detail and addressed in the *Supplementary information* added to the manuscript. As we explained, none of these effects define a discrete symmetry in the system, and without discrete symmetry, speaking of breaking it and DTC formation is irrelevant.

(b) As for Figs. 9 and 10 in [<https://doi.org/10.1103/PhysRevA.90.013811>], the Reviewer’s statement does not apply because as it is specifically stated in the text of the referenced paper, the solitons appearing in these Figures were injected into the cavity and did not rise because of subharmonic generation. This point appears on page 6, left column, line 23 in the *Physical Review A* paper cited by the Reviewer):

“An example of a single-cavity soliton that has been **injected** into the cavity to sit on top of a 16-bit clock pattern obtained for the modulation frequency $\Omega = 8$ is shown in Fig. 9. The corresponding frequency comb has about 600 modes with an intensity larger than -120 dB, which is somewhat broader than the cavity soliton comb for the single-pumped case. Another example featuring a pattern of multiple-cavity solitons is shown in Fig. 10.”

(In the excerpt quoted above, underlining and boldening of the text is done by us to draw attention to the explicit statement which refutes the objection in the above Reviewer comment.) Indeed, the said paper considers the possibility of making an “optical soliton buffer” in which CW background modulation acts as a clock signal “onto which the cavity solitons can be written.” The authors specify that this writing takes place by injecting solitons generated by a separate source, and as an example they particularly refer to an earlier work in which soliton pulses generated by a separate mode-locked laser were injected into a passive fiber cavity (Ref. [18] of the *Physical Review A* paper). Therefore, the cited paper lays no claim on subharmonic generation and does not question the novelty of what we have reported in this manuscript. Furthermore, with self-injection locking and its accompanying benefits, distinctions of our results are clear.

(c) Regarding the recent conference paper highlighted by the Reviewer, i.e.,

[M. Karpov, M. H. P. Pfeiffer, A. Lukashchuk, J. Liu, and T. J. Kippenberg, “Spectral multiplexing of dissipative Kerr solitons in a single optical microresonator,” in Conference on Lasers and Electro-Optics, OSA Technical Digest (Optical Society of America, 2020), paper SF2B.6],

https://doi.org/10.1364/CLEO_SI.2020.SF2B.6,

after careful review of this paper and its references, we observed that this work does not demonstrate any trace of subharmonic generation: two lasers are pumping *two independent solitons* in different regions of the spectrum (i.e., 1300 nm and 1550 nm). The lasers are *not* self-injection-locked to their respective cavity modes, and even though they are claimed to drive modes of the same family, they generate independent solitons which, because of dispersion, are pumped at different free spectral ranges (FSRs), i.e., 197.4 vs. 196.3 GHz for one of the cavities used in the experiments and 1000.4 vs. 998.6 GHz for the other. In all but one of the reported experimental traces, the spectra of the solitons do not even overlap, and no comb lines—even vaguely and distantly—indicating subharmonic generation appear between the pumps, so they clearly do not generate subharmonics. Finally, in the only spectral trace showing overlapping spectra, it is specifically stated that the solitons have *different comb line spacings* (page 2 in the cited conference paper):

“Here [i.e., in Fig. 1(d)], the system operates in a single soliton states at 1300 nm, while having a multiple-soliton state at 1550 nm. The soliton combs overlap, and as in the case of 200-GHz device have difference in comb line spacing.”

Interestingly, one of the main claims of the conference paper is the ability to tune the line spacing in one of the two soliton spectra without affecting the other one, which by itself shows the independence of the two solitons and the irrelevance of the results to subharmonic generation and DTCs. Therefore, in this work there is no clear relationship between the beatnote of the two driving lasers and the repetition rate of the two sets of soliton/pulse trains leaving the cavity in the time domain. Therefore, no subharmonic generation, no time translation symmetry breaking (TTSB), and hence no DTC formation is demonstrated in the conference paper the Reviewer has highlighted.

It is worth emphasizing that we do not contend the possibility of demonstrating DTCs in the system discussed in the referenced conference paper (with two pumps with widely separated frequencies); that would indeed be a *proof of our predictions*. But as detailed above, the experiments reported in the conference proceeding *definitely do not show DTC formation*. This, in turn, underscores the significance and novelty of our approach: we managed to identify an operation regime which more simply and more elegantly demonstrates DTCs.

(d) The Reviewer indicates that soliton trapping “demonstrated earlier with single-soliton states by pulsed-pumping experiments and modulated pumps” are relevant to the current work. We presume the Reviewer refers to the following paper:

[Obrzud, E., Lecomte, S. & Herr, T. Temporal solitons in microresonators driven by optical pulses. Nature Photon 11, 600–607 (2017)],

<https://doi.org/10.1038/nphoton.2017.140>.

Subharmonic generation is not demonstrated in this work either because, as is clear from Fig. 1(b) in this paper, all pulses leaving the resonator carry a pulse on top of them, so that no discrete TTSB occurs in the system.

(e) Last but not least, we should emphasize that none of the works cited by the Reviewer (neither the *Physical Review A* paper nor the conference article) have even recognized (or even mentioned

in passing) any relationship between the discussed systems and the possibility of realizing DTCs in them. Therefore, apart from the certain key differences we enumerated above (externally generated injected solitons and no trace of subharmonic generation and TTSB), the novelty of our results from the perspective of DTC generation in an optical platform remains valid and strong.

Comment:

3. The distinction proposed by the authors between the soliton ordering induced by the second pump and soliton crystals seems artificial and looks as attempt to exaggerate the novelty of the observed results: three distinctions of DTC {2} suggested by the authors, which are in “drastic” contrast to soliton crystals are not clearly formulated: (i) is not drawing any distinction between soliton crystals and DTC. (ii) DW is indeed defined by the pump, but there is a clear sign of symmetry appearance in the soliton crystals. (iii) the repetition rate of the pulse train of the soliton crystal depends on the pump frequency, it particularly depends on its position with respect to a modal crossing. Nevertheless, I accept the distinction suggested by the authors between the soliton crystals ordered using external drive or intrinsic resonator properties and following the notation of recent paper [<https://doi.org/10.1038/s41467-021-23172-2>] I advise authors to call these states synthesized soliton crystals once they will experimentally prove that the observed comb states are indeed solitons.

Response:

(a) As we explained in detail in the text and the *Supplementary Information* section of the revised manuscript submitted in the first review round, for discrete TTSB to occur, a discrete symmetry should first exist in the system. This requirement, as the formulation in the *Supplementary Information* (SI section III) explicitly shows, is not met in the presence of avoided mode crossing (AMC) which underlie the existence of soliton crystals. Therefore, point (i) in the comment above has already been addressed explicitly in our previous response letter.

(b) It is obvious that there is *some* relationship between the frequency of the pump and that of the dispersive wave (DW). But, as we explained at length in our previous response letter, this relationship is not one which could define a discrete time symmetry in the system which is the prerequisite for TTSB and DTC formation. It is true that pump frequency chooses the excited mode family in the cavity and then the dispersion of the cavity defines the spectral position of the DW. However, as we explained clearly in the formulation of microcombs supporting DW emission in the presence of 3rd and higher order dispersion in section II of the *Supplementary Information*, this relationship does not translate into the definition of a discrete symmetry in the system. We emphasize that before breaking the symmetry, that symmetry should exist. The Reviewer refers to a relationship between the pump and DW frequencies, but fails to show how it translates into a discrete temporal symmetry in the driven cavity. That is because, as the formulations in the *Supplementary Information* (SI section II) unequivocally clarifies, this symmetry does not exist.

(c) Regarding the following paper referenced by the Reviewer,

[Lu, Z., Chen, HJ., Wang, W. et al. Synthesized soliton crystals. *Nature Communications* 12, 3179 (2021)],

<https://doi.org/10.1038/s41467-021-23172-2>,

we should note that this paper does not demonstrate subharmonic generation: the repetition rate of the microcombs matches the beatnote of the pumps. As a result, this work does not report DTCs. Furthermore, as we explain at length below, even though the match between theory and experiment sufficiently supports soliton and DTC formation in our system, the generation of a soliton (i.e., a hyperbolic secant shaped pulse) is not a requirement for establishing subharmonic generation and the realization of a DTC.

We should emphasize that our earlier theoretical work [Taheri, H., Matsko, A. B. & Maleki, L. Optical lattice trap for Kerr solitons. *The European Physical Journal D* 71, 153 (2017)], cited by this published *Nature Communications* paper as Ref. [51] in the main text and as Ref. [4] in its Supplementary Information, forms the theoretical backbone of the paper referred to by the Reviewer.

Comment:

4. Similar to the first point, claim on {5} “It was not shown before that it is possible to achieve locking of two lasers to the same cavity using this approach”, is incorrect as I already pointed out above. It was demonstrated in [doi: 10.1038/ncomms8957]. The authors may claim that it was first done for soliton microcomb generation. In the current version of the statement, however, it seems that the try to claim not-existing novelty. In my comment in original reply I specifically highlighted that I was talking not about self-injection locking of two lasers, but about “pumping scheme”.

Response:

This Comments constitutes a few parts. We respond to each separately.

- (a) The Reviewer has first repeated their Comment 1 above, which conflates the self-injection locking (SIL) of two *relatively strong* lasers to two modes in the *same mode family* (the current work), with the SIL of two *weak* lasers in *two different mode families* (the cited *Nature Communications* paper from 2015). As we have mentioned before, the novelty claim is completely correct when we speak about a “Kerr cavity”. Previously, SIL operation was observed if low-power lasers were locked to two different mode families. In this configuration the lasers cannot impact each other. In the current work, the second laser pumps the same mode family as the first one and the power of the lasers is large enough to generate harmonics that might mutually interact and disrupt the SIL process. As we have reported, this disruption did not happen because of a novel subharmonic generation process. Therefore, again, the configuration and results reported here are completely novel.
- (b) In the second part of their Comment, the Reviewer has stated that

“The authors may claim that it [i.e., SIL] was first done for soliton microcomb generation. In the current version of the statement, however, it seems that the try to claim not-existing novelty.”

This statement made by the Reviewer misses that the primary claim of our paper, as is explicitly expressed in the manuscript, constitutes “the first experimental demonstration of a dissipative DTCs, as well as the concurrent self-injection locking of two continuous-wave lasers to different modes of a Kerr cavity.” This is what we have claimed and we firmly stand by.

- (c) We are surprised that the Reviewer has changed their statement compared to their first referee report (the “original reply,” as they have called it in the above comment). We repeat *verbatim* the Reviewer’s statement from their comments in the first review round:

“Second, *the dual-pumped self-injection locked scheme experimentally exploited by the authors can be claimed to be new.*”

In other words, it seems that the Reviewer had previously *accepted* the novelty of using SIL in our two-pump driving scheme, but they withdrew it in the second set of comments. As we explained in our first response letter and again detailed above in this document, the first comment was a correct assessment by the Reviewer. The objection they had made in this Comment (the second of the un-numbered comments in their original reply) was:

“... however, the results that the soliton could be arranged by introducing the second pump seem quite obvious and expected to me.”

This objection we had addressed in detail in our previous response letter (on page 4 of 12 response to Reviewer #2).

Comment:

5. I couldn’t find claimed on {5} “solid theoretical study” of DTC in the referred author’s previous work [<https://doi.org/10.1140/epjd/e2017-80150-6>]. Ordered soliton states, observed there in simulations are also called “soliton crystals”, no distinction to the DTC is being proposed or discussed.

Response:

The referenced earlier paper of ours

[Taheri, H., Matsko, A.B. & Maleki, L. Optical lattice trap for Kerr solitons. *Eur. Phys. J. D* 71, 153 (2017)],

<https://doi.org/10.1140/epjd/e2017-80150-6>,

did not mention DTCs because it did not demonstrate subharmonic generation. However, it provided a generic formulation of the dual-pump excitation of Kerr microcombs which we have referred to for completeness. The cited paper, unlike the current work, focused on a *weak second/auxiliary pump which was introduced after the generation of a soliton and swept near one of its harmonics to lock to it.* This general formulation remains valid in the case of a strong auxiliary pump (which was not studied in the previous work) and includes both symmetric and asymmetric pumping schemes. The asymmetric pumping scheme is similar to what is performed in the current work

with the *important* difference that in the previous paper the second pump was *very weak* compared to the main pump while the symmetric pumping scheme reduced to the one used in the *Physical Review A* paper [<https://doi.org/10.1103/PhysRevA.90.013811>] highlighted by the Reviewer in their Comment 2 above.

In the case discussed in our previous work referenced by the Reviewer, a weak auxiliary pump is slowly turned on after a soliton is formed by the stronger main pump. Then the auxiliary pump is swept in the vicinity of its nearest soliton harmonic to lock to it. This procedure is clearly very different from subharmonic generation reported in the present manuscript, even though the mathematical framework proposed in the previous work stays valid in the presence of a strong auxiliary pump. We have noted numerous times so far that the differences between DTCs and soliton crystals created in the presence of AMCs have been detailed in the *Supplementary Information* section.

Comment:

6. In the Response to my comment 3 {8} the authors say that “*the assumption that the soliton states demonstrated in our platform “arise from chaotic MI” is completely wrong.*” As well as “*the manuscript shows clearly, both theoretically and experimentally, that the solitons do not appear as a result of any laser sweeping from the MI regime (supporting Turing rolls) to that allowing soliton formation.*” This is incorrect. First, as I pointed in my initial comments the description of experimental results are very sparse, and no information about the procedure was given. So the reader has to draw his own conclusions about this. Second, Caption of Fig.2 of the manuscript explicitly says: “*Fig.2. Comb power versus detuning curves for (a) 5-FSR and (b) 8-FSR separation between the two pumps, when laser powers are kept fixed.*” Noise-like MI region can be observed in both Fig.2 (a) and (b) at Detuning ~ 2 and sweep of the detuning is clearly shown in simulations.

Response:

We take issue with the Reviewer’s comments for reasons describe below.

The text of the manuscript (both in the original and the 1st-review-round revised submissions) explicitly clarifies that “Figures 2(a, b) show **steady-state**” values of the intra-cavity comb energy (the paragraph right below Fig. 2, where this figure is first introduced in the text). Indeed, to emphasize the discrete nature of the curve (the fact that the curve does not pertain to a continuous sweep of the detuning on the x -axis and that each data point corresponds to one run/integration of the governing equations till steady state is achieved), the data in Fig. 2 were specifically displayed with *separate dots*.

Again, we emphasize that our experimental data pertain to running the experiments in the soliton-forming region of Fig. 2 panels (the step-like regions), so they do not entail sweeping the laser at or from the modulational instability (MI) and chaotic regions, and they have nothing to do with the chaotic MI operation regime. This confirms that the germane statement in the text and the explanations provided in the previous response letter are correct.

Apart from the above response and to further clarify, we have added the following sentences to the caption of Fig. 2 to make sure there is no room for misinterpretation.

“Each data point corresponds to one numerical integration of the governing equations from a random initial waveform till a steady state (for non-chaotic states) prevailed.”

Also, to further emphasize this point and for extra clarity, we have added the following statement to the discussion of Fig. 2 in the manuscript (column 1 on page 4):

“Each data point in Fig. 2 indicates one simulation run till steady states, for non-chaotic microcombs, prevailed.”

Comment (6 – Continued):

Furthermore, no evidence of the repeatability is shown. From experimental point of view the paper is still based on two spectra shown in Fig. 4 (which are the same as in SI, Fig.2S), and a non-soliton comb state shown in Fig.2S(e), where the pumps are separated by 2-FSR. This worries me, as no evidence of non-stochasticity is shown, and my question essentially remained unanswered.

Response:

We attempted to answer this question in detail in the previous response letter (first review round). As we had detailed before and further explained in response to the previous part of this Comment, the generation of the DTC states in our platform does not entail passing through the chaotic regime, therefore the repeated objection does not apply. Further evidence of the “non-stochasticity” of the generated DTC structure is the low-noise microwave beatnote signals generated by the subharmonics on a fast photodiode; it is well known and widely accepted in the Kerr microcomb community that stochastic signals do not generate spectrally pure microwave signals.

With respect to repeatability, we have shown the pumps separated by two, three, and four FSRs can all generate subharmonics. After the appropriate parameters/region for subharmonic generation was identified, we were always able to observe low phase noise, stable subharmonic generation with the same microcomb spectral envelope. To further clarify these and address the Reviewer’s concern, we have added the following to the main text (page 5, bottom of column 1 and continuing over to column 2):

“In our experiment, we have observed multi-stability similar to what is depicted in Fig. 2, where each middle branch corresponds to a different DTC realization; see also Figs. S3(c) and S4 (blue curves) in the *Supplementary Information*. These multi-stable states consisted of (1) four-wave mixing (FWM) of the two pumps creating harmonics separated by their beatnote [as in Fig. S1(b), with the corresponding RF beatnote plotted in blue in Fig. 4(b)], and (2) subharmonic generation between the pumps, which correspond to DTC formation [as in Figs. 4(c, f), with RF beatnote plotted in red in Fig. 4(b)]. Note that different subharmonic arrays translate into different pulse numbers per round-trip time. After finding the appropriate parameter regime for subharmonic generation in our experiments, we always observed one of the states (1) or (2) in every subsequent run. In particular, in every realization of subharmonics between the two pumps, we observed the same

envelope and a narrow RF beatnote, hinting at the stable nature of the realized DTC states. In this work we have focused on a few FSRs separating the two pumps, but suspect that the marriage of dichromatic pumping with self-injection locking of both pumps to Kerr cavity modes of the same modal family involves very rich physics, warranting dedicated investigations, e.g., as to the role of pump beatnotes.”

We reiterate that our focus in the current work remains demonstrating the existence of DTCs. Therefore, controllability is outside the scope of this manuscript.

Comment (6 – Continued):

In addition, looking at the characteristic shape of the state obtained in Fig.2S(e) I can conclude that it is more relevant to the comb states earlier demonstrated in [<https://doi.org/10.1364/OL.44.001472>] and [<https://doi.org/10.1038/s41467-020-19804-8>], which have different mechanism and characteristic hyperbolic-like spectral envelope generated by one of the pumps.

Response:

None of these papers demonstrate subharmonic generation, nor do they discuss or allude (implicitly or explicitly) to discrete TTSB. They are not, and do not claim to be, DTCs. In both papers, the second pulse train is either pumped in a different mode family (TE vs. TM) or, if pumped in the same mode family, it exists independently.

These papers, which we discuss in detail separately below, are:

Changjing Bao, Peicheng Liao, Arne Korodts, Lin Zhang, Andrey Matsko, Maxim Karpov, Martin H. P. Pfeiffer, Guodong Xie, Yinwen Cao, Ahmed Almainan, Moshe Tur, Tobias J. Kippenberg, and Alan E. Willner, “Orthogonally polarized frequency comb generation from a Kerr comb via cross-phase modulation,” *Opt. Lett.* 44, 1472-1475 (2019),

<https://doi.org/10.1364/OL.44.001472>,

and

Zhang, S., Silver, J.M., Bi, T. *et al.*, “Spectral extension and synchronization of microcombs in a single microresonator,” *Nat. Commun.* 11, 6384 (2020),

<https://doi.org/10.1038/s41467-020-19804-8>.

We should start by noting that we are *intimately familiar* with the first paper above, as one of us (Andrey Matsko, underlined in the citation above) is a co-author on this paper. This *Optics Letters* paper does not demonstrate subharmonic generation, but the generation of two microcombs in two different polarizations/mode families (TE and TM) which synchronize their repetition rates, essentially through entrainment of the weaker pulse train by the strong soliton train. One of the combs (pumped at 1555.9 nm) is strong and seeds the other weak one (pumped at 1544.6 nm) through cross phase modulation at the same comb teeth spacing and with an offset. The fact that this is not subharmonic generation but entrainment is further clarified by noting that the dynamics is described

by two coupled modified Lugiato-Lefever equations (LLEs), Eqs. (1) and (2) in this paper, one for each mode.

As for the second paper referenced by the Reviewer, it is concerned with driving two different combs (either in the same or in two different mode families, and in the latter case, similar to the conference paper referenced by the Reviewer in their Comment 2 above). The authors examine the repetition rates and mutual offset of the generated combs by focusing on their overlap regions near 1410 nm and argue that the repetition rates lock. As noted above, equal microcomb spacing evidences *entrainment, not subharmonic generation*. The authors have shown that even though the repetition rates synchronize, *the microcombs maintain an offset of approximately 700 MHz*. More specifically, for the set of experiments where both pumps drive microcombs in the same mode family in this paper, the plotted downshifted RF spectra (Fig. 3) shows that there is an offset between the two comb teeth in the overlap region. See in particular the paragraph right below Fig. 2 on page 3 in this paper:

“Considering the 0.02 nm (4 GHz) resolution of the OSA, we can conclude that the optical spectra shown in Fig. 2 are composed of two different frequency combs, resulting from the bichromatic pumping. Each comb line seen on the OSA in the overlapping region actually consists of two comb lines, one from each pump.”

(Underlining and italicization is by us.) This paragraph pertains to the case where both pumps excite modes in the *same* mode family, and means that even though the two microcombs synchronize to have the same repetition rate, they do not connect to form a continuous comb between the two pumps. That is why their overlap has a ≈ 700 MHz offset, as seen in Fig. 3. This notion is also explicitly stated in the Introduction (page 2, left column):

“Our results show that the two combs’ repetition rates synchronize, while maintaining an overall offset as a result of the two independent pump lasers.”

(Again, underlining by us). This is unequivocal evidence that the observed phenomenon, while remarkable in its own right, is not subharmonic generation and differs from what we have demonstrated, i.e., harmonics generated and supported by and between the two pumps.

Comment:

7. In response to my comment 4 {8}, authors claim the absence of avoided modal crossings in their system, despite AMCs are clearly present in their experimental spectra:

Response:

We note at the outset that our technical results and detailed discussion below fully address the Reviewer’s objection to our results regarding AMCs and confirm our conclusions. But before that discussion, we emphasize that the Reviewer’s statement above is evidently based on some misinterpretation. Indeed, what the Reviewer has attributed to us are not our words. We did *not* claim the *total absence* of AMC in the resonator, but the fact that they do not impact the subharmonic generation process. This was clearly communicated in our previous response letter (on page 9 of Response to Reviewer 2): “We should add that it is an accepted fact among all experimentalists in the Kerr microcomb community that AMCs will occur in microresonators and their signatures will appear in the spectra of microcombs. The main question to be asked is whether the dynamics and stability of the experimentally achieved microcomb will be impacted by the AMC or not.”

The same notion is also acknowledged in the *Supplementary Information* Section IV: “While AMCs are practically inevitable in experiments,” To ensure the Reviewer’s misunderstanding is avoided by the reader, we have replaced “strong comb tooth” by “strong disruptions in the comb teeth” so that the rest of the sentence reads (last paragraph on page 3 in the *Supplementary Information*):

“While AMCs are practically inevitable in experiments, whether or not they impact the excitation and stability of a microcomb generated in a certain region can partially be judged based on the existence of strong disruptions in the comb teeth pinned in the spectrum and fixed as the pump frequency is shifted.”

As explained in our previous response letter and previous review round revised text, in our experiments the AMCs are weak enough to ensure that their impact on microcomb stability is negligible. This notion is supported by the fact that with and without subharmonic generation, the RF

beatnote signal is very narrow in our experiments; see Fig. 4(b) in the manuscript. As our detailed analysis reported below shows, when fictitious AMCs are added to the dispersion profile, they will destabilize the microcombs, which would translate into wide RF signals. Only when added AMCs are small and do not interfere with subharmonic generation, can microcombs survive and exist stably. It is worth further emphasis that, as already stated in the manuscript (both the main text and the *Supplementary Information*), our numerical simulations clearly showed that subharmonic generation in our system does not rely on AMCs. Indeed, as is shown in

[T. Herr, V. Brasch, J. D. Jost, I. Mirgorodskiy, G. Lihachev, M. L. Gorodetsky, and T. J. Kippenberg, “Mode Spectrum and Temporal Soliton Formation in Optical Microresonators,” *Phys. Rev. Lett.* 113, 123901 (2014)],

<https://doi.org/10.1103/PhysRevLett.113.123901>

AMCs will significantly limit the region of soliton existence in the pump detuning vs. power plane and only when AMCs are very weak near the pump can solitons form stably in the resonator. This notion is clearly stated in the text of the above reference, e.g., in the abstract,

“Avoided mode crossings induced by linear mode coupling in the resonator mode spectrum are found to prevent soliton formation when affecting resonator modes *close to the pump laser frequency*.”

(Underlining and italicization is by us.) This notion is one of the key findings of the above work (hence highlighted by the authors in its abstract) and is evident also in Fig. 4(d). Therefore, we stand by our statement in the response letter and manuscript *Supplementary Information* because AMCs, especially as close to the pumps as suggested by the Reviewer, even if present, ought to be weak and their impact on subharmonic generation has to be fully dominated by the two pumps to allow stable subharmonic generation.

Comment (7 – Continued):

- First, AMCs in the soliton spectra represent themselves as a deviation from a typical sech^2 soliton profile, and they are not necessarily revealed as a single pronounced optical mode in the comb spectrum. Besides deviations marked in red on the figure, which appear in all three spectra at the same position and clearly constitute an AMC, I can also see AMCs on other positions, such as ~ 1542.3 nm and supposingly at 1544 nm, which coincide with one of the pumps. Particularly, the position of AMX at 1544 is evidenced from the figure (e), where this mode (first mode on the left from the left pump) apparently starts to break out from the smooth profile of the comb state.

Response:

As noted above, the claim that strong AMCs so close to the pump can exist and impact subharmonic generation is not supported by the consensus among the Kerr microcomb community that AMC near the pump disrupts and avoids the generation of stable pulses [Herr, T., Brasch, V., Jost, J. D., Mirgorodskiy, I., Lihachev, G., Gorodetsky, M. L. &

Kippenberg, T. J., “Mode Spectrum and Temporal Soliton Formation in Optical Microresonators,” Phys. Rev. Lett. 113, 123901 (2014)]. This is in fact one of the main messages of the cited PRL paper by Herr, *et al.* Furthermore, we have run extensive numerical simulations, described below, which support the fact that if non-negligible AMCs existed so close to the pumps as suggested by the Reviewer, they would *not* assist subharmonic generation, but instead *destabilize* soliton microcombs, leading to the eradication of the steps; see Figs. R1, R2, and R3. Figures R1 and R2 pertain to the case of an AMC 4 FSRs away from the stronger pump and Fig. R3 to that of an AMC 5 FSRs from it, as the Reviewer has suspected. Each curve shows the results of hundreds of numerical simulations: each data point corresponds to one numerical simulation of the system till steady state (if at all existent) prevailed. As is vividly evident in these results, subharmonic generation is the consequence of dual pumping alone and AMCs suspected by the Reviewer, if present, would have only disrupted soliton formation and would definitely not have assisted subharmonic generation. This is in opposition to the Reviewer’s point on the impact of AMCs, because they suspect AMCs might be responsible for the creation of subharmonics. Below we expound the results.

We had noted in the manuscript that our experiments were performed in a regime where none of the pumps could generate solitons independently. This is verified in Fig. R1(a) below, the comb power vs. detuning curve, which illustrates that with 1 pump, the soliton formation regime is very narrow such that we operated to the right of this sliver, indicated by the dashed vertical line and the red arrow pointing to the right. (We had already expounded in detail that a Kerr cavity pumped by a single laser does not possess discrete time-translation symmetry and hence is irrelevant to DTCs, no matter with or without AMCs; see the Supplementary Information sections I and III.) System parameter values match those of our experiments and are the same as those in Fig. 4(d, g) in the main text. Figure R1(a) corresponds to the stronger pump. The weaker pump, as noted in the text, is sub-threshold and so does not obviously result in hyperparametric sideband generation or soliton formation.

Figure R1(c) shows how vastly different the behavior of the system becomes with the addition of the second pump. The second pump is added 4 FSRs away from the first one (as in panel (a) of the figure repeated above and marked on by the Reviewer). Instead of 8 very narrow steps in Fig. R1(a), 5 wide steps are clearly visible in the figures, as we expected from the theory described in the text. The lowest-energy step corresponds to no solitons, and the higher-energy ones to 1, 2, 3, and 4 solitons per roundtrip time respectively. All steps are very well-defined with successive runs of the numerical simulation with random initial conditions all resulting in the same steady states with the same comb power at each realization.

In all other panels in Fig. R1, i.e., panels (b, d, e, f), an AMC is added at the frequency suspected by the Reviewer. We have used the same model used in [Herr, *et al.*, Phys. Rev.

Lett. 113, 123901 (2014)] to implement the mode crossing. In this model, the parameter a indicates a measure of the strength of the mode crossing and we have represented it with a_{AMC} (normalized to the cavity FWHM) in Fig. R1, while parameter b (non-dimensional) determines its mode number (essentially, frequency). In Fig. R1, $b = -4$; note that our experimental data are reported vs. wavelength, not frequency, so that this choice matches exactly the mode/wavelength the Reviewer has suspected an AMC may exist in the comment above. Without a second pump and with $a_{\text{AMC}} = 1$, Fig. R1(b), some of the steps in the narrow soliton formation region of Fig. R1(a) show clear signs of destabilization, an observation which matches the findings of [Herr, *et al.*, Phys. Rev. Lett. 113, 123901 (2014)].

In Figs. R1 (d, e, f), a_{AMC} is increasing from 1 to 3, respectively. With two-pump driving and the small value of $a_{\text{AMC}} = 1$, even though the lower-level stairs do not change much, destabilization of the solitons starts to occur noticeably. The destruction of stable soliton steps is especially clear from the top-most step (boxed with a red dashed-line rectangle) and would translate into large subharmonic phase noise. Figures R1(e, f), where $a_{\text{AMC}} = 2$ and 3, demonstrate how by increasing the strength of the AMC, solitons are completely washed away, leaving only the bottom step (no solitons) and a fuzzy trace of the destabilized third step in panel (f). The trend is clear: further increase of the AMC strength would completely eradicate soliton steps. It is worth noting that with smaller a_{AMC} (e.g., $a_{\text{AMC}} = 0.5$) the differences with the no-AMC case were unremarkable, indicating that the impact of AMCs is completely dominated and overshadowed by the cooperation of the two pumps in our experiments, as if AMCs did not exist at all. It should be emphasized that the same destabilizing effect of AMCs is observed (and is even more severe) for $M = 2$ and 3, as the overlaid plots in Fig. R2 show. Additionally, this disruptive effect is also clearly and strongly present when the AMC is 5 FSRs away from the first pump ($b = -5$), as observed in Fig. R3. In our experiments, we have observed stable microcombs with subharmonics in all of these cases ($M = 2, 3$, and 4) which would not have been possible if the AMCs were dominant. This shows that our statements about AMC in the manuscript were indeed correct and accurate.

Taken collectively, these results show that even if AMC were present in our experiments, it would be *completely dominated by the pumps* and would *not* impact subharmonic generation. Most importantly, and contrary to what the Reviewer states, the results discussed above illustrate that AMCs so close to the pumps would only hurt, and definitely not help, subharmonic generation, such that their existence would hinder the emergence of the stable and low phase noise subharmonics that herald DTC formation. This, in turn, more vividly shows the significance and novelty of our approach.

Figure R2. Investigation of the impact of AMC on subharmonic generation for 4 FSRs between the two pumps. Each curve shows the total comb power vs. detuning at steady state (i.e., each data point on each curve refers to one run till steady state, if existent, was reached). (a, b): Single-pump driving without an AMC (a) and with an AMC parametrized with $a_{\text{AMC}} = 1$ (normalized) and $b = -4$. This is an AMC at the frequency the Reviewer has suspected it may exist. (b). As noted in the main text (see Methods), the operation region of our system was to the right of the vertical dashed line in (a), where a single pump cannot sustain stable soliton formation. (c-f): Double-pump driving without (c) and with (d-f) the AMC at $b = -4$. In (d-f) the strength of the AMC is increased from 1 in (d), to 2 in (e), and 3 in (f). As the AMC grows stronger, it leads to the destabilization of the solitons, evident here as the replacement of clean soliton steps by wider regions where the chaotic power of the unstable microcomb may land after hundreds of cavity lifetimes. An example is highlighted by the boxed regions (dashed red) in (c) and (d), where the top-most step has vanished in (d). In (e) and (f) one and two other high-power steps have vanished as well.

Figure R1. Investigation of the impact of AMC on subharmonic generation, similar to Fig. R1, for 2 (a) and 3 (b) FSRs between the two pumps.

Figure R3. Investigation of the impact of an AMC at $b = -5$ on subharmonic generation for 4 FSRs between the two pumps (similar to Fig. R1 but with the AMC affecting a different mode). The destabilization of the solitons even at small a_{AMC} is clear through the vanishing steps; c.f., Fig. R1(c), which shows the same system without AMC.

The discussion presented above has been added to the *Supplementary Information* on pages 4-6 as Section V. SUPPRESSION OF MODE DISRUPTIONS BY THE TWO PUMPS AND THE DESTABILIZING IMPACT OF MODE ANTI-CROSSINGS, starting with

“We had noted in the main text that our experiments ...”

on page 4 and ending with the paragraph

“The results discussed above show that AMCs, especially ...”

on page 6. Figures R1 and R2 have also been added to the *Supplementary Information* as Figures S3 and S4.

Comment (7 – Continued):

- Second, crystalline microresonators are known to have very large number of spatial modes, and as a consequence large number of modal crossings. Please see dispersion measurements of a crystalline microresonator in [<https://doi.org/10.1103/PhysRevLett.113.123901>]. It is quite hard to achieve low number of spatial modes in such photonic platform, because that would need very small size of the protrusion [<https://doi.org/10.1364/OPTICA.2.000221>], and it is certainly not the case with the radius of curvature of 0.2mm. To relieve my doubts and prove that used microresonator has been indeed dispersion engineered to exclude AMCs around pumped region as stated in the manuscript, I would suggest authors to demonstrate dispersion measurements of the cavity in the region of interest, which would help to alleviate my concerns. Currently, from

presented spectra I can only see multiple AMCs in the results, which can contribute to the ordering of the pulses.

Response:

It was explained in detail in response to an earlier part of this comment that we do not expect and did not claim that AMCs were excluded from the resonator. Instead, we claimed that they are weak enough not to interfere with the subharmonic generation process which constitutes the focus of this work and we strongly stand by this statement. This notion is emphasized in the current wording in the Discussion Section (page 5, last paragraph of the second column, in the *previously* submitted version): “In our dichromatically-pumped microresonator, the cavity is engineered to ensure higher-order dispersion and mode anti-crossings do not interfere with soliton formation in the proximity of the pumps.”

Furthermore, the analysis and data in the response to the Reviewer’s preceding comment particularly shows that AMCs so close to the pumps do not help subharmonic generation and their contribution is instead destructive. As a result, dispersion measurements would not help. Additionally, such measurement data would be unnecessary and unrelated to the current work and our conclusions. As we have emphasized both the previous response letters, this work is not a device paper.

It should be added that since two of the authors of the manuscript (Andrey B. Matsko and Lute Maleki) discovered the impact of AMC on Kerr microcomb soliton generation [A. B. Matsko, W. Liang, A. Savchenkov, D. Eliyahu, and L. Maleki, “Optical Cherenkov radiation in overmoded microresonators,” *Opt. Lett.* 41, 2907–2910 (2016)], we are well aware of their effect on nonlinear optical processes in a Kerr cavity. Indeed, the spectrum of a microcavity is never regular because of the large mode number, as the Reviewer notes. However, this irregularity does not change the number of attractors if the deviation of the modes from the “ideal” mode sequence is not large (compared with the FWHM of the modes). The authors have proven this point in their earlier studies, and it is well supported by the added analysis presented above.

Finally, we have calculated the GVD for the relevant modes of our resonator using classical formulas for a spheroidal cavity [M. L. Gorodetsky and Y. A. Demchenko, “Accurate analytical estimates of eigenfrequencies and dispersion in whispering-gallery spheroidal resonators,” in *Laser Resonators, Microresonators, and Beam Control XIV* (SPIE, 2012), Vol. 8236, pp. 416–423.], and found that they closely match cavity modes observed experimentally in the region of interest; see Fig. R4. Furthermore, using this GVD in

Figure R4. Cavity GVD

the numerical simulations, we have obtained results in excellent agreement with the experimental data without adding AMCs. The fact that the theoretical model does not need to take AMCs into account to get robust subharmonic generation is further strong evidence that AMCs are not dominant in our experiments, and this fact is additionally supported by our more detailed analysis above.

Comment (7 – Continued):

- Third, for both experimental spectra demonstrated by the authors, one of the pumps is located at 1544 nm, where the AMX can be located, but masked by this pump. As I wrote above, even in the additional spectrum demonstrated in Supplementary the mode seems to be deviating from the others. The claim that “frequency pinning is absent” clearly incorrect, because comparing the only two provided spectra the line corresponding to the left pump is clearly pinned. As I already mentioned in my initial comment, the data provided in the manuscript are very limited, and do not support the author claims in my view.

Response:

This point has already been addressed in responding to the Reviewer’s first comment under Comment 7 above. Here we reiterate: Even if one of the pumps was placed on an AMC, it would strongly suppress and dominate its adverse impact on subharmonic generation. As we detailed above, and in reference to [Herr, *et al.*, Phys. Rev. Lett. 113, 123901 (2014)] and [A. B. Matsko, *et al.*, Opt. Lett. 41, 2907–2910 (2016)], it is impossible for an AMC so close to a pump (let alone right at a pump frequency) not to destabilize subharmonic generation unless it is negligibly small or totally dominated by the two pumps driving the cavity. Either way, the subharmonic generation process would be dictated by the two pumps alone.

Comment (7 – Continued):

- Fourth, the statement that the impact of AMC is vanishingly small in their microresonator, because they are not pronounced can not be accepted, because AMCs are clearly visible in experimental spectra, and as was shown in multiple resonators in [<https://doi.org/10.1038/s41567-019-0635-0>] crystallization can happen without pronounced AMCs. In principle this can be easily understood from the LLE equation – any deviation from pure D₂-only GVD would cause some oscillations on the CW background leading to the soliton interactions via their extended tails.

Response:

The Reviewer’s referenced study [M. Karpov, M. H. P. Pfeiffer, H. Guo, W. Weng, J. Liu, and T. J. Kippenberg, “Dynamics of soliton crystals in optical microresonators,” Nature

Physics 15, 1071–1077 (2019)] pertains to the effect of AMCs in the presence of 1 pump only and does not consider the significant impact of the second pump. As our foregoing analysis and discussion (surrounding Figs. R1-R3 above) show:

1. AMCs near the pumps, even if present, would have been dominated and overshadowed by the joint impact of the two strong pumps in our study and experiments.
2. Furthermore, in comparison to the paper referenced by the Reviewer, our analysis considers the AMC (i) *much closer* to the main pump (4 or at most 5 FSRs in our study, compared to 15 FSRs away in the paper referenced by the Reviewer), and (ii) with a *comparably strong or more than twice stronger* AMC ($a_{\text{AMC}} = 1$ to 3 in our analysis, compared to $2\Delta/\kappa = 1.3$ in the referenced paper).

Therefore, the crystallization referred to by the Reviewer cannot dominate dichromatic pumping and as we have shown can only have a detrimental effect.

Finally, we have already explained through careful formulation that deviations (such as higher-order dispersion or AMCs) from an ideal D_2 -only GVD profile do not amount to DTC formation and subharmonic generation because they do not define a discrete symmetry in the system; see Sections II and III in the *Supplementary Information*.

Comment (7 – Continued):

- Fifth, as the authors know from previous works on soliton crystals, the AMCs can be located very far from the pump region and contribute to the ordering. Another major concern about the data is that only central part of the comb is shown, which makes the claim of absence of AMCs unsupported.

Response:

Again, what the Reviewer notes applies to systems pumped by a single laser while, as we have shown above, in a two-pump system even AMCs very close to the pumps are suppressed by the joint effect of the two pumps, let alone AMCs much farther away. Indeed, if far away AMCs were strong and not dominated by the two pumps, they would have created observable preferences in the excited subharmonics as we increased the beatnote of the pumps in our experiments. But this effect is clearly not present because in the successive increase of the frequency separation of the pumps (2, 3, and 4 FSRs), we always observe the excitation of subharmonics filling all of the modes of the same family between the two pumps. It should be borne in mind that in our experiments, the pumping region remains the same.

Furthermore, in one of the first results showing comb spectra strongly affected by AMC away from the pump (again in a 1-pump system), the microcomb spectra were *not even stable* when modeled by the single-pump LLE without additional AMCs terms. This is expressly stated in [Del’Haye, A. Coillet, W. Loh, K. Beha, S. B. Papp, and S. A. Diddams,

“Phase steps and resonator detuning measurements in microresonator frequency combs,” Nature Communications 6, 5668 (2015)] (end of page 6, continuing over to page 7):

“While good qualitative agreement can be found between the measurement (Fig. 3a–c) and Fig. 6, it should be noted that this 14-soliton pattern is not stable when using the measured dispersion ($\tilde{D}_2 = 12$ kHz/FSR). For such a dispersion value, the stable solitons are too long, and packing 14 of them in the cavity makes them interact and collide. This indicates the presence of an additional nonlinear mechanism that is not fully understood.”

(Underlining is by us). Here, the authors tried to reproduce an experimental soliton crystal waveform consisting of 14 pulses in a 15-spot lattice on the modulated CW background using the LLE (Fig. 6 in the cited paper), but the waveforms they found were not stable because they did not add AMCs to their LLE model. The “additional nonlinear mechanism” suspected by the authors in the quoted paragraph above was indeed shown later on to be AMC; see [D. C. Cole, E. S. Lamb, P. Del’Haye, S. A. Diddams, and S. B. Papp, “Soliton crystals in Kerr resonators,” Nature Photonics 11, 671 (2017)] by the same group. This is in total contrast with our numerical modeling results where without AMCs (either close or far away from the pumps) we observe stable microcomb and subharmonic generation.

Comment

8. I let myself further express concerns about experimental results besides the ones above. Given very narrow measurement window of the comb spectra the solitonic nature of the combs is questionable (as also pointed out by Reviewer 1). I do not agree that measurements of the beatnote and the optical spectrum are sufficient to claim the presence of soliton pulses in the context of the present manuscript. First, narrow measurement window (only 6 nm) cuts at least half of the information-bearing comb spectrum, and does not let to conclude whether the envelope indeed follows the soliton shape. Second, many strong deviations from the sech^2 envelope profile in multiple places across narrow measurement window, which are clearly seen even in a log-scale, demonstrate significant deviation from the spectral soliton profiles shown in simulations. I would not be surprised if such deviations would be even stronger outside of the measurement window. Third, non-solitonic states can easily provide low-noise performance of the beatnote, as was demonstrated in the first and seminal paper on Kerr combs [<https://doi.org/10.1038/nature06401>]. Thus, given the spectrum uncertainty, the narrow beatnote cannot be used to claim the presence of soliton pulses and their arrangement. Fourth, I would consider the spectrum and beatnote measurements as a prove of solitonic nature if a clear soliton spectral shape was demonstrated, as for example in already cited by the authors paper [Bao et al, 2021], however once authors starting to claim such major advancement and breakthrough in the field of time crystals as they do, one cannot rely on doubtful or handwavy confirmations of key experimental results. I also note that all seminal papers on new **concepts** in the field of soliton microcomb such as demonstration of first solitons [Herr et al, 2014], demonstration of pulsed-pumped solitons [Obrzud et al, 2017] and even soliton crystals [Cole et al, 2017]

provided either FROG or other correlation measurements to undoubtedly demonstrate pulse formation.

To conclude, I do not see a major experimental issue in measuring presence of pulses via FROG or other techniques, and I could not find reasonable explanation of author's reluctance in doing such measurements given the importance of the claims they made in the manuscript.

Response:

We have already responded to the point about such measurements and convinced another Reviewer suggesting time-domain measurements (as confirmed by Reviewer 5) that there is indeed no need for utilizing such techniques as FROG, dispersive Fourier transform, time-stretch, or dispersive temporal interferometry to prove DTC formation. The key signature of DTC formation is the presence of robust sub-harmonic peaks in the Fourier spectrum. In the case of the period-doubling DTCs (the first experimental demonstrations) the amplitude of the sub-harmonic peak is defined as the order parameter which signals entering the time crystal regime [J. Zhang, *et al.*, and C. Monroe, "Observation of a discrete time crystal," Nature 543, 217–220 (2017)]. Therefore, even if the current Reviewer doubts that we precisely get solitons with a sech^2 profile, from the point of view of the formation of DTCs this point does not matter because we undoubtedly observe subharmonic generation, discrete time-translation symmetry is definitely spontaneously broken in our system, and the robust formation of n -tupling steady states in our experiments is unquestionable.

It is noteworthy that even if solitons are measured temporally, the background modulation is much weaker than the solitons and will hence not be observed experimentally in the pulse trace. Therefore, even with temporal pulse measurement techniques, comparison with the temporal modulation of the CW background to prove DTC formation would hardly be possible and TTSB *cannot* be verified using this approach. It would still be necessary to resort to frequency-domain measurements and that is what we have already done here.

The vast majority of peer-review published papers in the microcomb community rely on the stable spectral envelope and narrow beatnote of their experimentally reported microcombs to prove the observation of solitons, just as we have. This is not because they are careless, but because it has been observed and is now an accepted fact that these measurements provide *sufficient corroboration*. Additionally, small deviations from the sech^2 envelope are *commonly* observed. In our experiments, we have a narrow-band comb and hence deviations seem more pronounced, while similar deviations are common in the literature; see, for instance, the spectrum in Fig. 4(a) in the paper cited by the Reviewer, [E. Obrzud, S. Lecomte, and T. Herr, Nature Photonics 11, 600–607 (2017)].

To recap, the key concept we are putting forward in this work is not pulse formation by two pumps or the particular shape of the pulse, but robust subharmonic generation and therefore the narrow beatnotes and small phase noises we have observed in our experiments are ample proof for this notion.

Comment

9. The explanation why the observed soliton ordering can be called “quantum DTC” in the response {9-10} is still unclear to me. First the authors say that Spontaneous breaking of spacetime translation symmetry is a quantum phenomenon without mentioning that it was for years studied in classical systems as well []. Then they are making a statement that consequences of Hamiltonian in classical case is not as dramatic as in quantum case with the example of particle in periodic well. Authors say that “in quantum case realization of a symmetry broken steady state is nontrivial and requires an interacting many-body system”, and in the next sentence they give a definition of classical DTC: “classical DTCs are periodically driven many-body classical systems which break ergodicity and reveal synchronized motion and breaking of time translation symmetry is not an issue, because ...”. I may have to ask to be more concise and straightforward in a sense what a quantum DTCs are: what is their definition and how the manuscript results match the definition. I note, that in the manuscript, on one hand authors assert that they used LLE equation (which is classical) and have demonstrated excellent agreement with their observations, which means that the system is well described via classical approach, but on the other the results are claimed to be “quantum”.

Response:

In the responses to the previous round of Reviewer comments we addressed the question concerning the use of the mean-field approach (i.e. the LLE) in the description of DTCs, and explained quantum symmetry breaking phenomena as well as the consequences of symmetries in classical systems. Nevertheless, we do it again in a slightly different way.

Time-independent classical systems possess continuous time-translation symmetry. This symmetry is respected by a classical system only when it is at rest. Any non-trivial time evolution of a system breaks the time translation symmetry. This fact is not surprising and it is not associated with any spontaneous process. In the case of a periodically driven classical system, equations of motion are invariant when the time parameter is shifted by the period of the driving. Again, there is no requirement that all periodic trajectories must follow the discrete time translation symmetry of the equations of motion. In other words, even a single periodically driven classical particle can evolve along a stable trajectory with a period, e.g., twice longer than the driving period; see, e.g. [Buchleitner, A., Delande, D., & Zakrzewski, J. (2002). “Non-dispersive wave packets in periodically driven quantum systems,” *Physics reports*, 368(5), 409-547].

Now let us switch to the quantum case and, as an illustrative example, let us analyze the original Wilczek’s model of a quantum time crystal [doi:10.1103/PhysRevLett.109.160401]. Wilczek considered attractively interacting bosons on a ring. The ground state (and also any other eigenstate) of such a many-body system is also an eigenstate of the unitary operator that translates all bosons by the same arbitrary distance along the ring because the system possesses the continuous space translation symmetry. Consequently, the single particle probability density must be uniform along the ring [doi:10.1007/978-3-030-52523-1]. Such a symmetry preserving many-body state cannot be described by the mean field approach if the attractive interactions are sufficiently strong because it predicts the ground state in the form of a soliton pulse localized around a certain point on the

ring. However, the mean field soliton pulse emerges from the symmetric many-body ground state when the continuous space translation symmetry is spontaneously broken due to, e.g., measurement of the position of one boson [doi:10.1103/PhysRevLett.119.250602]. Indeed, detection of a single boson at a certain point causes collapse of the wave-function of the remaining particles to a soliton pulse localized around the point of the boson detection. The detection of the boson can happen with the uniform probability density along the ring and consequently the localization of the final soliton pulse is random (spontaneous). Once the soliton pulse is formed, the space translation symmetry is broken and the Bose system can be described by the mean field solution. Wilczek suggested that when the described spontaneous process took place in the presence of a magnetic-like flux, the spontaneously formed soliton would be moving periodically along the ring – it turned out that no motion is possible [doi:10.1103/PhysRevLett.119.250602] but it is a different problem which is not relevant in the current discussion.

To conclude, quantum many-body eigenstates of time-independent systems with space translation symmetry (or Floquet states in the case of periodically driven systems) must fulfill space (or time) translation symmetries because it is required by quantum mechanics. However, it may happen that the symmetry-preserving states are highly vulnerable and any perturbation may lead to spontaneous symmetry breaking. Once the symmetry is broken, the system can be described by the mean-field approach. Thus, it is a routine situation when symmetry broken states of quantum many-body systems are described by mean-field equations and what we have done in this work is perfectly justified.

It is befitting to note that the recently published experimental demonstration of dissipative DTCs in a driven atom cavity (a markedly different system) uses the truncated Wigner approximation (TWA); see [H. Keßler, P. Kongkhambut, C. Georges, L. Mathey, J. G. Cosme, and A. Hemmerich, “Observation of a Dissipative Time Crystal,” Phys. Rev. Lett. 127, 043602 (2021)]. The TWA is the mean-field approach with small quantum fluctuations included. That is, the classical fields are evolved within the mean-field equations with an additional stochastic term that mimics quantum fluctuations. The quantum fluctuations must be small, otherwise the TWA would not be valid. In other words, their experiments too can certainly be qualitatively well described by the mean-field approach. Addition of small quantum fluctuations is shown to not alter Kerr combs [Y. K. Chembo, “Quantum dynamics of Kerr optical frequency combs below and above threshold: Spontaneous four-wave mixing, entanglement, and squeezed states of light,” Phys. Rev. A 93, 033820 (2016)].

Comment

Furthermore, claim “quantum timing jitter of the generated soliton train is determined by the quantum frequency noise of the pumps” is not clear and is not supported. First, in contrast to cited work [Bao, 2021] authors didn’t measure their jitter noise down to a quantum noise level. Second, the results shown in the cited paper referred to the noise of two counterpropagating solitons generated from the same pump, which is AOM shifted for counter propagating modes by different amount and thus intrinsically highly coherent. While in the present manuscript the pumps originate from

different laser cavities and are not shown to be correlating down to quantum level [doi: 10.1038/ncomms8957].

Response:

We have shown that the DTC (not necessarily a soliton train) is phase locked to the pair of pumps creating the subharmonics. Our numerical simulation confirms the experimental observation. Because the Kerr comb experiences unavoidable phase diffusion (a quantum process) and since phase locking of an oscillator to a better oscillator suppresses the phase diffusion of the former, we claim that the quantum noise associated with the Kerr process can be reduced in the DTC. What occurs in this case is in principle what happens in phase noise reduction by frequency division, which is an established phenomenon [T. M. Fortier, *et al.*, “Generation of ultrastable microwaves via optical frequency division,” *Nature Photonics* 5, 425 (2011)]. It does not matter from which cavities the lasers originate; the phase noise of the subharmonic beatnote will be smaller than the phase noise of the beatnote between the pumps if they are truly generated by and locked to the two pumps. The main requirement for the quantum noise reduction is for the beatnote of the lasers to produce less frequency noise than the Kerr frequency comb does.

Comment

10. Regarding whether the pulse ordering of the solitons due to modulated background, can be claimed a DTC and how it is different from soliton crystals:

a. As authors correctly pointed out the time crystals are being a topic of debates since their first suggestion by Wilczek in 2012. As I can see from the manuscript and author explanation, a rigorous definition of the time crystallinity has not been proposed to the reader and no theoretical criteria-matching work has been carried out to confirm that the observation can be claimed discrete time crystals. The definition given in the introduction corresponds to the *closed* systems (*isolated* in the manuscript), and then is being immediately reduced to just “A discrete time crystal (DTC) emerges when the period of the system response is an integer multiple of the drive period”. This form is used for the rest of the manuscript. The prove that the observed system can indeed be classified as DTC in the manuscript is mainly focused on explanation that certain ordered soliton states obtained in simulations or potentially observed in experiments reveal a sign of symmetry breaking. I find such approach to be dangerously superficial, as it misleads not only authors but the readers suggesting that a single not-distinctive signature of the phenomena can be used to claim its observation.

Response – part (a):

Spontaneous breaking of discrete time translation symmetry into another discrete time translation symmetry in a periodically driven quantum many-body system corresponds to the formation of DTCs. All of the conditions which have to be fulfilled in order to observe such a phenomenon have already been mentioned throughout the manuscript, with careful referencing to the relevant literature (e.g. see the 2nd, 3rd, and 4th paragraphs of the manuscript) and described extensively in our

previous reply to the Reviewer #2 (see the text starting with “Below we again reiterate the general elements needed to call a system a DTC ...” on page 6 of the previous Response Letter). For instance, discrete time translation symmetry breaking as a criterion for DTC formation was mentioned on page 1, in columns 1 and 2 (“... A discrete time crystal (DTC) emerges when the period of the system response is an integer multiple of the drive period.”) while rigidity and long-range order was emphasized in the first column on page 2 (“... , and illustrating that these photonic DTCs possess temporal long-range order and can be realized robustly over a range of system parameters.”). Here, we repeat these conditions, now as a numbered list:

1. The time-dependent term in the equation of motion possesses discrete time translation symmetry (true in our case and clearly shown using the equations in the Methods Section), yet its steady-state solutions do not evolve with the period dictated by the drive but with an integer multiple of the driving period (true in our case evidenced by the emergence of subharmonic between the two pumps).
 2. The symmetry broken steady states must be stable solutions over a range of parameters and no fine tuning is necessary (true in our case and evident from robust subharmonic generation in 2-, 3-, and 4-FSR separation of the pumps; also supported by numerical modeling data for our system).
 3. The many-body system has to be in the thermodynamic limit (true in our case, with a large number of photons in the cavity) otherwise the life-time of symmetry broken states is not infinite. In other words, symmetry broken states can be very accurately described by the mean-field approach (true in our case, because the mean-field modified LLE quite accurately describes system).
- The above criteria apply both to closed and dissipative systems. The difference between these two kinds of DTCs relies on the difference in energy transfer from the periodic drive. In the case of closed systems, the average transfer of energy to a system due to periodic driving is zero. In the case of dissipative systems, however, the average transfer is not zero but the system still forms a periodically evolving steady state because it releases energy due to damping. Theoretical research of DTCs both in closed and dissipative systems is nowadays developing very rapidly and the relevant literature is cited in our manuscript where the criteria of the formation of DTCs can be also found.

To ensure the Reviewer’s concern is fully addressed, the above-mentioned criteria and how they are met in our system are carefully laid out at the beginning of the Discussion Section (page 5):

“A periodically forced dissipative many-body system qualifying as a DTC meets certain criteria. First, it possesses discrete TTS, evident from the time-dependence of its equation of motion. However, steady states (i.e., steady-state solutions of the equation of motion) evolving with a cycle that is an integer multiple of the period dictated by the drive can emerge spontaneously in the system. Second, the symmetry-broken steady states emerge without relying on fine tuning and are stable over a range of parameters. Third, the many-body system is in the thermodynamic limit and the symmetry-broken states can be accurately described by a mean-field approach. Subharmonic generation in the dichromatically pumped Kerr microcavity system introduced in this work qualifies

as a dissipative DTC because it meets all of the said criteria. The two pumps define a discrete TTS in the system, manifest in the equations of motion (detailed in Methods), which is broken by the realization of certain periodic steady states of the system. These states, accompanied by subharmonic generation, demonstrate SSB at integer response to drive periodicity ratios. The subharmonics in symmetry broken states emerge robustly, both in numerical modeling and experiments, with various (e.g., 2-, 3-, and 4-FSR) frequency separations of the pumps and possess temporal long-range order; see also *Supplementary Information* Section V. Finally, the DTCs are in the thermodynamic limit of infinitely many photons and are well captured by a mean-field model (the modified LLE, see Methods).”

b. No definition of Dissipative DTC is being proposed in the manuscript. The authors silently imply that dissipative DTC is a DTC, which has just been defined by themselves as the state where the period of the system response is an integer multiple of the drive period, but with a dissipation. The authors recognize both of these features in parametrically driven damped NLSE and claim that the ordered soliton states in such systems can be claimed dissipative DTCs. Following such logic one can name “dissipative DTC” just a periodically driven pendulum experiencing period doubling bifurcation, which is obviously not the case, because no new *phase state of matter* exist here. The same would apply to other nonlinear parametrically driven systems revealing the response at multiple integer of the drive.

Response – part (b):

Dissipative DTCs have already been defined in the literature (as our manuscript bibliography shows). Therefore, there is no need for devising a new definition and we are certainly not redefining anything in this manuscript. Most of the referee’s objections described in point (b) have been already addressed in our response to point (a). Here, let us only address the Reviewer’s suggested analogy of our dissipative DTC to a periodically driven pendulum (assuming a *quantum* periodically driven particle is what the Reviewer is referring to, not a classical pendulum like a ball hanging by a string from a fixed pivot point). Space crystals and time crystals are quantum many-body systems with many degrees of freedom, yet when we look at them again and again we still see the same crystal. That is, detection of positions of a small fraction of particles does not destroy the crystalline state of a quantum system. A periodically driven pendulum is a single particle system (or a single degree of freedom) and when we perform measurement of the position of a particle, the initial quantum state is totally destroyed. Therefore, we need a many-body quantum system if we want to realize a “phase state of matter” and obviously a periodically driven pendulum does not fulfil this criterion.

c. To confirm the above, I would like to bring to authors attention the review on Discrete time crystals [<https://doi.org/10.1146/annurev-conmatphys-031119-050658>], which has been already cited by more than 80 times since 2020, and which while being one of the key papers in the field, to my surprise, was ignored by the authors:

“it might be tempting to classify several superficially similar-looking nonequilibrium phenomena (e.g., period doubling bifurcations, second harmonic generation, Faraday waves, etc.) as time crystals (5, 13–15). However, such time-dependent phenomena generally cannot be classified as phases of matter and share few similarities with equilibrium order.”

Response – part (c):

We are well familiar with the content of the referenced paper. This paper is a review on DTCs in closed systems, which is not directly related to our system of interest. We have included proper citations to relevant literature already and due to the limit on the bibliography size have naturally cited only the most relevant papers. Especially with respect to the literature on DTCs, we believe our bibliography is quite generous. Particularly, we have cited other comprehensive review articles and more directly germane papers reporting original results.

The quote from the paper is a correct general statement and we agree with it. We would like to note that we have presented our results reported in this manuscript in the most comprehensive conference held on time crystals to date (see <http://indico.ictp.it/event/9504/>), where almost all major players in the field were present, and we have only received encouraging feedback from these researchers.

d. Another important feature of a DTC, which was not mentioned by the authors and was not verified in their work is that the system is expected to reveal the spontaneous symmetry breaking in *its lowest energy state*. Lowest energy state of the bichromatically driven system is not a state with multiple solitons but, depending on the detuning and system parameters, is either a modulated background or a single soliton state, which both don't reveal any symmetry breaking as confirmed by the authors.

Response – part (d):

We like to emphasize one of the most fundamental properties of Floquet systems. DTCs form in periodically driven systems. For periodically driven systems, it is not possible to calculate energy eigenvalues because energy is not conserved. One can calculate *quasi*-energies of a periodically driven system which are eigenvalues of the Floquet Hamiltonian, i.e., Hamiltonian supplemented by the differential operator $-i\hbar(d/dt)$, see e.g. [doi:10.1007/978-3-030-52523-1] or Chapter 5 in [Dittrich, *et al.*, Quantum transport and dissipation. Vol. 3. Weinheim: Wiley-VCH, 1998]. However, the quasi-energy spectrum is periodic with the period $\hbar\omega$ (where ω is the driving frequency) and consequently there is no ground state because the spectrum is not bounded from below. Briefly speaking, it is well known that in the case of DTCs, no ground state exists.

e. I would like to draw the author's attention to recently published Nature Communications paper [<https://doi.org/10.1038/s41467-021-23172-2>] on the same concept of ordering microcavity solitons using two pumps (not self-injection locked though), named “synthesized soliton crystals”. I would like to highlight two aspects – first is the quality of data, demonstrating the *concept*, and the

ability of the authors reproducibly realize crystals with different separations, and second – is the absence of any qualitative distinction claimed by the authors between dichromatically synthesized and native soliton crystals.

Response – part (e):

We are aware of this recent article. As noted in our response to the Reviewer’s Comment 3 above, this paper, while valuable in its own right, does not demonstrate subharmonic generation and is therefore not germane to DTCs. As a result, there is no surprise that they have called their microcombs synthetic soliton crystals. We note that this paper cites our earlier theoretical formulation of dually-pumped Kerr microcombs twice, both in the main text and in the paper’s *Supplementary Information*.

We would like to draw the Reviewer’s attention to two points in panel (c) in the Supplementary Figure 1 in this paper, copied below in Fig. R5, which depicts measured integrated dispersion D_{int} as a function of mode number:

- (i) Some data points (empty red circles) near mode number 0 are missing;
- (ii) Notwithstanding the missing data points, the deviations of the existing data points in the plot from the fitted dispersion (dashed blue curve). In other words, some modes show erratic behavior, which could very well be a *signature of avoided mode crossings (AMCs)*.

This observation serves as further evidence for our argument and data, presented above in response to Comment 7: weak AMCs do not contribute to subharmonic generation (as the cited paper has not reported subharmonics between the two pump) and, more importantly, the two pumps can indeed dominate and suppress weak modal irregularities such as AMCs.

Figure R5. Panel (c) in Supplementary Figure 1 from [Lu, et al., Nat. Commun. 12, 2021], <https://doi.org/10.1038/s41467-021-23172-2>, referenced by the Reviewer in Comment 10(e). Measurement data near mode number 0 are sparse, while clear signs of AMC-like dispersion disruption are present.

f. To resolve the question of what can be called time crystal (discrete time crystal, dissipative discrete time crystal), and whether the claimed observation of pulse ordering in the manuscript should

be distinguished from the soliton crystals, I suggest to the authors give their concrete definitions (without pulling together “convenient” 1-2 matching signatures from literature) and implement detailed criteria-matching work. As one of the prominent examples of such careful work in the field of time crystals I can refer the authors to the work by Sullivan *et al*, which first appeared in arxiv (<https://arxiv.org/abs/1807.09884>) with the title “Dissipative discrete time crystals”. However, after detailed investigation of their results it was later published in New Journal of Physics with changed title “Signatures of discrete time crystalline order in dissipative spin ensembles”. I consider this as responsible and honest move.

Response – part (f):

In our response to point (a) of this Comment, we have presented the conditions for DTCs and confirmed that they are fulfilled by our system. In that response, we also noted that these criteria were already included in the text of the manuscript. Concerning the paper by Sullivan *et al.*, which is pointed out by the referee, we would like to let the Reviewer know that shortly after posting our manuscript on arXiv, we received an encouraging note from the corresponding author of the paper. This work is cited as Ref. [26] in our manuscript.

Comment:

As a general remark, I suggest authors to be straighter and more succinct in their further replies. It was often admittedly hard to follow even simple logic of their explanations hidden in the ornate replies spiced with multiple-times iterated statements of novelty and uniqueness of the manuscript results. Being more straightforward and clear would definitely help authors to make less inaccuracies and contradictions, as well as would not require me to follow up with even longer reply. In conclusion, I must note that all above represents merely a half of questions and concerns, which I got after reading the authors response. This is very unusual case for my peer-reviewing practice of many years, which makes me express my concerns rather strongly. I suggest the authors to carefully redo their experiments following mine and other Reviewers concerns, provide more representative data, and carefully revise their claims on the observations of time-crystallinity. In suggesting this, I am particularly worrying for the professional reputation of the authors and would not like to see similar case as to recent *Science* publication [DOI: 10.1126/science.abl5286] happen to them.

Response:

We appreciate the Reviewer’s concern. They posed many questions and addressing them naturally led to a response letter of several pages. However, we included a summary of the rebuttal at the beginning of the letter which they may find more useful. Evidenced by the results contained in the manuscript and the responses to their comments above, we are quite confident in the validity of our data and arguments.

REVIEWERS' COMMENTS

Reviewer #5 (Remarks to the Author):

The authors have modified the paper by adding some appropriate comments about the issue of repeatability and the connection with modulational instability. In my opinion the paper is now suitable for publication in Nature Communications.